# ACLY inhibition promotes tumour immunity and suppresses liver cancer

Jaya Gautam[1,2,14], Jianhan Wu[1,2,14], James S. V. Lally[1,2,14], Jamie D. McNicol[3], Russta Fayyazi[1], Elham Ahmadi[1,4], Daniela Carmen Oniciu[2,5], Spencer Heaton[2], Roger S. Newton[2], Sonia Rehal[1], Dipankar Bhattacharya[6], Fiorella Di Pastena[1,2], Binh Nguyen[7,8], Celina M. Valvano[1], Logan K. Townsend[1], Suhrid Banskota[1], Battsetseg Batchuluun[1], Maria Joy Therese Jabile[1], Alice Payne[1], Junfeng Lu[1], Eric M. Desjardins[1], Naoto Kubota[9], Evangelia E. Tsakiridis[1], Bejal Mistry[1], Alex Aganostopoulos[1], Vanessa Houde[1], Ann Dansercoer[10,11], Koen H. G. Verschueren[10,11], Savvas N. Savvides[10,11], Joanne A. Hammill[3], Ksenia Bezverbnaya[3], Paola Muti[1,4], Theodoros Tsakiridis[1,4], Wenting Dai[12], Lei Jiang[12], Yujin Hoshida[9], Mark Larché[7,8,13], Jonathan L. Bramson[3], Scott L. Friedman[6], Kenneth Verstraete[10,11], Dongdong Wang[1] & Gregory R. Steinberg[1,2✉]

Immunosuppressive tumour microenvironments are common in cancers such as metabolic dysfunction-associated steatohepatitis (MASH)-driven hepatocellular carcinoma (HCC) (MASH-HCC)[1-3]. Although immune cell metabolism influences effector function, the effect of tumour metabolism on immunogenicity is less understood[4]. ATP citrate lyase (ACLY) links substrate availability and mitochondrial metabolism with lipid biosynthesis and gene regulation[5-7]. Although ACLY inhibition shows antiproliferative effects in various tumours, clinical translation has been limited by challenges in inhibitor development and compensatory metabolic pathways[8-12]. Here, using a mouse model of MASH-HCC that mirrors human disease, genetic inhibition of ACLY in hepatocytes and tumours reduced neoplastic lesions by over 70%. To evaluate the therapeutic potential of this pathway, a novel small-molecule ACLY inhibitor, EVT0185 (6-[4-(5-carboxy-5-methyl-hexyl)-phenyl]−2,2-dimethylhexanoic acid), was identified via phenotypic screening. EVT0185 is converted to a CoA thioester in the liver by SLC27A2 and structural analysis by cryo-electron microscopy reveals that EVT0185-CoA directly interacts with the CoA-binding site of ACLY. Oral delivery of EVT0185 in three mouse models of MASH-HCC dramatically reduces tumour burden as monotherapy and enhances efficacy of current standards of care including tyrosine kinase inhibitors and immunotherapies. Transcriptomic and spatial profiling in mice and humans linked reduced tumour ACLY with increases in the chemokine CXCL13, tumour-infiltrating B cells and tertiary lymphoid structures. The depletion of B cells blocked the antitumour effects of ACLY inhibition. Together, these findings illustrate how targeting tumour metabolism can rewire immune function and suppress cancer progression in MASH-HCC.

Cancer cells reprogram their metabolism to support unchecked growth and to evade immune surveillance[13,14]. These metabolic adaptations often involve enhanced glycolysis and de novo lipogenesis, enabling sustained biomass production and proliferation. At the same time, these pathways reshape the tumour microenvironment by limiting the availability of key nutrients, and by generating immunosuppressive byproducts such as lactate and succinate[13,14]. As a result, tumour metabolism not only fuels intrinsic cancer cell growth but also contributes to immune evasion. HCC exemplifies the interplay between metabolic dysfunction and immune suppression. Although once primarily linked to viral hepatitis, alcohol and toxins, HCC is increasingly driven by MASH[1,15,16]. MASH-HCC arises in a distinctly immunosuppressive hepatic microenvironment that is resistant to immunotherapy compared with viral-driven disease[1,15,16]. Although recent clinical observations suggest that features such as intratumoural steatosis[17], reduced B cell infiltration[3] and downregulation of the chemokine CXCL13 (ref. 17) may influence immune responsiveness in MASH-HCC, the mechanisms linking tumour metabolism to immune regulation remain poorly defined.

ACLY functions at a key metabolic junction, converting citrate into acetyl-CoA and oxaloacetate, thereby linking carbohydrate availability to fatty acid and cholesterol synthesis and histone acetylation[5-7]. Through these roles, ACLY supports both the anabolic growth demands of tumour cells and epigenetic regulation of gene expression[5-7]. Although ACLY inhibition has been shown to suppress tumour

proliferation in many preclinical models[18–20], its therapeutic potential has been questioned due to compensatory upregulation of alternative acetyl-CoA-generating pathways, including acetate CoA synthetase 2 (ACCS2)[8,10], pyruvate dehydrogenase (PDH)[9,10], pantothenate kinase 2 (PANK2)[11] and fatty acid oxidation[12], that may bypass ACLY dependency. However, most studies have largely relied on cell lines and xenograft models that lack intact immune systems and do not reflect the immune-metabolic complexity of solid tumours such as MASH-HCC. Of note, although ACLY is upregulated in HCC and reduced expression correlates with improved survival, previous studies have focused predominantly on viral-related or toxin-related HCC[1,21–25], leaving its role in MASH-HCC unexplored. Given the immunosuppressive nature of the MASH-HCC microenvironment and the emerging links with metabolism, we hypothesized that ACLY may serve as a key metabolic regulator of tumour–immune interactions.

## A new mouse model of MASH-HCC

To model MASH-HCC, male C57BL/6J mice were injected with diethyl nitrosamine (DEN) and then maintained on a control chow (control-DEN) or a high-fat/fructose Western diet with physiologically relevant levels of cholesterol[26] (WD-DEN) starting at 8 weeks for 28 weeks (Extended Data Fig. 1a). Compared with control-DEN mice, WD-DEN mice showed increased liver weights, liver:body weight ratios and significantly elevated levels of plasma α-fetoprotein (AFP), a marker of HCC (Extended Data Fig. 1b–d). Histological analysis confirmed enhanced steatosis, ballooning and inflammation in WD-DEN livers (Extended Data Fig. 1e,f), along with a marked increase in surface tumours and tumour burden (Extended Data Fig. 1g–k). Lesions were histologically classified as non-neoplastic or neoplastic, the latter showing features typical of human MASH-HCC, including macrovesicular or microvesicular steatosis, ballooning, Mallory–Denk bodies and lymphocyte infiltration (Extended Data Fig. 1l). WD-DEN mice developed more neoplastic lesions (Extended Data Fig. 1m) with significant macrovesicular or microvesicular steatosis, ballooning and lymphocyte infiltration that were pathologically similar to those of humans with MASH-HCC[27] (Extended Data Fig. 1n). To assess relevance to human MASH-HCC, we performed bulk RNA sequencing of tumours and compared gene expression with patient samples and two other models: control-DEN[28] and WD-CCl₄ (FAT-MASH)[29] (Extended Data Fig. 1o). The majority (8 out of 9) of WD-DEN tumours uniquely matched the Hoshida S1 subtype found in patients with MASH-HCC, whereas control-DEN and WD-CCl₄ aligned with S2 or S3 subtypes (Extended Data Fig. 1p,q). Single-cell RNA sequencing (scRNA-seq) validated immunological similarity, with lymphocytes and AFP⁺ and GPC3⁺ tumour cells in WD-DEN and WD-CCl₄ mice having similar correlation coefficients to humans with MASH-HCC (Extended Data Fig. 1r). *Acly* expression was broadly distributed across cell types in WD-DEN livers, mirroring the human condition (Extended Data Fig. 1s,t). Together, these findings demonstrate that the WD-DEN model recapitulates key pathological, molecular and immune features of advanced human MASH-HCC.

## *Acly* genetic inhibition reduces MASH-HCC

Having validated the relevance of the WD-DEN model, we replicated the study using *Acly*-floxed (*Acly^{fl/fl}*) mice[8]. To control for baseline tumour burden, mice were stratified by plasma AFP levels at 28 weeks, and two groups with equivalent AFP levels were injected with hepatocyte-targeted adeno-associated virus 8 (AAV8)-TTR vectors expressing either YFP (wild type (WT)) or Cre recombinase (*Acly* knockout (KO)) to induce hepatocyte-specific *Acly* deletion (Fig. 1a and Extended Data Fig. 2a,b). Eight weeks post-injection, immunohistochemistry showed marked reductions in ACLY expression in both tumoural and non-tumoural hepatocytes of *Acly*-KO mice, but not in infiltrating immune or mesenchymal cells such as endothelial and

Kupffer cells (Fig. 1b–e). Pathology analysis showed similar scores for liver steatosis, ballooning and inflammation between genotypes (Extended Data Fig. 2c), but biochemical assessment of non-tumour liver tissue revealed reduced levels of several fatty acids in *Acly*-KO mice (Extended Data Fig. 2d). Despite comparable liver pathology, *Acly*-KO mice exhibited markedly fewer surface tumours (Fig. 1f–i) and reduced tumour surface area (Extended Data Fig. 2e) than WT controls. Fewer than 30% of *Acly*-KO mice had more than 30 surface tumours, versus 90% of WT mice (Fig. 1h). Histological classification indicated that this reduction was driven primarily by an approximately 70% decrease in neoplastic lesions, with many *Acly*-KO mice having none (Fig. 1i), alongside a trend towards fewer non-neoplastic proliferative lesions ($P = 0.0563$; Extended Data Fig. 2f). Neoplastic lesions in *Acly*-KO mice had reduced steatosis (Fig. 1j), consistent with smaller lipid droplet size (Extended Data Fig. 2g), lower tumour lipid area (Fig. 1k) and reductions in fatty acids (Fig. 1l). These findings demonstrate that hepatocyte-specific *Acly* deletion in a physiologically relevant MASH-HCC model reduces tumour number, size, severity and lipid content.

## Identification of the ACLY inhibitor EVT0185

Over the past three decades, several ACLY inhibitors have been described[5]. Bempedoic acid competitively inhibits ACLY activity and allosterically activates AMP-activated protein kinase-β1 (AMPKβ1)-containing complexes when converted to bempedoic acid-CoA in hepatocytes[30]. In mice, it lowers MASH[26], and in phase III studies, reduces LDL-cholesterol and cardiovascular events[31]. Potent allosteric ACLY inhibitors have also been developed[32], but poor bioavailability and/or cell permeability have hindered development[5]. To identify more potent and cell-permeable ACLY inhibitors, we conducted a phenotypic screen in primary mouse hepatocytes, evaluating fatty acid and cholesterol synthesis across a library of compounds varying in hydrocarbon chain length, terminal functional groups and core chemical substitutions. Several candidates inhibited lipogenesis (Supplementary Table 1 and originally disclosed in ref. 33), with EVT0185, selected for further study due to strong inhibition of de novo lipogenesis (−84% at 100 μM, half-maximal inhibitory concentration ($IC_{50}$) = 0.46 μM; Fig. 2a). Like bempedoic acid, EVT0185 was converted to EVT0185-CoA in rat liver microsomes (Extended Data Fig. 3a–c). Because HEK293 cells lack long-chain acyl-CoA synthetase activity[34], we engineered stable lines expressing *SLC27A1* (also known as *FATP1* and *ACSVL5*), *SLC27A2* (also known as *FATP2* and *ACSVL1*), *SLC27A4* (also known as *FATP4* and *ACSVL4*) and *SLC27A5* (also known as *FATP5* and *ACSVL6*) to identify the activating enzyme (Extended Data Fig. 3d,e). Incubation of HEK293 cells with EVT0185 followed by mass spectrometry analysis indicated that only cells expressing *SLC27A2* generated the CoA thioester (Extended Data Fig. 3f–h). Transcriptomic analyses revealed that *SLC27A2* is upregulated in livers of people with MASH or MASH-HCC compared with healthy controls (Extended Data Fig. 3i), with no expression in immune cells per the protein atlas and single-cell datasets (Extended Data Fig. 3j). Consistent with this, EVT0185 had no effect on B cell or T cell proliferation in vitro (Extended Data Fig. 3k–o). These data indicate that EVT0185 is converted to a CoA thioester by SLC27A2, in the liver and tumours but not in immune cells of MASH-HCC.

ACLY is inhibited by palmitoyl-CoA[5], therefore we hypothesized that the conversion of EVT0185 to its CoA thioester may drive the inhibition of ACLY activity. In cell-free assays, EVT0185-CoA (not the unconjugated diacid) inhibited recombinant human ACLY (hACLY) activity (Fig. 2b) and this effect was competitive with CoA (Fig. 2c,d and Extended Data Fig. 4a). hACLY is a 1,101-residue polypeptide forming a functional 0.5-MDa tetramer and featuring an N-terminal citryl-CoA synthetase (CCS) module, consisting of CCSβ and CCSα regions, and a C-terminal citrate synthase homology (CSH) domain that serves as the oligomerization platform of the ACLY enzyme and contains the

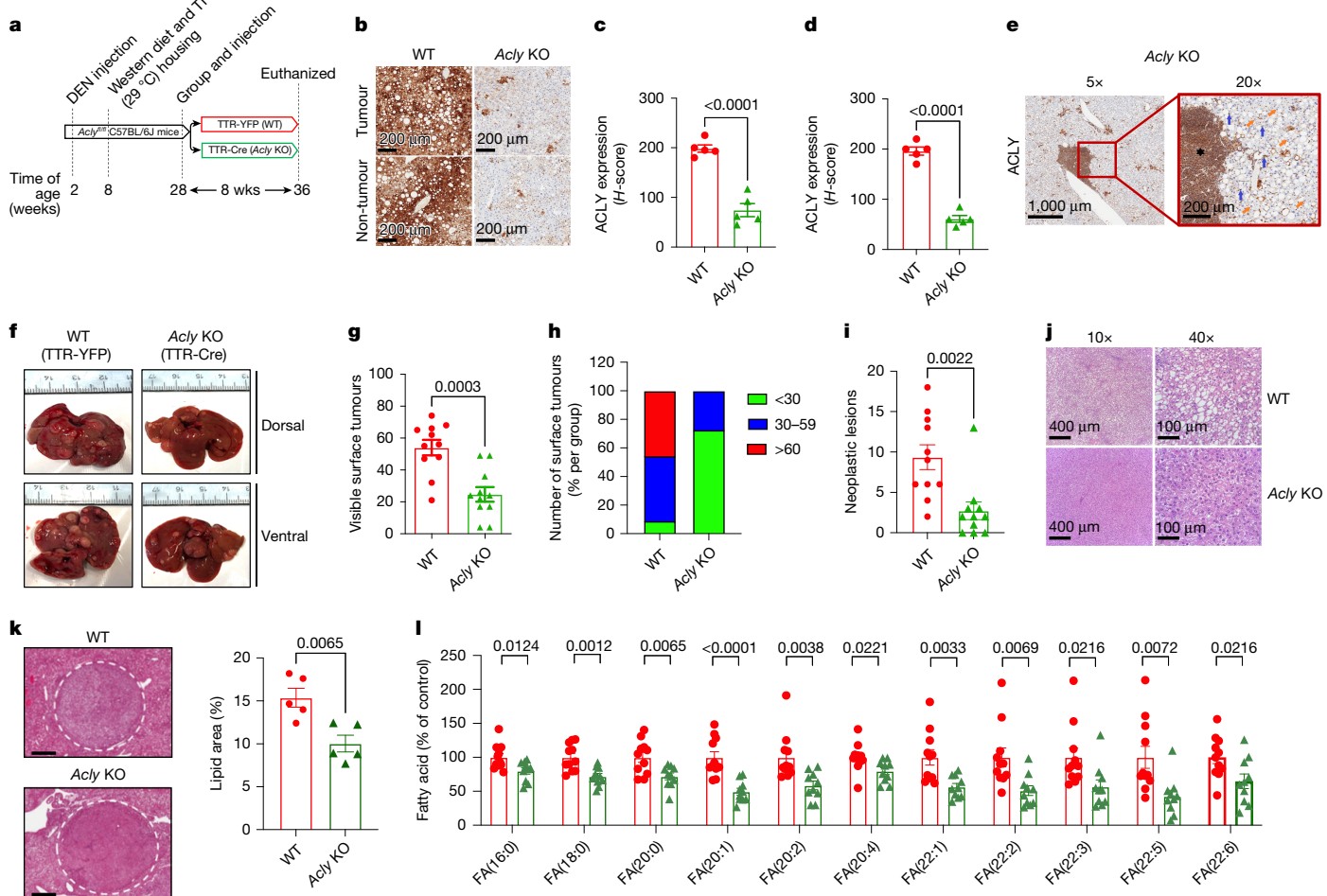

**Fig. 1 | Genetic inhibition of *Acly* reduces tumour burden in a mouse model of MASH-HCC. a**, Experimental scheme of the *Acly*-KO model. TN, thermoneutral. **b**, Representative images of ACLY protein expression in WT and hepatocyte-specific *Acly*-KO liver and tumours. **c**,**d**, The *H*-score of ACLY protein expression in tumour (**c**) and tumour-adjacent liver (**d**). Data are presented as mean ± s.e.m. *n* = 5 *Acly*-KO versus *n* = 5 WT livers. *P* values were determined by two-tailed, unpaired Student's *t*-test: *P* = 3.54 × 10⁻⁵ (**c**) and *P* = 1.23 × 10⁻⁶ (**d**). **e**, Representative *Acly*-KO mouse liver stained with ACLY antibody. The asterisk indicates inflammatory cell aggregation-positive ACLY staining; the blue arrows denote mesenchymal cell (endothelial or Kupffer cells)-positive ACLY staining; and the orange arrows show hepatocyte-negative ACLY staining. **f**, Representative dorsal and ventral images of WT and *Acly*-KO liver with tumours. **g**, Visible number of tumours on the liver surface. Data are presented as mean ± s.e.m. *n* = 11 WT and *n* = 11 *Acly*-KO mice. *P* values were determined by unpaired, two-tailed Student's *t*-test. **h**, Percentage distribution graph of tumour numbers from livers of WT (*n* = 11) and *Acly*-KO (*n* = 11) mice. **i**, Number of neoplastic lesions in livers from WT (*n* = 11) and *Acly*-KO (*n* = 11) mice. Data are presented as mean ± s.e.m. *P* values by unpaired, two-tailed Student's *t*-test. **j**, Representative images of neoplastic lesions from the livers of WT and *Acly*-KO mice. **k**, Representative images showing tumour with lipid (left) and calculation of the percentage lipid area of WT (*n* = 5) and *Acly*-KO (*n* = 5) livers (right). Scale bars, 600 μm. Data are presented as mean ± s.e.m. *P* values were determined by unpaired, two-tailed Student's *t*-test. **l**, Fatty acids (FAs) in tumours from WT (*n* = 11) and *Acly*-KO (*n* = 10) mice. C:D, the total number of carbon atoms to the number of carbon–carbon double bonds. Data are presented as mean ± s.e.m. *P* values were determined by unpaired, two-tailed Student's *t*-test.

CoA-binding site[32,35] (Fig. 2e). The catalytic itinerary of ACLY starts in the CCS module, which catalyses the ATP-driven formation of citryl-CoA (Fig. 2e). Shuttling of this high-energy reaction intermediate to the CSH module of ACLY, facilitated by flipping of the long pantothenyl-arm of citryl-CoA, positions the citryl-thioester moiety in the CSH active site where citryl-CoA undergoes retro-aldol cleavage into oxaloacetate and acetyl-CoA[35,36]. To directly examine the molecular interactions by which EVT0185-CoA inhibits hACLY, the ACLY−(*R*,*S*)-EVT0185-CoA complex was characterized using single-particle cryo-electron microscopy (cryo-EM) analysis (Fig. 2f and Extended Data Fig. 4b). A 3D reconstruction without the application of symmetry had a resolution of 3.7 Å following a gold-standard refinement (Extended Data Fig. 4c) and revealed a pseudo-D2-symmetric ACLY assembly where four CCS modules arrange around the central CSH module (Fig. 2g). Markedly, the cryo-EM map for one CCS module was poorly defined,

indicating structural heterogeneity in this region. Inspection of the cryo-EM map revealed clear density corresponding to the ligand at all four CoA-binding sites situated at the interface between the four CCS modules and the central CSH module of the ACLY holoenzyme (Extended Data Fig. 4d). To resolve this pseudo-D2-symmetric assembly at a higher resolution, we applied symmetry expansion in combination with local refinement followed by 3D classification (Extended Data Fig. 4c). This approach improved the overall map quality (Fourier shell correlation of 0.143 (FSC₀.₁₄₃) = 3.3 Å; Fig. 2h,i, Extended Data Fig. 4e and Supplementary Table 2) and ligand density, which we interpreted as representing the adenosine 3′-phosphate 5′-diphosphate moiety of bound EVT0185-CoA with the pantothenyl arm and dicarboxylic acid moiety disordered (Fig. 2j). On the basis of this experimentally observed binding mode, which is analogous to the binding of CoA to ACLY (Extended Data Fig. 4f,g), we speculate that ACLY-bound EVT0185-CoA may adopt

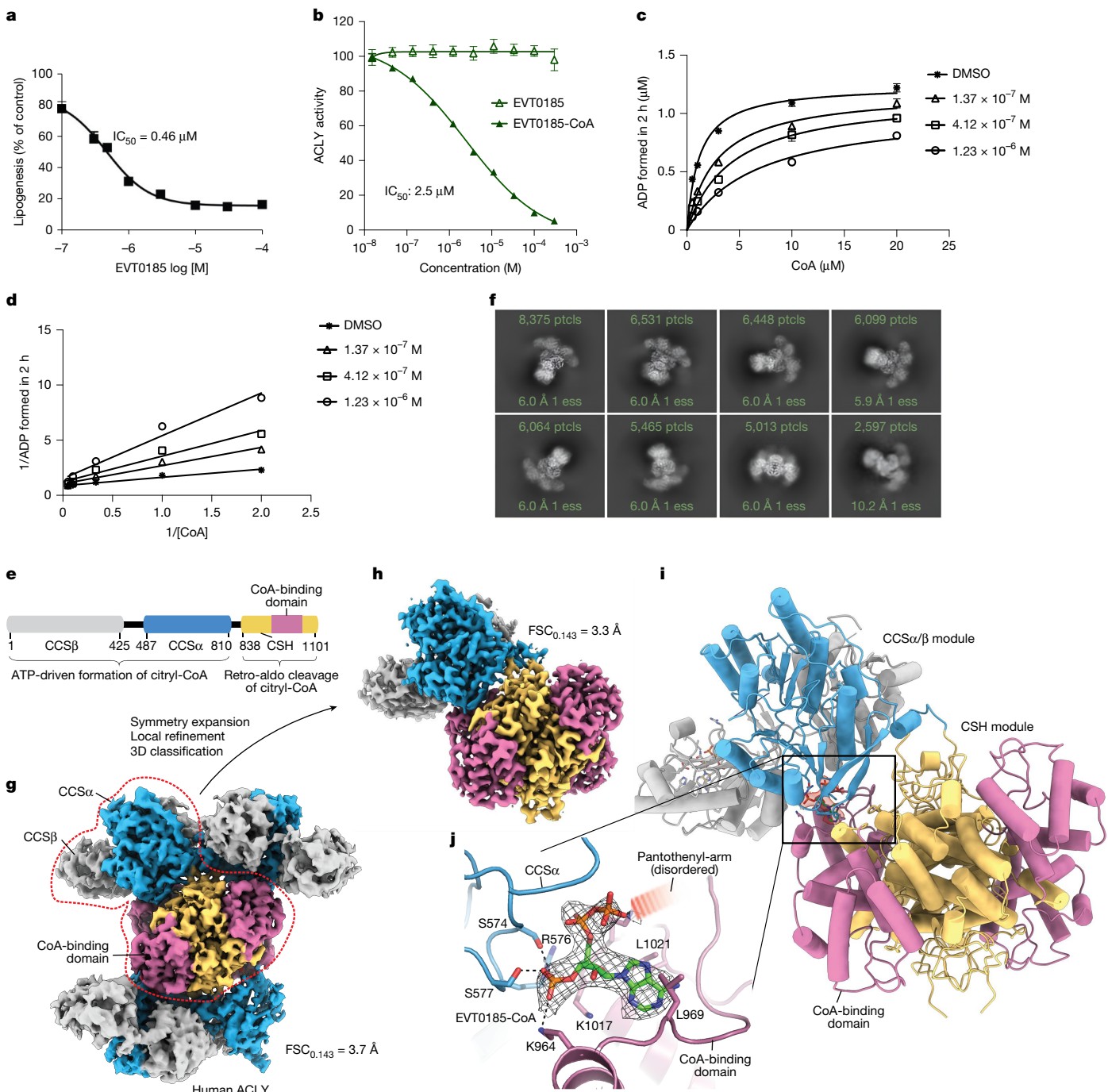

**Fig. 2 | Identification of EVT0185, a novel dicarboxylic acid prodrug, that is converted to a CoA thioester and inhibits ACLY through CoA binding.** **a**, ¹⁴C-acetate incorporation into fatty acids and cholesterol in mouse primary hepatocytes treated with varying doses of EVT0185 (0.1 and 0.5 μM ($n = 4$ samples per group), 0.3, 1 and 3 μM ($n = 10$ samples per group), and 10, 30 and 100 μM ($n = 6$ samples per group)). Data are presented as mean ± s.e.m. **b**, Effect of EVT0185 and EVT0185-CoA on ACLY activity. Data are presented as mean ± s.e.m. Inhibition of ACLY activity by EVT0185 ($n = 3$ samples) and EVT0185-CoA ($n = 3$ samples). **c,d**, Michaelis–Menten (**c**) and Lineweaver–Burk (**d**) plots for ACLY with EVT0185-CoA at three distinct concentrations. Data are presented as mean ± s.e.m. $n = 3$ independent experiments. **e**, Domain architecture of ACLY. **f**, 2D cryo-EM class averages for the ACLY–(*R,S*)-EVT0185-CoA complex.

ess, effective sample size; ptcls, particles. **g**, Cryo-EM reconstruction without symmetry applied. The sharpened map is contoured at $8.74\sigma$ (absolute level of 0.176) and coloured by the different structural domains as in panel **e**. The red dashed line indicates the region used for local refinement. **h,i**, Sharpened cryo-EM map contoured at $9.94\sigma$ (**h**; absolute level of 0.12) and real space refinement atomic model (**i**) following local refinement after symmetry expansion and 3D classification. **j**, Adenosine 3′-phosphate 5′-diphosphate moiety of EVT0185-CoA modelled in the sharpened cryo-EM map at the CoA-binding pocket located at the interface between the CCS and CSH modules of ACLY. The map is carved around the ligand with a carve radius of 2 Å. The interacting residues of ACLY are labelled. The disordered pantothenyl arm is tentatively indicated.

both compact and extended conformations in which the terminal carboxylic acid moiety may anchor itself in the citrate-binding sites of the CSH or CCS module, respectively.

In addition to inhibiting ACLY, fatty acyl-CoAs or bempedoic acid-CoA can activate AMPK, which suppresses fatty acid and sterol synthesis by phosphorylating and inhibiting acetyl-CoA carboxylase (ACC) and

HMG-CoA reductase[30,37] (Extended Data Fig. 5a). To assess potential ACLY-independent effects of EVT0185, we measured lactate incorporation into fatty acids and sterols in hepatocytes from WT and *Acly*-KO mice, comparing results to bempedoic acid at equimolar doses (Extended Data Fig. 5b). We also tested both compounds for their ability to inhibit acetate incorporation, a process independent of ACLY but sensitive to ACC and/or ACSS2 inhibition (Extended Data Fig. 5c). In WT hepatocytes, EVT0185 more potently inhibited fatty acid and cholesterol synthesis from lactate and acetate than bempedoic acid, with differences amplified in *Acly*-KO cells, suggesting that additional targets may be involved (Extended Data Fig. 5b,c). Cell-free assays revealed that, unlike bempedoic acid[30], EVT0185-CoA inhibited rather than activated AMPKβ1-containing complexes (Extended Data Fig. 5d), and also inhibited ACC1, ACC2 and ACSS2 (Extended Data Fig. 5e–g). EVT0185 also suppressed clonogenic survival of human (Hep3B) and mouse (Hepa1-6) HCC cell lines more effectively than bempedoic acid (Extended Data Fig. 5h,i). Given the role of AMPK in pro-survival signalling[38] and the compensatory upregulation of ACSS2 upon ACLY inhibition[8,9,12], these findings highlight key mechanistic differences between EVT0185 and bempedoic acid and supported continued development of EVT0185 for HCC.

## EVT0185 reduces MASH-HCC in mice

Having established mechanisms for EVT0185 inhibition of ACLY, we subsequently examined in vivo efficacy. Compared with vehicle-treated controls, oral administration of EVT0185 reduced the respiratory exchange ratio within 1 h, an effect sustained for up to 5 h, indicating rapid and sustained inhibition of lipogenesis and/or increase in fatty acid oxidation (Extended Data Fig. 6a). Oral EVT0185 dosing inhibited hepatic lipogenesis from $^{14}$C-glucose in a dose-dependent manner, with a trend at 10 mg kg$^{-1}$ (−18%) and significant reductions at 30 mg kg$^{-1}$ (−56%, $P = 0.02$) and 60 mg kg$^{-1}$ (−69%, $P = 0.005$; Extended Data Fig. 6b), indicating oral bioavailability. We then tested EVT0185 in three mouse models, spanning both prevention and treatment of MASH-HCC. In the WD-DEN model (described initially in Extended Data Fig. 1), mice were randomized by AFP and treated daily with vehicle, EVT0185 (30 or 100 mg kg$^{-1}$) or bempedoic acid (100 mg kg$^{-1}$) for 1 month (Extended Data Fig. 6c,d). EVT0185 reduced tumour burden, whereas bempedoic acid had limited efficacy (Fig. 3a,b). Similar to *Acly*-KO mice, EVT0185 also reduced tumour surface area and lipid accumulation (Extended Data Fig. 6e–g). In the WD-CCl$_4$ prevention model, in which MASH and fibrosis are present by 12 weeks but tumours have not yet formed[29,39], EVT0185 nearly eliminated tumour development (Fig. 3c–e). In an 18-month WD model without DEN or CCl$_4$, mice with elevated levels of AFP were randomized to vehicle or EVT0185 (100 mg kg$^{-1}$) for 4 weeks (Extended Data Fig. 6h). EVT0185 again drastically reduced tumour number (Fig. 3f,g). To assess therapeutic potential, we treated WD-CCl$_4$ mice with established tumours starting at 19 weeks. EVT0185 reduced tumour burden and nodule size comparably with sorafenib and lenvatinib (Fig. 3h–j and Extended Data Fig. 6i), and in combination with lenvatinib, led to complete tumour remission in 8% of animals (Fig. 3j). Finally, combining EVT0185 with anti-PDL1 and VEGFR antibodies in the WD-CCl$_4$ model markedly reduced tumour burden and the proportion of animals with more than 25 tumours, overcoming the limited efficacy of immunotherapy alone (Fig. 3k–m and Extended Data Fig. 6j). These data indicate that EVT0185 reduces tumour burden in diverse mouse models of MASH-HCC and improves the efficacy of current standards of care.

## Inhibiting ACLY promotes tumour-infiltrating B cells

To explore mechanisms underlying reduced tumour burden in *Acly*-KO mice, we performed bulk RNA-seq on liver tumours from WT ($n = 9$) and *Acly*-KO ($n = 12$) mice at two timepoints following AAV injection: a late timepoint (8 weeks; Fig. 1a–h and Extended Data Fig. 2) and an early timepoint (4 weeks), before detectable differences in tumour burden (Extended Data Fig. 7a,b). Including the early timepoint allowed us to assess whether transcriptional changes precede tumour reduction. *Acly* expression was significantly reduced at both timepoints (Fig. 4a and Extended Data Fig. 7c,d), although less so at the later stage, possibly due to clonal selection or infiltration by non-hepatocyte cell types. In contrast to previous reports in hepatocytes[26,40], but similar to findings in pancreatic islets[41], the expression of *Acss2* was not significantly upregulated at either timepoint (Fig. 4a).

Time-adjusted analysis revealed 367 upregulated and 333 downregulated genes in *Acly*-KO tumours, with enrichment of antitumour gene sets, including reduced tumour growth size (Extended Data Fig. 7e,f). Metabolically, *Acly*-KO tumours showed higher citrate and lower succinate levels, corresponding with reduced expression of *Acly* and succinate-CoA ligase subunits *Suclg1* and *Sucla2* (Extended Data Fig. 7g,h). Unexpectedly, genes linked to leukocyte proliferation and migration were significantly upregulated in *Acly*-KO tumours (Fig. 4b). Semantic clustering identified nine immune-enriched biological process clusters, including T cell and B cell activation, IFNα or IFNγ responses, and leukocyte adhesion (Fig. 4c and Extended Data Fig. 7i). As tumour burden was unchanged at 4 weeks (Extended Data Fig. 7b), these early immune-related signatures suggested a potential causal role. Supporting this, gene networks inversely correlated with both *Acly* expression and tumour burden were enriched for immune processes (Extended Data Fig. 7j,k). Finally, gene expression deconvolution, using TIMER2, revealed a consistent negative correlation between *Acly* levels and B cell infiltration (Fig. 4d), suggesting a potentially important connection between *Acly*, immunogenicity and tumour burden.

To further investigate tumour–immune interactions, we performed spatial transcriptomics on livers from WD-DEN, WT or *Acly*-KO mice (Fig. 4e,f) and from WD-CCl$_4$-treated, vehicle-treated or EVT0185-treated (100 mg kg$^{-1}$) mice (Fig. 4k,l). Gene Ontology enrichment analysis of spatially resolved tumour cells showed increased fatty acid and lipid metabolism in both *Acly*-KO mice (Fig. 4g,h) and EVT0185-treated mice (Fig. 4m,n). Spatial analysis also revealed a selective increase in the number of B cells, but not in the number of T cells, macrophages or natural killer T cells, in tumours from *Acly*-KO and EVT0185-treated mice (Fig. 4f,l and Extended Data Fig. 8a,b). Using established markers, we found that these B cells were predominantly plasma cells, which mediate antibody production (Fig. 4i,o). Gene Ontology analysis of B cells confirmed enrichment of fatty acid metabolism pathways, supporting their role in plasma cell differentiation (Extended Data Fig. 8c). Similar findings were observed in scRNA-seq datasets from EVT0185-treated WD-DEN and WD-CCl$_4$ tumours (Extended Data Fig. 8d,e). The levels of *Cxcl13*, a key B cell chemoattractant known to be reduced in MASH-HCC[17], were elevated in tumours from both *Acly*-KO and EVT0185-treated mice (Fig. 4j,p), a finding replicated in publicly available RNA-seq data from WT and *Acly*-KO DEN-induced tumours cultured in vitro[10] (GSE223966; Extended Data Fig. 8f). These data indicate that genetic or pharmacological inhibition of ACLY in MASH-HCC leads to increases in tumour CXCL13 levels and plasma B cell numbers.

## Antitumour effect of ACLY requires B cells

To further interrogate the inferred cell populations obtained from the RNA-seq and spatial transcriptomic analysis, we examined similar-sized, neoplastic lesions from the livers of WT and *Acly*-KO mice using multiplexed ion-beam imaging by time of flight (MIBI-TOF) imaging, which enables single-cell protein profiling[42]. A 21-antibody Lanthanide panel (Supplementary Table 3) and FlowSOM clustering identified nine cell types in amounts greater than 0.5% of the total cell population: B cells, CD4$^+$ T cells, CD11b$^+$ macrophages, CD11c$^+$ macrophage/dendritic cells, F4/80$^+$ macrophages, myofibroblasts, endothelial cells,

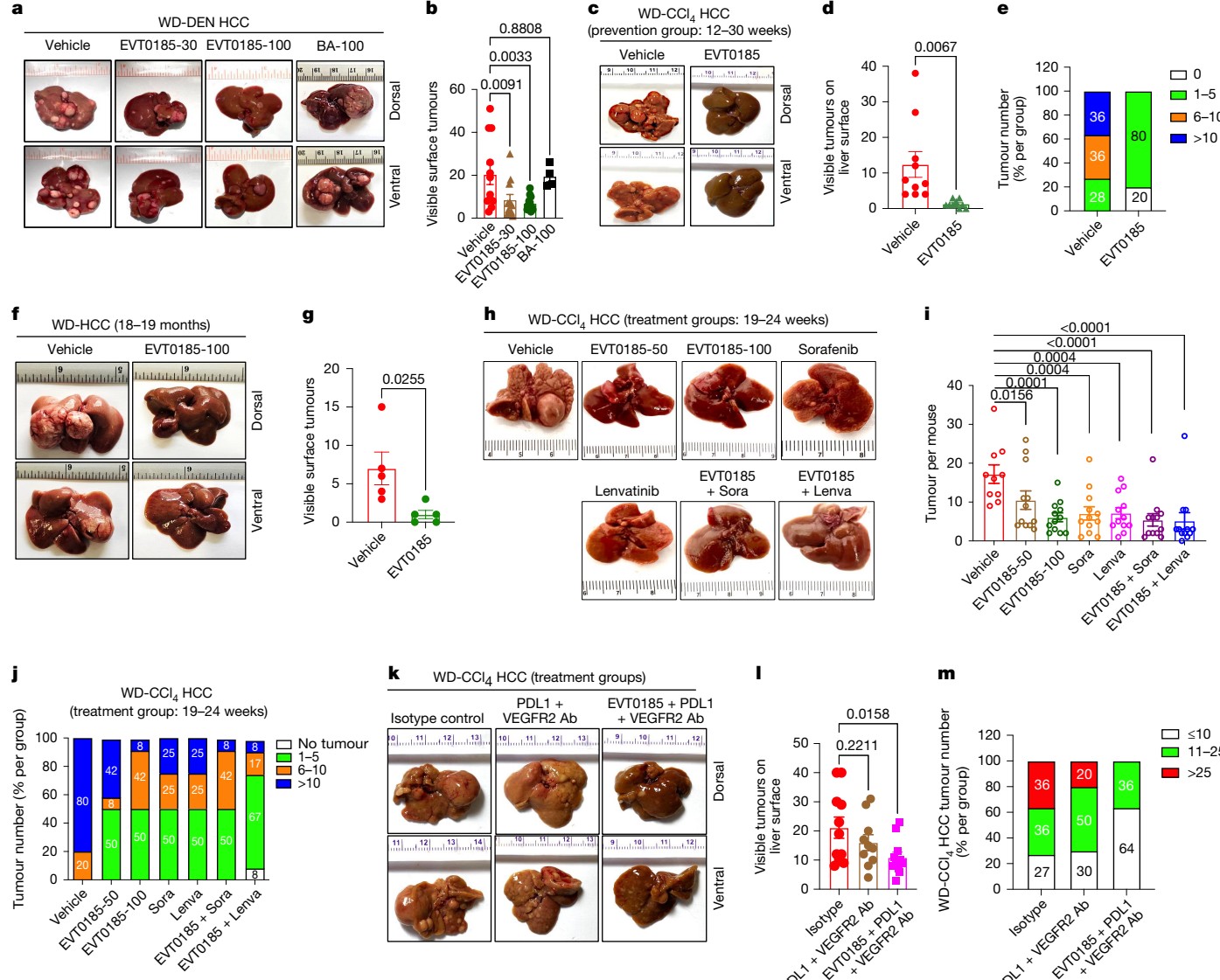

**Fig. 3 | Oral delivery of the ACLY inhibitor EVT0185 reduces tumour burden in distinct mouse models of MASH-HCC. a**, Representative images of livers isolated from mice WD-DEN for 8 months, then treated with vehicle, EVT0185 (30 or 100 mg kg⁻¹) or bempedoic acid (BA-100 mg kg⁻¹) for 1 month.
**b**, Quantification of visible surface tumours from livers of mice treated with vehicle (*n* = 12), EVT0185 (30 or 100 mg kg⁻¹; *n* = 12) or bempedoic acid (BA-100 mg kg⁻¹; *n* = 4). Data are mean ± s.e.m. *P* values were determined by one-way analysis of variance (ANOVA) with Fisher's least significant difference (LSD).
**c**, Representative images of livers isolated from WD-CCl₄ mice (12 weeks) treated with vehicle or EVT0185 (100 mg kg⁻¹) for 18 weeks. **d**, Visible surface tumours on livers. Data are mean ± s.e.m. *n* = 10 mice per group for vehicle or EVT0185 (100 mg kg⁻¹). *P* value was determined by an unpaired, two-tailed Student's *t*-test.
**e**, Percent distribution of tumour numbers per group. **f**, Representative images of livers isolated from mice maintained on WD for 18 months and treated with vehicle or EVT0185 (100 mg kg⁻¹) for 4 weeks. **g**, Visible number of surface tumours. Data are mean ± s.e.m. *n* = 5 mice per group. *P* value was determined

by unpaired, two-tailed Student's *t*-test. **h**, Representative images of livers isolated from WD-CCl₄ mice treated with vehicle, EVT0185 (50 or 100 mg kg⁻¹), sorafenib (Sora; 15 mg kg⁻¹), lenvatinib (Lenva; 7 mg kg⁻¹), EVT0185 + sorafenib or EVT0185 + lenvatinib for 6 weeks. **i**, Tumour counts in the liver. Data are mean ± s.e.m. *n* = 12 mice in the treatment groups and *n* = 10 mice in the vehicle group. *P* values were determined by one-way ANOVA with Fisher's LSD: *P* = 4.20 × 10⁻⁵ (EVT0185 + sorafenib versus vehicle) and *P* = 3 × 10⁻⁵ (EVT0185 + lenvatinib versus vehicle). **j**, Percent distribution of tumour numbers.
**k**, Representative images of livers isolated from WD-CCl₄ mice treated with isotype control or PDL1 and VEGFR2 antibody (Ab; 200 µg) with or without EVT0185 (100 mg kg⁻¹). **l**, Visible surface tumours on the livers. Data are mean ± s.e.m. *n* = 11 isotype control, *n* = 10 PDL1 + VEGFR2 antibody and *n* = 11 EVT0185 + PDL1 + VEGFR2 antibody injected mice. *P* values were determined by one-way ANOVA with Fisher's LSD. **m**, Percent distribution of tumour numbers per group.

endothelial–mesothelial cells and tumour cells (Fig. 5a). Consistent with the transcriptomic datasets, *Acly*-KO mice had increases in the number of B cells at the tumour periphery, with non-significant increases in T cells or other immune cell populations (Fig. 5b–d); findings that were confirmed using immunohistochemical staining of CD19⁺ B cells (Fig. 5e). These changes were accompanied by reductions in the levels of lipid droplet-associated protein (PLIN2) and increases in CXCL13 protein expression (Extended Data Fig. 9a–c). Tertiary lymphoid structures

(TLSs) are B cell aggregations that have been associated with improved HCC outcomes[43]. Detailed analysis of the tumour B cell infiltrates by a pathologist blinded to the treatments revealed that although diffuse and peripheral B cell infiltrations were observed in tumours from both genotypes, B cell aggregations resembling TLSs were predominantly observed in *Acly*-KO mice (80%; Fig. 5f and Extended Data Fig. 9d,e). MIBI spatial mapping showed B cells near antigen-presenting cells (CD11c⁺, CD4⁺) in *Acly*-KO tumours (Fig. 5g–i and Extended Data Fig. 9f). This

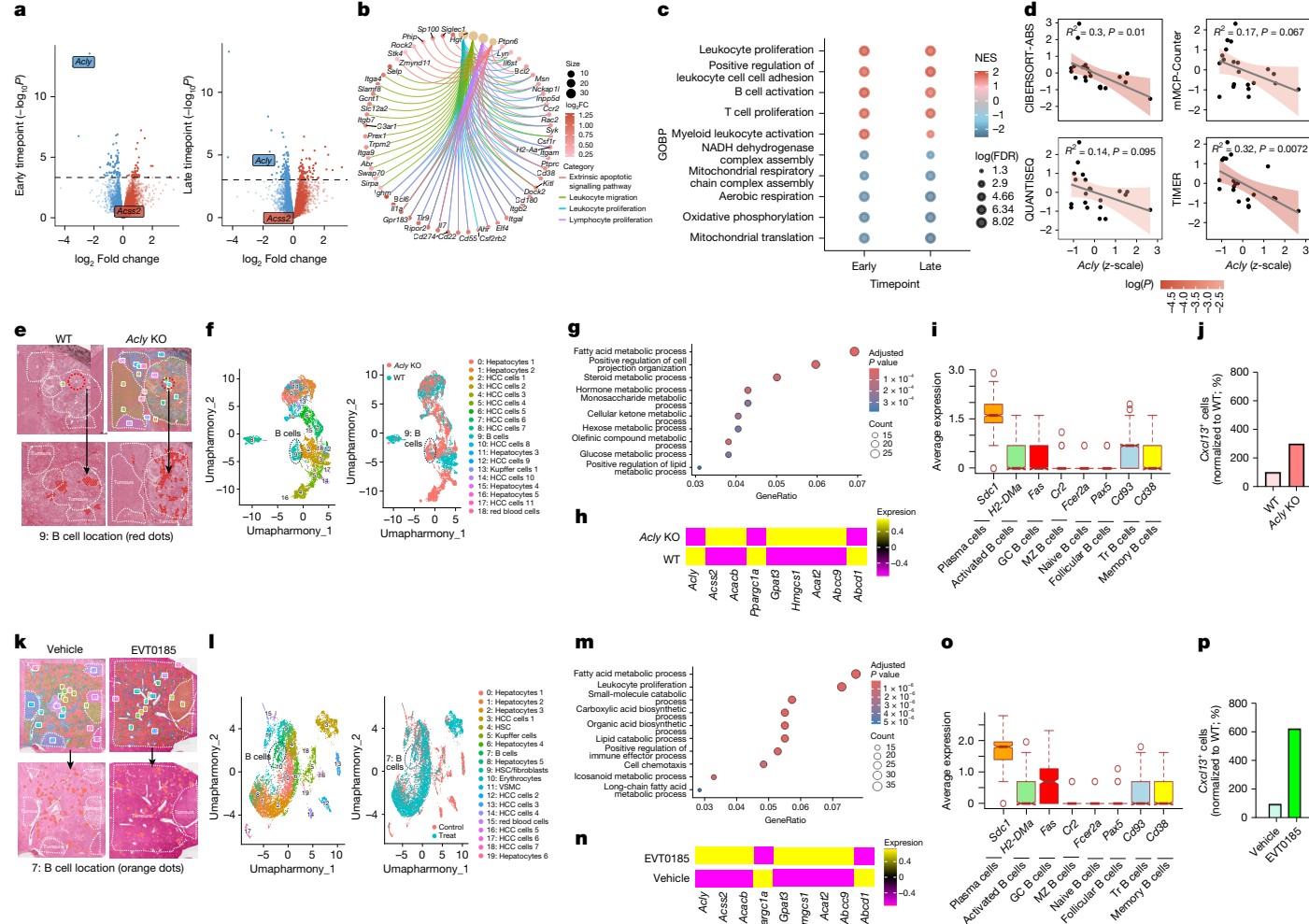

**Fig. 4 | Genetic inhibition of *Acly* or treatment with EVT0185 in MASH-driven HCC promotes tumour-infiltrating B cells. a**, Volcano plot of bulk RNA-seq of tumours showing upregulated and downregulated genes at early or late timepoints from *Acly*-KO (*n* = 12) versus WT (*n* = 9) mice. Significance was determined by Wald test with a false discovery-adjusted threshold of 5% as implemented in DESeq2. Horizontal dashed lines demarcate the *P* value threshold at a 5% false discovery rate (FDR). **b**, Gene Ontology analysis of selected biological processes involving significantly upregulated genes in tumours from *Acly*-KO (*n* = 12) versus WT (*n* = 9) mice. FC, fold change. **c**, Top 10 Gene Ontology biological processes (GOBP) from clusters identified among significantly upregulated gene sets in tumours from *Acly*-KO (*n* = 12) versus WT (*n* = 9) mice. NES, normalized enrichment score. **d**, Correlation between B cell populations and *Acly* expression in tumours from WT (*n* = 9) and *Acly*-KO (*n* = 12) mice. Confidence bands denote the upper and lower bounds of the 95%

confidence interval. Significance of association was determined by a two-sided Student's *t*-test of regression coefficients and at a false discovery-adjusted threshold of 5%. **e–p**, Spatial transcriptomic analysis of livers from WT and *Acly*-KO mice and vehicle or EVT0185-treated mice. Cluster analysis representing the number of cell types in the liver and tumour (**e,k**). Umapharmony integration analysis showing increased B cells in *Acly*-KO (**f**) and in EVT0185-treated (**l**) mice. The top upregulated pathways in HCC cells from *Acly*-KO (**g**) and EVT0185-treated (**m**) mice. Statistical analysis was performed using Fisher's exact test. Expression level of metabolic genes (**h,n**). Expression of markers of subtypes of B cells (**i,o**). The box-and-whisker plots are defined by the median with the first quartile (Q1), third quartile (Q3), minimum (Q1 − 1.5 × interquartile range (IQR)) and maximum (Q3 + 1.5 × IQR). *Cxcl13* expression levels in HCC cells (**j,p**). GC, germinal centre; HSC, hepatic stellate cell; MZ, marginal zone; Tr, regulatory; VSMC, vascular smooth muscle cell.

pattern was also seen in tumours from WD-CCl₄ EVT0185-treated mice without significantly altering T cell counts (Extended Data Fig. 9g–n). Supporting their functional role, scRNA-seq revealed upregulation of complement activation and cytotoxicity pathways in B cells from EVT0185-treated mice (Extended Data Fig. 9o). Reduced tumour burden in *Acly*-KO and EVT0185-treated mice was accompanied by lower Ki67 and higher cleaved caspase 3 levels at the tumour edge (Extended Data Fig. 10a–f). As AAV8-TTR-Cre does not target immune cells (Fig. 1e) and EVT0185 is inactive in them (Extended Data Fig. 3), these findings across multiple platforms (RNA-seq, spatial transcriptomics, scRNA-seq, MIBI and immunohistochemistry) strongly implicate B cells as key mediators of the antitumour effects of ACLY inhibition.

To determine whether the immunogenic response was important for reducing tumour burden in response to genetic inhibition of *Acly*,

we generated an inducible system for knocking down ACLY expression in the human Hep3B cell line (Extended Data Fig. 10g) for orthotopic implantation in immunodeficient NRG mice. As anticipated, *Acly* knockdown reduced glucose incorporation into triglycerides and led to compensatory upregulation of acetate incorporation into triglycerides (Extended Data Fig. 10h,i). In addition, inducible knockdown increased fatty acid oxidation (Extended Data Fig. 10j). However, despite these anticipated changes in metabolic activity, *Acly* knockdown did not impair orthotopic tumour growth in immunodeficient mice (Extended Data Fig. 10k,l). This suggests that alterations in lipid metabolism without a functional adaptive immune system was not sufficient for reducing tumour burden.

To more specifically address whether an increase in tumour-infiltrating B cells is critical for the reduced tumour burden in *Acly*-KO

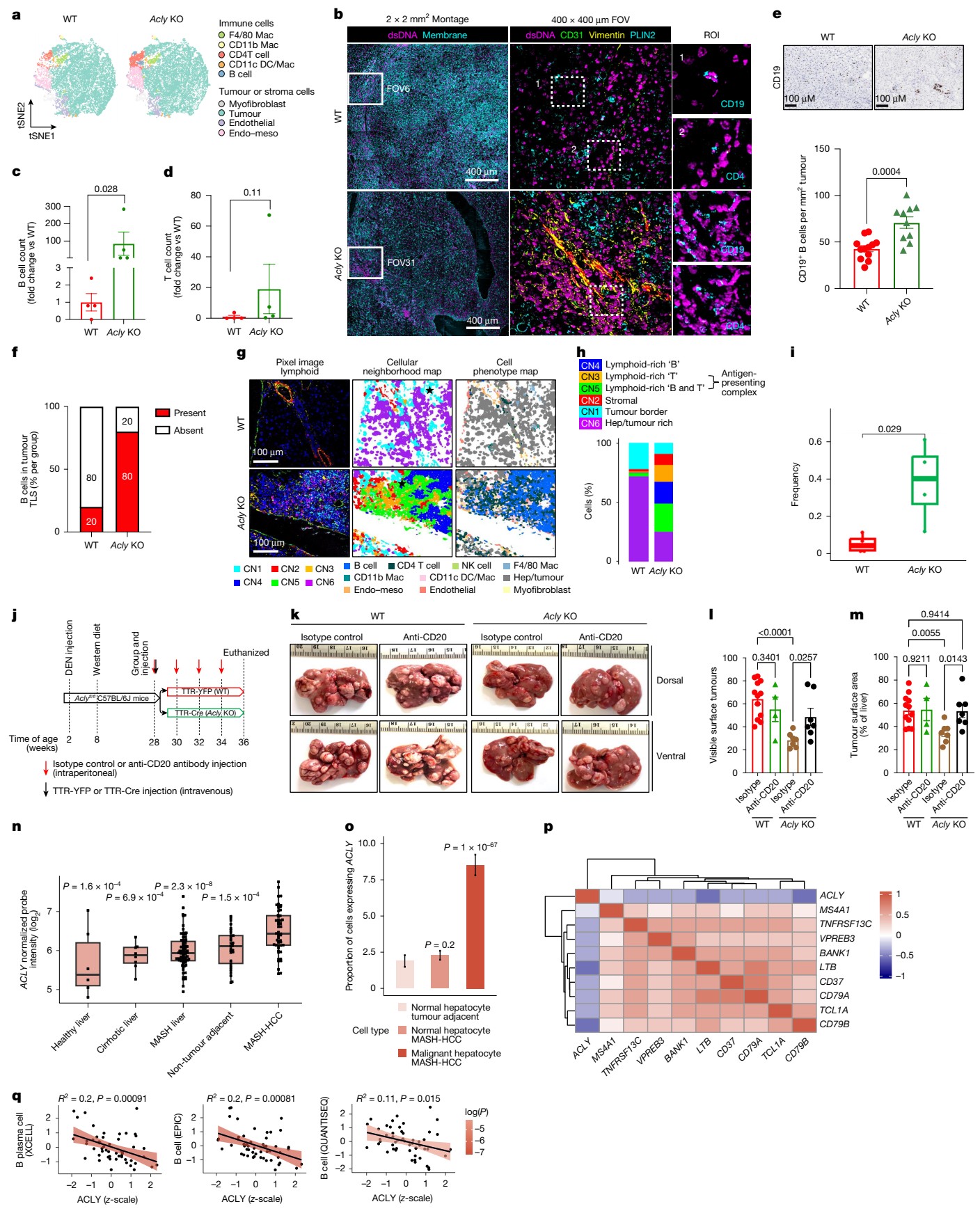

**Fig. 5 | See next page for caption.**

**Fig. 5 | Infiltrating B cells are important for reducing tumour burden in MASH-HCC. a**, *t*-distributed stochastic neighbour embedding (tSNE) plots showing cell phenotypes in WT and *Acly*-KO mice. DC, dentritic cell; endo−meso, endothelial−mesothelial; Mac, macrophage. **b**, Representative tumour images: $2 \times 2$ mm$^2$ montage (25 fields of view (FOVs)), single $400 \times 400$ µm FOV, and $100 \times 100$ µm region of interest (ROI). dsDNA, double-stranded DNA. **c,d**, B cell (**c**) and T cell (**d**) counts at the tumour−liver interface. Data are mean ± s.e.m. $n = 4$ mice per group. *P* values were determined by Wilcoxon unpaired, two-sided test. **e**, Representative image of CD19$^+$ staining (top) and quantification of CD19$^+$ cells per mm$^2$ tumour area in WT ($n = 12$) and *Acly*-KO ($n = 10$) lesions (bottom). Data are mean ± s.e.m. *P* value was determined by unpaired, two-tailed Student's *t*-test. **f**, Percentage of mice with B cells present or absent in tumour TLSs. **g**, Pixel images of lymphoid marker levels with cell neighbourhood and phenotype maps overlaid on cell-boundary masks. The tumour interior is marked by a black star. CN1, cellular neighbourhood 1; Hep, hepatocyte; NK cell, natural killer cell. **h**, Distribution of cells across cellular neighbourhoods. **i**, CN3 and CN5 (antigen-presenting enriched) represented as 'antigen-presenting complex'. The box-and-whisker plots show the median as hinge and a box representing the middle 50% of the data, bounded by the lower and upper quartiles. The whiskers extend to data within $1.5 \times$ IQR. *P* value was determined by Wilcoxon non-parametric unpaired, two-sided test. $n = 4$ WT and $n = 4$ *Acly*-KO tumour border regions. **j**, Schematic of B cell depletion protocol in the *Acly*$^{fl/fl}$ WD-DEN model. **k**, Representative liver images. **l,m**, Tumour count (**l**) and surface area (**m**). Data are mean ± s.e.m. $n = 12$ WT isotype-injected, $n = 4$ WT anti-CD20-injected, $n = 7$ *Acly*-KO isotype-injected and $n = 7$ *Acly*-KO anti-CD20-injected mice. *P* values were determined by one-way ANOVA with Fisher's LSD: $P = 7.51 \times 10^{-5}$ (*Acly*-KO isotype versus WT isotype). **n**, *ACLY* upregulation in human MASH-HCC ($n = 53$) relative to non-tumour adjacent tissue ($n = 29$), MASH liver ($n = 74$), cirrhotic liver ($n = 8$) and healthy liver ($n = 6$). The box-and-whisker plot lines represent the Q1, median and Q3. The whiskers connect the minimum and maximum values. *P* values were determined by two-tailed Student's *t*-test as implemented in limma. **o**, Hepatocyte-specific *ACLY* expression in MASH-HCC tumour and adjacent tissues (scRNA-seq). *P* values were determined by $\chi^2$ test of independence. Error bars represent 95% confidence interval. **p,q**, Correlation of *ACLY* expression with B cell-specific markers (**p**), and B cell abundance based on cell-type deconvolution methods (**q**). Confidence bands denote 95% confidence interval bounds; two-tailed Student's *t*-test of regression coefficients (FDR < 0.05).

mice, we established MASH-HCC in *Acly*$^{fl/fl}$ mice as described in Fig. 1. After 28 weeks, AFP levels were assessed, and mice were randomized to four distinct groups based on this parameter before injection with either AAV8-YFP (WT) or AAV8-TTR-Cre (KO), and treatment with an anti-CD20 antibody, which depletes B cells[44] (Fig. 5j). As anticipated, the anti-CD20 antibody eliminated B220$^+$CD19$^+$ B cells compared with isotype control-injected mice (Extended Data Fig. 11a,b). In WT animals, the depletion of B cells had no effect on the number of visible surface tumours (Fig. 5k−m). Consistent with our previous findings (Fig. 1), *Acly*-KO mice had large reductions in visible surface tumours when treated with isotype control; however, these effects were eliminated following the depletion of B cells (Fig. 5k−m). These data indicate that the reduction in tumour burden elicited by the inhibition of ACLY in mice with MASH-HCC requires B cells.

## ACLY upregulation dampens immunity

To assess the relevance of ACLY inhibition in human MASH-HCC, we first analysed the microarray dataset[15] (GSE164760), comparing tumour, non-tumour adjacent, MASH, cirrhotic and healthy liver tissue. ACLY was one of 73 genes consistently upregulated in MASH-HCC across all pairwise comparisons (Fig. 5n and Extended Data Fig. 12a,b), suggesting that its overexpression in malignant tissue is not confounded by intermediate disease states. This pattern was not observed for other lipogenic enzymes such as *ACACA*, *ACACB*, *ACSS2* or *FASN* (Extended Data Fig. 12c). Supporting this, snRNA-seq datasets confirmed that increased ACLY expression in malignant hepatocytes versus normal hepatocytes from patients with MASH-HCC (Fig. 5o and Extended Data Fig. 12d).

We next explored the relationship between ACLY and tumour immunogenicity. Similar to our mouse models, ACLY expression inversely correlated with B cell marker expression in human MASH-HCC (Fig. 5p), and cell deconvolution analyses showed higher *ACLY* levels associated with reduced B cell infiltration (Fig. 5q). To understand the pathways underlying *ACLY* overexpression, we performed differential gene expression modelling followed by weighted gene correlation network analysis (WGCNA) module analysis (Extended Data Fig. 12e,f). *CXCL13* expression was significantly reduced in tumours with high ACLY expression and increased in those with low expression, mirroring mouse data (Extended Data Fig. 12g). Gene co-expression modules linked to high levels of ACLY were enriched for metabolic pathways—including macromolecule biosynthesis, aerobic respiration and acetylation— whereas they were negatively associated with immune-related features (Extended Data Fig. 12h,i). These findings reinforce a conserved inverse relationship between ACLY expression and tumour immunogenicity across human and mouse MASH-HCC.

This study reveals an unexpected immunoregulatory role for ACLY in MASH-HCC. Although ACLY is well known for its metabolic function in suppressing fatty acid and cholesterol synthesis, we have demonstrated that hepatocyte-specific *ACLY* deletion not only suppresses intratumoural steatosis and proliferation but also enhances antitumour immunity. Genetic inhibition of *ACLY* increased tumour infiltration by B cells and upregulated the B cell chemoattractant CXCL13, promoting the formation of TLSs, immune niches associated with favourable prognosis in HCC and other solid tumours[17,43,45–47]. Crucially, B cell depletion abolished the antitumour effect of ACLY inhibition, confirming their essential role. As these tumour-suppressive effects in the *Acly* KO occurred independently of changes in liver histology, and the TTR-Cre promoter used to induce deletion does not affect immune cells[26], these studies provide surprising findings about how inhibiting ACLY not only suppresses tumour proliferation but also enhances immunogenicity. Our findings indicating a critical role for B cells are consistent with emerging studies involving mouse models and patients with HCC that have shown favourable outcomes linked to increased tumoural B cell infiltration and TLSs[48–54]. These effects were associated with increases in the B cell chemokine CXCL13, which has been shown to be important for predicting response rates to immunotherapy in MASH-HCC. The precise mechanisms by which ACLY inhibition increases CXCL13 and TLS formation remain undefined; however, future studies investigating whether reductions in the levels of tumour metabolites such as fatty acids and succinate or increase in the levels of citrate directly impact tumoural B cell recruitment and formation of TLSs or, alternatively, whether there are reductions in the level of *CXCL13* promoter acetylation will be important.

We developed and characterized EVT0185, a novel, orally available small-molecule ACLY inhibitor, which phenocopied the immune and antitumour effects of genetic *ACLY* deletion. EVT0185 is activated in hepatocytes via SLC27A2-dependent conversion to its CoA-thioester form, allowing liver-specific action while sparing non-hepatic tissues. In multiple MASH-HCC mouse models, EVT0185 reduced tumour burden and synergized with current standards of care, including lenvatinib or anti-PDL1 and VEGFR antibodies. Structural cryo-EM analysis demonstrated that EVT0185-CoA binds directly to the ACLY CoA-binding pocket, supporting a clear mechanistic basis for its inhibitory action. Like the *Acly*-KO model, EVT0185 treatment was associated with increased plasma cell differentiation and complement-mediated cytotoxic pathways. Although our data strongly support ACLY as the principal target of EVT0185 in driving tumour immunogenicity, it remains possible that other targets may be important as we found that this compound also reduced acetate incorporation into fatty acids and sterols, an effect that was independent of ACLY. However, analysis

of MASH-HCC tumours in humans mirrored our findings in *Acly*-KO mice, with ACLY expression but not ACSS2, ACC1/ACC2 or FAS being inversely correlated with B cell infiltration. Together, these data in both mice and humans strongly suggest that inhibition of ACLY is the primary driver of the enhanced tumour immunogenicity, mediated by EVT0185. EVT0185 was inactive in cultured immune cell populations, consistent with their low expression of *Slc27a2*, as activated T cells and macrophages require ACLY to elicit effector function. Together, these data position EVT0185 as a promising candidate for targeting ACLY in MASH-HCC.

In conclusion, these studies have shown that genetic inhibition of ACLY in MASH-driven mouse models of HCC promotes the infiltration of tumour-infiltrating B cells and that this is critical for reducing tumour burden. Similar effects are also observed with EVT0185, a small-molecule competitive inhibitor of ACLY. These findings expand the role of tumour ACLY from a regulator of metabolism and proliferation to a mediator of antitumour immunity and suggest that targeting of tumour metabolism may be a largely untapped mechanism to improve immunotherapy response rates. As reductions in *ACLY* in clinical datasets are also associated with lower tumour burden and increased B cell infiltration, further studies investigating whether pharmacological inhibition of ACLY also increases immunosurveillance and reduces tumour burden in clinical populations will be important.

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

[1]Centre for Metabolism, Obesity and Diabetes Research, Department of Medicine, McMaster University, Hamilton, Ontario, Canada. [2]Espervita Therapeutics, Ann Arbor, MI, USA. [3]McMaster Immunology Research Centre, Department of Medicine, McMaster University, Hamilton, Ontario, Canada. [4]Department of Oncology, McMaster University, Hamilton, Ontario, Canada. [5]Department of Chemistry, University of Florida, Gainesville, FL, USA. [6]Division of Liver Diseases, Department of Medicine, Icahn School of Medicine at Mount Sinai, New York, NY, USA. [7]Division of Clinical Immunology and Allergy, McMaster Immunology Research Centre, Schroeder Allergy and Immunology Research Institute, McMaster University, Hamilton, Ontario, Canada. [8]Division of Respirology, McMaster Immunology Research Centre, Schroeder Allergy and Immunology Research Institute, McMaster University, Hamilton, Ontario, Canada. [9]Division of Digestive and Liver Diseases, Department of Internal Medicine, University of Texas Southwestern Medical Center, Dallas, TX, USA. [10]Department of Biochemistry and Microbiology, Ghent University, Ghent, Belgium. [11]Unit for Structural Biology, VIB Center for Inflammation Research, Ghent, Belgium. [12]Department of Molecular and Cellular Endocrinology, Diabetes and Metabolism Research Institute, City of Hope Medical Center, Duarte, CA, USA. [13]Firestone Institute for Respiratory Health, The Research Institute at St. Joe's, St. Joseph's Healthcare, Hamilton, Ontario, Canada. [14]These authors contributed equally: Jaya Gautam, Jianhan Wu, James S. V. Lally. ✉e-mail: gsteinberg@mcmaster.ca

## Methods

### Syntheses of EVT0185 and its mono-CoA derivative

EVT0185 and EVT0185-CoA were synthesized at Symeres (www.symeres.com); experimental methods and compound characterization are presented in the Supplementary Information.

**EVT0185**

**EVT0185-CoA**

### Mouse models

Animal experiments were carried out using the guidelines approved by the Animal Research Ethics Board at McMaster University (Steinberg Laboratory Animal Utilization Protocol #16-12-42, 21-01-04) or the Institutional Animal Care and Use Committee (IACUC) at Icahn School of Medicine at Mount Sinai (IACUC approval# PROTO202100080). The tumour development was monitored based on the established progression timelines for mouse HCC models. The welfare of the animals was evaluated through observation of visible masses, impaired mobility and marked weight loss. However, no specific tumour volume end points were defined for intrahepatic tumours. All experimental procedures did not exceed the limits set by the Animal Utilization Protocol or IACUC.

**WD-DEN mouse model.** C57BL/6J male mice were housed within the McMaster University Central Animal Facility. At 2 weeks of age, mice were injected with DEN at 25 mg kg$^{-1}$ body weight. After weaning, mice were housed with littermates and fed a normal chow diet (8640 22/5, Teklad). At 10 weeks of age, mice were maintained on chow diet (control) or switched to a high-fat and high-fructose diet (WD), which consisted of 40% kcal fat (mostly palm oil), 20% kcal fructose and 0.02% wt cholesterol (D19101102, Research Diets) and housed at thermoneutral conditions (26–29 °C)[26]. Throughout the experiments, mice were housed in ventilated cage racks with ad libitum access to food and water. An automatic timing device was used to maintain an alternating 12-h cycle of light and dark. After 29 weeks, mice were euthanized, and blood and tissues were collected.

(1) For the WD-DEN *Acly*-KO mouse model (Figs. 1 and 5), at 2 weeks of age, C57BL-6/*Acly*$^{fl/fl}$ mice (obtained from JAX) were injected with DEN at 25 mg kg$^{-1}$ body weight. After weaning, mice were housed with littermates and fed a normal chow diet (8640 22/5, Teklad). At 10 weeks of age, mice were switched to a high-fat and high-fructose diet (WD): 40% kcal fat (mostly palm oil), 20% kcal fructose and 0.02% cholesterol (D19101102, Research Diets) and housed at thermoneutral conditions. When the mice reached 7.5 months, they were bled from the tail vein, and plasma AFP levels were determined. Animals were grouped based on their AFP levels, so there were no differences between groups, and were then injected with AAV intravenously (AAV8-YFP or AAV8-TTR-Cre) as previously described[26]. After 4 (early timepoint) or 8 (late timepoint) weeks following AAV injection, mice were anaesthetized using ketamine–xylazine, and terminal blood was collected by cardiac puncture. For the B cell depletion experiments, AFP levels were assessed as described above after 7.5 months, and mice were randomized before being injected with AAV intravenously (AAV8-YFP or AAV8-TTR-Cre) and intraperitoneally injected with 250 μg of isotype control (400566, BioLegend) or anti-CD20 antibody (152104, BioLegend) with tissues collected 8 weeks later.

(2) For the WD-DEN EVT0185 and bempedoic acid study (Fig. 3), male mice housed within the McMaster University Central Animal Facility were used for these experiments. C57BL/6J mice were injected with DEN at 2 weeks of age and followed the above-mentioned protocol. Animals were grouped based on their AFP levels and orally gavaged with a single daily dose of vehicle (1.5% CMC and 0.2% Tween-20), EVT0185-30 or 100 mg kg$^{-1}$ or bempedoic acid-100 mg kg$^{-1}$. After 4 weeks of treatment, mice were anaesthetized using ketamine–xylazine, and terminal blood and tissues were collected.

**WD-CCl$_4$ HCC model.** Male C57BL/6J mice obtained from Jackson Laboratory were used for these experiments. Mice at 6–8 weeks of age were fed ad libitum with WD (Envigo; TD.120528, Teklad Custom Research Diet) and maintained on sugar water (fructose (23.1 g l$^{-1}$) + glucose (18.9 g l$^{-1}$)) for the duration of the study. Animals received weekly intraperitoneal injections of CCl$_4$ (0.2 μl, 100% CCl$_4$ per gram of body weight) throughout the study as described below.

The prevention study was conducted after 12 weeks of CCl$_4$ injections + WD at McMaster University with mice orally gavaged with vehicle or EVT0185 (100 mg kg$^{-1}$) for 18 weeks.

The combination treatment study was conducted after 19 weeks of CCl$_4$ injection + WD at Mont Sinai with mice orally gavaged with EVT0185 (50 or 100 mg kg$^{-1}$), sorafenib p-tosylate (sorafenib; at 15 mg kg$^{-1}$; S-8502, LC Laboratories), lenvatinib mesylate (lenvatinib; 7 mg kg$^{-1}$; 29832, LC Laboratories) or sorafenib + EVT0185 (100 mg kg$^{-1}$) or lenvatinib + EVT0185 (100 mg kg$^{-1}$).

The combination study with immunotherapy was conducted after 24 weeks of CCl$_4$ injections + WD at McMaster University with mice being injected with InVivoPlus anti-mouse PDL1 (B7-81; BP0101), and InVivoPlus anti-mouse VEGFR2 (DC101; BP0060) alone or in combination with EVT0185 (100 mg kg$^{-1}$). Isotype controls received InVivoPlus rat IgG1 isotype control, horseradish peroxidase (HRP; BP0088) and InVivoPlus rat IgG2b isotype control, anti-keyhole limpet haemocyanin (LTF-2; BP0090) as a single intraperitoneal injection containing 200 μg of each antibody in a total volume of 100 μl. Antibodies were prepared in InVivoPure pH 7.0 Dilution Buffer (IP0070). After 6 weeks of treatment, animals were anaesthetized using ketamine–xylazine, and terminal blood was collected by cardiac puncture.

**WD-HCC mouse model.** Male C57BL/6J mice obtained from Jackson Laboratory were used for these experiments. Mice at 6 weeks of age were fed ad libitum with WD, which consisted of 40% kcal fat (mostly palm oil), 20% kcal fructose and 0.02% wt cholesterol (D19101102, Research Diets). Throughout the experiments, mice were housed at thermoneutral conditions (26–29 °C). An automatic timing device was used to maintain an alternating 12-h cycle of light and dark. After 18 months, AFP-positive mice were divided into two groups and received a single daily dose via gavage of either vehicle (1.5% CMC and 0.2% Tween-20) or EVT0185 (100 mg kg$^{-1}$). After 4 weeks of treatment, mice were anaesthetized using ketamine–xylazine, and terminal blood and tissues were collected. Blood samples were centrifuged at 10,000 rpm for 10 min at 4 °C and serum was collected and stored at −80 °C freezer. The liver was weighed, a photo was taken and the number of visible lesions on the surface of the liver was counted and recorded. A portion of tumour-free liver (approximately 200 mg, caudate lobe) and tumours were removed, flash-frozen in liquid nitrogen and stored for molecular analysis. The remainder of the liver was placed in 10% formalin, fixed for 48 h and stored in 70% ethanol until it was embedded in paraffin, sectioned and mounted on slides for histological analysis.

**Orthotopic liver cancer model.** 1 day before surgery, male NOD-$Rag1^{null}IL2rg^{null}$ (NRG) received intraperitoneal injections of 100 mg kg$^{-1}$ cyclophosphamide and subcutaneous injections of 5 mg kg$^{-1}$ carprofen. One hour before surgery, Hep3B-Luc cells were resuspended on ice in a 1:1 dilution of phenol red-free Matrigel (356237, Corning) and cold PBS (10010023, Gibco), aliquoted in 50 μl volumes at a cell density of $2 \times 10^7$ cells per millilitre. During surgery, mice were anaesthetized with 2% isoflurane (1001936040, Baxter), hair removed with 3-in-1 hair removal lotion (061700222611, Nair) and placed in a supine position on a heating pad, with nose fitted in an anaesthesia nose cone. The abdomen was disinfected and an approximately 2-cm horizontal incision was made below the left coastal margin from the midline of the abdomen. Separating the skin from the peritoneum, a second and smaller horizontal incision was made across the peritoneum to expose the liver. Next, the left lobe of the liver was withdrawn, stabilized and tumour cells were injected into the liver parenchyma while applying pressure using a cotton tipped applicator (4305, Dynarex) to enable haemostasis. Following injection, the liver was returned into the peritoneal cavity and the peritoneum was closed with multiple single interrupted sutures using 4-0 Vicryl sutures (J743D, Ethicon). The skin was closed using 9-mm wound clips (RS-9262, Roboz). Finally, mice received post-surgical subcutaneous injections of 5 mg kg$^{-1}$ carprofen and were placed in cages on a heating pad containing diet recovery gel (72-06-5022, Clear H$_2$O) and monitored until they were ambulatory. Mice were monitored for recovery for 7 days after surgery. Wound clips were removed on day 6, and mice were imaged for tumour progression on day 7 and weekly thereafter. Induction of in vivo short hairpin RNA (shRNA) expression was achieved via supplementing drinking water with 2 mg ml$^{-1}$ doxycycline (Dox; D9891, Sigma-Aldrich) and 5% sucrose starting on day 7 post-surgery and every 2–3 days thereafter. In parallel, control mice were fed with drinking water supplemented with 5% sucrose without Dox. Tumour progression was followed until the end point, which is defined as the loss of more than 20% of the body weight. Mice were culled collectively when the first mouse in the experimental cohort (Hep3B-shACLY-Luc) reached the end point. Liver and tumour tissues were weighed and collected for downstream analysis. Tumour progression was monitored via bioluminescence imaging using the IVIS Spectrum In Vivo Imaging System (124262, Perkin Elmer). Bioluminescence was achieved through intraperitoneal injection of D-luciferin dissolved in saline (S8776, Sigma-Aldrich) at a concentration of 0.15 mg g$^{-1}$ of body weight. IVIS imaging was performed on the auto-exposure setting at 10-min post-injection.

## DNL assays

**Compound screening.** Primary mouse hepatocytes were isolated from C57BL/6J mice as previously described[55] and seeded in white opaque 96-well plates, and the next day were serum starved for 2 h followed by the treatment with EVT compounds (0, 0.1, 0.3, 0.5, 1, 3, 10, 30, 60 and 100 μM) in the presence of $^{14}$C-acetate (1 μCi ml$^{-1}$) in a concentration-dependent manner. After 4 h of treatment, plates were washed two times with 1× PBS and 100 μl microscint fluid (Microscint O, part #601361) was added to each well. Plates were wrapped with aluminium foil and were shaken at 250 rpm for 2 h. After 2 h of shaking, $^{14}$C incorporation into the lipid fraction was determined by liquid scintillation counting using a TopCount NXT Microplate Scintillation and Luminescence Counter (Perkin Elmer).

To compare the effect of bempedoic acid and EVT0185 in WT and *Acly*-KO mice, *Acly*$^{fl/fl}$ mice were injected with hepatocyte-targeted AAV8-TTR expressing either YFP (WT) or Cre recombinase (*Acly* KO) via the tail vein to induce *Acly* genetic deletion as previously described[26]. After 2 weeks, hepatocytes from each mouse were collected and de novo lipogenesis (DNL) was assessed as previously described[26]. In brief, the cells were resuspended in complete William's Media E and plated in six-well plates and allowed to adhere for 4 h. Cells were then washed with warm PBS and switched to fresh fetal bovine serum (FBS)-free (serum-free) William's Media E for 2 h. Following a 2-h serum starvation,

cells were washed in warm PBS, then treated with serum-free media supplemented with 1 μCi ml$^{-1}$ [$^{14}$C]-lactate (NEC599050UC, Perkin-Elmer) or 0.5 μCi ml$^{-1}$ [$^{14}$C]-acetate (NEC553050UC, Perkin-Elmer) for 18 h in the presence of 1 μM of EVT0185 or bempedoic acid both of which were dissolved in DMSO. After incubation, cells were washed three times with ice-cold PBS, scraped with 1 M KOH/EtOH, and incubated for 2 h at 70 °C with gentle agitation. After cooling, a 1:2 mixture of H2O:n-hexane was added to each sample, vortexed and centrifuged for 5 min (1,500 rpm). The top phase was transferred to a scintillation vial combined with Ultima Gold (NC0169557, Revvity) for counting of the sterol fraction. 2 N HCl and petroleum ether were added to the remaining bottom phase, which was vortexed and centrifuged for 5 min (1,500 rpm) at room temperature. The top phase was transferred to a scintillation vial for counting of the fatty acid fraction using a TopCount NXT Microplate Scintillation and Luminescence Counter (Perkin Elmer).

**In vivo assessment of liver DNL.** Male C57BL/6J mice (8 weeks of age) were obtained from Jackson Laboratory and were fed a normal chow diet upon arrival. At approximately 10 weeks of age, mice were given a diet containing high-fat and high-fructose (rodent diet with 40% kcal fat (mostly palm oil), 20% kcal fructose and 0.02% cholesterol (D19101102, Research Diets) and housed at thermoneutral conditions (26–29 °C). After 7–8 months, the mice were divided into four groups and received a single daily dose via gavage of either vehicle (1.5% CMC and 0.2% Tween-20) or EVT0185 at a dose of 10, 30 or 60 mg kg$^{-1}$ for 7 days. On the morning of day 8, animals received a final dose, and after 1 h, $^{14}$C glucose (PerkinElmer) was administered at a concentration of 12 μCi per mouse in a volume of 0.1 ml in 0.9% saline (intraperitoneal). One hour after $^{14}$C glucose was given, animals were anaesthetized by intraperitoneal injection of ketamine–xylazine (150 mg and 12.4 mg kg$^{-1}$, respectively). Blood was drawn through cardiac puncture; the liver was removed, and a sample from the left lobe was frozen in liquid nitrogen. Liver tissue was chipped on dry ice and the weight of the chip was weight recorded (30–50 mg of tissue). Liver tissue was homogenized in 1 ml of 2:1 chloroform:methanol using a bead homogenizer at 5,000 rpm for 2 × 12 s. Samples were incubated with gentle shaking at 4 °C for 2 h, vortexed for 2 × 12 s, and then centrifuged at 7,000 rpm for 10 min at 4 °C. The supernatant was transferred to a 1.5-ml Eppendorf tube and 200 μl of 0.9% saline was added. Samples were vortexed for 2 × 12 s and centrifuged at 3,000 rpm for 10 min at 4 °C. Next, 200 μl of the lower organic phase was removed and added to 5 ml of scintillation fluid. The amount of radioactivity in the sample was measured by scintillation counting. The number of disintegrations per minute was determined over 5 min and normalized to the amount of liver tissue. Of plasma obtained at termination, 5–10 μl was also counted and lipid per gram of tissue counts were normalized to plasma counts.

## Measurement of respiratory exchange ratio by indirect calorimetry

Male C57BL/6J mice were fasted overnight (approximately 12 h) until food and 30% fructose water were again made accessible the next day at 7:30. Animals were allowed to feed ad libitum for approximately 2 h. The respiratory exchange ratio (RER) was monitored in metabolic cages using Oxymax/CLAMS (Comprehensive Laboratory Animal Monitoring System) equipment and software (Columbus Instruments). The calorimetry system consists of metabolic cages each equipped with water bottles and food hoppers connected to monitor food intake. All animals had ad libitum access to standard rodent chow and water throughout the study. Those animals with RER > 1 were administered with or without EVT0185 at a dose of 100 mg kg$^{-1}$ via gavage. Animals remained in the metabolic cages and RER was monitored for 24 h.

## Analytical methods

**AFP ELISA.** Plasma AFP levels were determined using a mouse AFP Duoset ELISA kit (DY5369-05, R&D Systems) following the manufacturer's

protocol. In brief, 100 μl sample (1:1,000 in reagent diluent) was added to each well of a 96-well plate coated with mouse AFP capture antibody. Next, the plates were washed with wash buffer and 100 μl detection antibody was added to each well, followed by the Streptavidin HRP and substrate solution. The reaction was stopped by stop solution and the optical density of each well was measured in a microplate reader set to 450 nm.

**Histological analysis.** Following fixation with 10% neutral-buffered formalin for 48 h, the cassettes with a medial lobe of the liver were switched to 70% ethanol. Livers were then processed, paraffin-embedded, serially sectioned and stained with haematoxylin and eosin (H&E), antibodies targeted to CD3[+] and CD19[+] cells by the McMaster Immunology Research Centre Histology Core Facility. Images were taken using a Nikon 90i Eclipse upright microscope at indicated magnifications.

Liver histology scores were assigned to liver sections by a pathologist who was blinded to the treatment conditions. Steatosis, inflammation and hepatocellular ballooning scores were assigned from H&E-stained liver sections as described by Kleiner et al.[56]. NAFLD activity scores (NAS) were obtained from the sum of these three scores. The surface area of tumour in the liver was quantified using qupath software from H&E-stained slides. The quantification of tumour CD3 and CD19-positive staining was completed from the same histological sections using HALO software by an analyst who was blinded to the treatment conditions.

The H-score system was used for ACLY immunohistochemistry evaluation on tumoural and non-tumoural hepatic tissues. This was determined by the multiplication of the percentage of cells with cytoplasmic staining intensity ordinal value (0 for no, 1 for weak, 2 for medium and 3 for strong staining), which ranges from 0 to 300 possible values. The percentage of each staining intensity was taken into account by the eye and multiplied by the staining intensity. The whole tumoural areas of each liver were evaluated in ×10 magnification as well as non-tumoural areas, and H-scores were calculated separately.

$$H-\text{score} = (\%\text{ weak staining})(1) + (\%\text{ medium staining})(2)$$
$$+ (\%\text{ heavy staining})(3)$$

Hepatic lesions were characterized into neoplastic and non-neoplastic proliferative lesions by a histopathologist according to the International Harmonization of Nomenclature and Diagnostic Criteria for Lesions in Rats and Mice (INHAND) guidelines[57]. All lesions had a hepatocellular phenotype. Non-neoplastic proliferative lesions, including foci of altered hepatocytes (FAHs), are referred to as premalignant lesions because they are composed of phenotypically altered cells demonstrating a higher risk of progression to malignancy than normal cells[58,59]. FAHs were defined as sharply demarcated circular, oval or irregularly shaped foci with preserved hepatic architecture and without compression of surrounding liver parenchyma. The basophilic-type FAH was the dominant type observed in this study, composed of slightly smaller hepatocytes with basophilic cytoplasm and slightly enlarged pleomorphic nuclei with prominent nucleoli, showing increased mitotic activity compared with background hepatic parenchyma. Scattered eosinophilic FAHs were also identified, composed of enlarged, polygonal hepatocytes with distinctly granular and pale pink, intensely eosinophilic or ground-glass appearance, as well as few clear cells and mixed cell types of FAHs. Neoplastic lesions as HCCs were characterized by loss of normal architecture (thickened hepatocytes plates), revealing mostly solid patterns as well as focal pseudoglandular and trabecular (micro or macro) patterns, including unpaired arteries or arterioles, consist of polygonal cells with more nuclear atypia (irregular nuclear membrane, pleomorphism, increased nuclear-to-cytoplasmic ratio, multinucleation and prominent nuclei), higher mitotic rate and cytoplasmic alterations include Mallory–Denk bodies and hyaline bodies[57].

## Transcriptome analysis

**Bulk RNA-seq analysis. RNA isolation.** Frozen tumour tissue (approximately 30 mg) was homogenized with Buffer RLT (RNeasy Mini Kit, 74106, Qiagen), and RNA isolation was performed using the RNeasy kit (74106, Qiagen) following the manufacturer's instructions. Sample quality was assessed using the Agilent 2100 Bioanalyzer G2938C with the Aligent RNA 6000 Nano Kit (Agilent).

**RNA-seq differential gene expression analysis.** Only samples passing the RNA integrity number threshold of 7.0 were used for sequencing. mRNA was enriched using the NEBNext Poly(A) mRNA Isolation Module (NEB) followed by library preparation using the NEBNext Ultra II Directional RNA library kit (NEB). Next-generation sequencing was conducted at the McMaster Genomics Facility, Farncombe Institute, McMaster University, using Illumina HiSeq 1500 (Illumina). Samples were randomly distributed across lanes of a HiSeq Rapid v2 flow cell to eliminate lane-specific effects, and single-end 50-bp reads were generated at 12.5 million clusters per sample. Sequence quality was assessed using FastQC followed by the removal of low-quality reads and adapter sequences using Cutadapt. Genome alignment was performed using HISAT2 and the *Mus musculus* mm10 reference genome. Quantification of reads was performed using feature Counts.

**Comparison of mouse models.** We obtained raw RNA-seq files for tumour samples derived from control-DEN and WD-CCl₄ models from PRJNA488497 (ref. 28) and PRJNA386995 (ref. 29), respectively. Sequencing files were processed using the same method as described for in-house samples. Count-level expression data were adjusted for surrogate batch effect across all samples using ComBat-Seq as implemented in the sva package in R. Subsequently, adjusted expression data were normalized to transcript per million. Molecular classification of Hoshida subtypes for human sample and mouse models were performed using the nearest template prediction module as implemented in the Gene Pattern platform online. Pairwise correlation between each mouse sample and each human sample were averaged on a per-model basis for mouse models, and on a per-subtype basis for human samples. Consistent with a previous report[15], we validated the unique transcriptomic enrichment of the Hoshida S1 subtype[60] and depletion of the S2 subtype among patients with MASH-HCC relative to HCC of mixed aetiology shown in Extended Data Fig. 1o.

**Microarray.** Microarray CEL files were downloaded from GSE164760. Normalization was performed using the robust multiarray average expression method implemented in Affy (v1.74.0). The HGU219 custom chip definition file was obtained from the Brainarray Microarray Lab (http://brainarray.mbni.med.umich.edu/Brainarray/Database/CustomCDF/) and used for mapping probes to gene identifiers in a one-to-one format. Surrogate variable analysis was performed using sva (v3.44.0) to identify the presence of any unknown sources of variation by modelling different types of tissue as the variable of interest.

**Differential gene expression analysis.** For RNA-seq, overall and timepoint-specific differential gene expression results were obtained using the DESeq2 package (v1.36.0) by modelling the additive and interaction effect of timepoint and *Acly* KO. To obtain the overall result, we adjusted association of gene expression with *Acly* KO by a binary indicator of experimental timepoint (that is, early or late) as a confounding variable using an additive model (that is, time + *Acly* KO). Timepoint-specific results were obtained by modelling gene expression association using only samples belonging to either early or late timepoints. Finally, we modelled the modifying effect of timepoint on differential gene expression by using an interaction model (that is, time × *Acly* KO). We used a false discovery-adjusted threshold of 5% to determine gene significance. Individual sample read counts underwent variance stabilizing transformation. Gene ranks were obtained for downstream gene set enrichment analysis by performing natural logarithmic transformation of P value and applying the numerical sign of log₂ fold change based on DESeq2 result.

Differential expression analysis for microarray data was performed using limma (v3.52.3). Comparisons between tissue were performed by specifying all combinations of contrast involving MASH-HCC. Comparison using *ACLY* expression level as an independent variable was performed by first using a linear model to identify overall fold change direction followed by a natural cubic splines model with 5 degrees of freedom to identify gene significance.

**WGCNA.** WGCNA (v1.71) in mouse tumour tissues was restricted to genes with greater than or equal to 15 counts in 75% of samples. A soft power of 14 was selected, resulting in a scale-free topology fit of 0.875 WGCNA for human MASH-HCC samples utilized a soft power of 12, resulting in a scale-free topology fit of 0.812. We constructed signed module networks for both mouse and human datasets using the robust correlation method and restricted the number of excluded outliers to 0.10 as recommended. Modules below the hierarchical clustering height of 0.25 were merged. Significant module genes and associated traits were identified using Pearson correlation and asymptotic $P$ value for correlation.

**Gene Ontology and gene set enrichment analysis.** Over-representation and gene set enrichment analysis were conducted using clusterProfiler (v4.4.4). For over-representation analyses, we further calculated fold change as the ratio of a gene-to-background ratio. We used an adjusted $P$ value of 0.05 as the threshold for significance. To identify clusters of similar Gene Ontology terms, we utilized the semantic similarity and binary cut method implemented in simplifyEnrichment (v1.6.1).

**Cell-type deconvolution.** Mouse cell-type deconvolution was performed using transcript per million normalized gene expression data and mMCP-counter (v1.1.0). Validation of B cell enrichment was performed using methods implemented in the online TIMER2.0 tool[61]. For human cell-type deconvolution, we used TIMER2.0 in addition to the ESTIMATE package (v1.0.13). We filtered cell types for those containing greater than 0 count in at least 75% of samples. Association with gene expression was assessed using univariate linear regression. Significance was determined using a false discovery-adjusted threshold of 5%.

**Spatial transcriptomic analysis.** Liver tissues were collected and fixed with 10% neutral-buffered formalin for 36–48 h. After fixation, the samples were immersed in a 70% alcohol solution. The liver tissues were then processed, paraffin-embedded and sectioned. RNA quality of formalin-fixed, paraffin-embedded tissues was tested by TapeStation system with DV200 > 30% (DV200 (distribution value 200) refers to the percentage of RNA fragments that are greater than 200 nucleotides in length). Slides with 5 μm of tissue sections were processed with deparaffinization, H&E staining, imaging, decrosslinking, hybridization, ligation, probe extension, pre-amplification and probe-based library construction to generate gene expression libraries for each tissue for sequencing. Images were taken using the Nikon 90i Eclipse upright microscope. RNA-seq was performed using the Illumina NextSeq 2000 (P2 Flow cell, 2 × 50 bp configuration) system. Sequence data from formalin-fixed, paraffin-embedded tissues were analysed using the Space Ranger count pipeline from 10X Genomics for raw data processing and quality control. Downstream analyses included quality control, normalization, dimensional reduction and clustering, variable gene selection, spatially variable feature detection, annotation, differential expression and integration with multiple samples using Seurat[62]. Spatial transcriptomics data analyses were performed using the Linux system, R, RStudio software and Python programming language. The B cell subtypes within the tumours were examined using established markers[63].

### scRNA-seq analysis
Frozen mouse liver samples (WD-DEN and WD-CCl₄ HCC models) were minced, fixed and dissociated following the 10X Genomics 'tissue fixation and dissociation for chromium-fixed RNA profiling' protocol (document number CG000553, Rev B). Probe hybridization,

library construction and sequencing followed the 10X Genomics 'chromium-fixed RNA profiling reagent kits for multiplexed Samples' (document number CG000527, Rev D). Sequencing was performed using the Illumina Novaseq 6000 sequencing system (S4, 2 × 150, 2–2.5 billion reads, targeting 5,000 cells per sample at a depth of 25,000 read pairs per cell). Demultiplexing was performed using Cell Ranger via the 10X Genomics Cloud Analysis platform. Demultiplexed samples were integrated by condition and processed in R using Seurat v5.

Mouse–human correlation analysis was performed in R. Mouse gene symbols were converted to their 1-to-1 orthologous human EnsemblIDs using the biomaRt package. Normalized count data were filtered for each dataset to include only genes common between all datasets. Each dataset was then filtered to include only cells expressing specific marker genes (those encoding CD45, AFP or GPC3, respectively). Pairwise Pearson correlation values were calculated between the human MASH-HCC data[64] and WD-CCl₄ mouse data[15], using normalized gene expression in the relevant cell type. Results were visualized with the pheatmap package. Human gene-specific mean expression data were generated with Seurat (v5) in R. Wilcoxon rank-sum tests were used to assess the significance of differential expression of *ACLY* and *SLC27A2* in different human conditions, relative to the MASH-HCC condition ($P < 0.05$ considered significant).

**Single-nucleus sequencing analysis.** Single-nucleus sequencing of cells derived from healthy, MASLD, MASH-HCC tumour and tumour-adjacent tissue were obtained from GSE174748 and GSE189175 (ref. 64) and processed using the Seurat package (v5.1.0) in R. We applied sctransform to normalize RNA read counts using the default selection of 3,000 genes as the number of variable features. Transformed counts underwent dimension reduction using principal component analysis. The 30 most variable components were then selected for uniform manifold approximation and projection. Clusters were then identified using the Louvain clustering method at a resolution of 0.2. Cell types were identified using the automated cell-type assignment pipeline as implemented in the sc-type package in R[65]. We utilized the custom gene set containing liver-specific cell-type expression markers provided by the GepLiver resource[66]. SCT normalized counts were used for this analysis. Low-confidence clusters based on the default score threshold was assigned with the 'unknown' cell type. The proportion of cells expressing *ACLY* was determined for hepatocytes and malignant hepatocytes and compared between tumour and non-tumour adjacent tissues using $\chi^2$ test of independence.

### MIBI-TOF methods
**Tissue staining.** Tissues were sectioned (4-μm-thick sections) onto gold-coated MIBI slides. Slides were baked at 65 °C for 1 h, followed by deparaffinization and rehydration in sequential washes in xylene (3×), 100% ethanol (3×), 95% ethanol (2×), 70% ethanol (2×) and MIBI-water. Antigen retrieval in a pH 9 Target Antigen Retrieval Solution (DAKO Agilent) occurred at 125 °C for 40 min in a Decloaking Chamber (Biocare Medical). After cooling to room temperature, slides were washed twice in TBS-T (Ionpath). The tissue was incubated in a blocking buffer consisting of 3% normal donkey serum (Jackson ImmunoResearch) in TBS-T for 20 min. The antibody panel consisting of metal-tagged antibodies supplied by Ionpath was diluted in blocking buffer and added to the tissue for overnight incubation at 4 °C in a moisture chamber. After overnight incubation, tissues were fixed and dehydrated in sequential washes in TBS-T (3×), 2% glutaraldehyde (5 min), Tris pH 8.5 (3×), MIBI-water (2×), 70% ethanol (2×), 90% ethanol (2×), 95% ethanol (2×) and 100% ethanol (3×). Slides were stored in a desiccator before MIBIscope analysis. The antibody panels and reagents are listed in Supplementary Tables 3 and 4.

**MIBI-TOF image acquisition.** Spectral images of stained liver lesions were collected using an Ionpath MIBIscope with multiplexed ion-beam

imaging technology. Xenon primary ions from a Hyperion ion gun rastered across the slide to sputter stained tissue, which was detected by mass spectrometry by time of flight to reconstruct the spectral images, on a pixel-by-pixel basis, of each channel consisting of a single stained antibody. A more detailed description of the multiplexed ion-beam imaging technology has been previously described[42]. Spectral images of 400 × 400-µm regions of interest (ROIs) were selected with the assistance of a pathologist and the images were collected using an Ionpath MIBIscope with multiplexed ion-beam imaging technology. These high-resolution images were denoised and filtered, and with the nuclear and membrane channels, we created segmentation masks using Mesmer[67]. For *Acly*-KO cohort, 35 400 × 400-µm regions for each experimental group, representing 43,974 cell objects, were acquired, whereas for the EVT0185 cohort, 24 400 × 400-µm regions for each experimental group were collected for a total of 30,191 cell objects.

**Low-level image processing.** Multiplexed raw image sets were denoised and aggregate filtered using Ionpath's MIBI/O software using the default correction settings. Processed image data were stored as TIFF files for further data processing and analysis.

**Cellular segmentation.** Whole-cell segmentation was performed with the input of the nuclear stained and the membrane-stained marker channels using Mesmer[67], and the segmentation mask images were stored as TIFF files for further analysis in R Studio.

**Single-cell phenotyping.** Single-cell data were extracted for all cell objects defined by the segmentation masks using a custom R script and packages described in Supplementary Table 5 (refs. 68–71). The aim of the single-cell analysis was to classify each of the objects in terms of their individual marker intensities to construct a profile of the tissue architecture and identify the proximity and relative spatial distribution of cells and features across the tumours. Marker intensities were asinh transformed by a cofactor of 1. To classify cell types based on their marker expression levels, the FlowSOM clustering algorithm was used with the Bioconductor 'FlowSOM' R package[72]. The algorithm clustered the single cells from the cohort into 100 FlowSOM clusters. By inspecting a heatmap displaying normalized individual marker intensities, we annotated each of the 100 clusters into ten cell types.

**Selection of representative ROIs for cellular neighbourhood spatial analysis.** The tumour ROIs were selected based on the presence or high abundance of tumour-infiltrating lymphocytes. Regions with the highest CD45-positive marker expression were selectively chosen as the representative of the cohort. The invasive margin was defined at the border of the malignant tumour and mostly consisted of approximately 50% malignant and 50% non-malignant tissue. The tumour and non-tumour regions were omitted from analysis. For each treatment group, four tumours were sampled for a total area of 1.6 mm². 

**Cellular neighbourhood analysis.** We used the imcRtools package to detect cellular neighbourhoods. A spatial graph was constructed by detecting the *k*-nearest neighbours in 2D per cell[73]. First, for each cell, we used the aggregateNeigbors function to compute the fraction of cells of a certain cell type among its neighbours. Second, for each cell, the function aggregated the expression counts across all neighbouring cells. On the basis of the fraction of the different cell types among the ten nearest neighbours, *k*-means clustering was used to group cells into a user-defined number of cellular neighbourhoods. The choice of six cellular neighbourhoods was determined by a parameter sweep across varying *k* values.

## Lipid droplet analysis

Morphometric lipid droplet analysis was adapted from the previously published article[74]. In brief, H&E sections of tumour-bearing livers were processed using ImageJ for the average size of lipid droplets and average lipid area. Original H&E-stained slides were converted to an 8-bit greyscale image, which was then black and white inverted. These images were used for lipid droplet analysis using the following plugin: run ('Analyze Particles…', 'size = 50–20,000', circularity = 0.50–1.00 display summarize'), which defined the upper and lower limits to a lipid droplet area (50–20,000 µm²)[75] to exclude irregular-shaped structures (lipid droplets are more spherical).

## Tumour free fatty acids

Quantitation of free fatty acid levels was carried out using ultra-high-performance liquid chromatography–multiple reaction monitoring–mass spectrometry (UPLCMRM–MS) method as previously described[76]. In brief, a stock solution of fatty acids was prepared with their standard substances in LC–MS grade isopropanol:methanol (1:1). This solution was serially diluted 1:4 (v/v) to make 10 working standard solutions with the same solvent. A volume of the extractant equivalent to 2 mg of raw tissue was dried under a nitrogen gas flow. The residue was resuspended in 20 µl of LC–MS grade isopropanol:methanol (1:1). Each sample solution or 20 µl of each standard solution was then mixed in turn with 40 µl of 200 mM 3-nitro phenylhydrazine (3-NPH)-HCl solution and 40 µl of 150-mM EDC-HCl-6% pyridine solution. The mixtures were allowed to react at 40 °C for 60 min. After the reaction, 50 µl of each solution was mixed with 100 µl of a mixture of 13C6-3NPH derivatives of all the targeted organic acids. Of aliquots of the resultant solutions, 6 µl was injected onto a C18 column (2.1 × 100 mm, 2.5 µm) to run UPLCMRM–MS with (−) ion detection on an Agilent 1290 UHPLC system coupled to a Sciex 4000 QTRAP MS instrument, with the use of 1 mM ammonium acetate in water (A) and acetonitrile:isopropanol (1:1, v/v) as the mobile phase for binary-solvent gradient elution in a range of 30–100% B over 16 min at 0.4 ml min⁻¹ and 60 °C.

## Tumour TCA metabolites

The tumour tissues isolated were snap-frozen and stored in liquid nitrogen. Around 20–30 mg tissues from each group were chipped and homogenized (with 80% methanol)[77]. The supernatant was dried for gas chromatography (GC)–MS sample preparation[78]. Dried samples were further derivatized in 50 µl of 10 mg ml⁻¹ methoxamine hydrochloride for 60 min at 42 °C, followed by 100 µl of *N*-tert-butyldimethylsilyl-*N*-methyl trifluoroacetamide for 90 min at 72 °C. Next, TCA metabolites were separated by Agilent 7890B gas chromatograph with HP-5ms Ultra Inert GC column (30 m, 0.25 mm, 0.25 µm, 7 inch; 19091S-433UI, Agilent). Metabolites were then analysed by Agilent 5977B mass-selective detector, using full-scan mode. Total metabolite abundances were measured as the area of the total ion counts normalized to protein content.

## Tumour CXCL13 protein

Liver tumour CXCL13 protein levels were determined using the mouse CXCL13/BLC/BCA-1 Quantikine ELISA Kit (MCX130, R&D Systems) following the manufacturer's instructions. In brief, 50 µl of tissue homogenate was added to each well of a 96-well plate coated with a monoclonal antibody specific for mouse BLC or BCA-1 for 2 h. Next, the plates were washed with wash buffer and 100 µl mouse BLC–BCA-1 conjugate was added to each well for the next 2 h followed by the addition of substrate and stop solution. The optical density was measured using a microplate reader set at 450 nm.

## Immunofluorescence microscopy

Paraffin-embedded, 5-µm sections of tumour-bearing livers were used for immunofluorescence. Slides were deparaffinized, rehydrated and subjected to antigen retrieval (10 mM sodium citrate buffer). Slides were blocked in BSA and subsequently incubated with primary antibody overnight at a concentration of 1:500 CCP3 (Cell Signaling Technology) and Ki67 (Thermo Fisher). Fluorophore-conjugated secondary

antibodies were then applied to tissues and mounted with coverslips. Slides were imaged on an inverted confocal microscope (Leica Microscope Systems). Representative images were acquired and analysed using ImageJ analysis.

## Cell-free activity assays for ACLY, ACC1, ACC2, ACSS2 and AMPK

All cell-free assays were conducted at Reaction Biology. Human ACC1 (50202), ACC2 (50201) and ACLY (50255) were obtained from BPS Biosciences. Recombinant protein of human ACSS2, transcript variant 1 (accession no. NM_018677) tagged with MYC–DDK was obtained from Origene (TP304260). ADP-Glo (V9101, Promega) was used to measure ACLY, ACC1 and ACC2 enzyme activity following the manufacturer's instructions. The assay was performed in two steps: (1) after the specific enzyme-mediated reaction that utilizes ATP and produces ADP, ADP-Glo reagent was added to terminate the kinase reaction and depleted the remaining ATP; and (2) kinase detection reagent was added to convert ADP to ATP and allow the newly synthesized ATP to be measured using a luciferase–luciferin reaction. The light generated, measured by the luminometer, correlated to the amount of ADP generated, which is indicative of ACLY, ACC1 or ACC2 activity. ACSS2 activity was assessed using the Bellbrook Transcreener AMP2/GMP2 Assay kit, a far-red competitive fluorescence polarization assay. AMP produced in the reaction was detected by AMP/GMP Alexa fluor 633 tracer bound to an AMP2/GMP2 antibody. AMP displaced the tracer allowing free rotation of the fluorophore and decreased fluorescence polarization, which was quantified using an appropriate multimode plate reader. Human AMPKα1–AMPKβ1–AMPKγ1 heterotrimers were co-incubated with the AMPK substrate peptide at 20 µM, AMP at 125 µM and ATP at 10 µM using the HotSpot assay.

## ACLY competitive assay

Human recombinant ACLY was obtained from Sino Biological (11769-H07B). Recombinant full-length hACLY (accession no. P53396) was expressed in insect cells using baculovirus expression host, with a poly histidine tag at the N terminus. The molecular weight was 123 kDa. The ACLY ADP-Glo assays were performed at room temperature. Enzyme and sodium citrate were prepared in reaction buffer at three times the reaction concentration and 5 µl was added to Corning 3572 reaction plate wells. Of compound concentrations, 100× were prepared by serial dilution in 100% DMSO. Then, the compound was transferred to the reaction plate, which contains enzyme and sodium citrate, using acoustic technology. The compound and enzyme were pre-incubated for 15 min at room temperature. Then, 5 µl of 3× coenzyme A was delivered to the reaction well, followed by 5 µl of 3× ATP to start the reaction. The assay was incubated for 2 h at room temperature. For the background wells, all assay components were added without enzyme. After a 2-h reaction, ADP-Glo reagent was added to stop the reaction and incubated for 40 min at room temperature following detection kit protocol. Then, ADP-Glo detection was added and incubated for 30 min at room temperature following the detection kit protocol. The luminescence signal was measured after 30 min of incubation with ADP-Glo detection. The luminescence signal was then converted to µM ADP using ADP standard curve. The compound was tested in triplicate and each dataset was analysed separately.

## Detection of EVT0185-CoA thioester using LC–MS/MS method

**HEK293 cell culture and transfection validation.** HEK293 cell line was obtained from American Type Culture Collection (ATCC) and grown in Dulbecco's modified eagle medium (319-005-CL, Wisent Bioproducts) supplemented with 10% FBS (098150, Wisent Bioproducts) in a humidified incubator at 37 °C and 5% $CO_2$. HEK293 cells were transfected with expression plasmid for SLC27A1 (NM_198580, human tagged ORF clone; RC209285, OriGene), SLC27A2 (NM_003645, human tagged ORF clone; RC221033, OriGene), SLC27A4 (NM_005094, human tagged ORF clone; RC209557, OriGene), SLC27A5 (NM_012254, human tagged ORF

clone; RC215758, Origene) or control vector (pCMV6-entry mammalian expression vector; PS100001, OriGene). Transfected cell lines were supplemented with G418 (A1720, Sigma-Aldrich) during maintenance at a concentration of 800 µg ml⁻¹. Transfected SLC27A2 protein was detected by western blot using the antibody DYK tag (2368, Cell Signaling); while endogenous SLC27A2 was detected using the polyclonal antibody (PA5-30420, Invitrogen).

**Cell treatment and extraction.** The day before the experiment, HEK293 cells overexpressing SLC27A1/A2/A4/A5 were plated in six-well plates at $1-1.5 \times 10^5$ cells per cm² in Dulbecco's modified eagle medium and cultured overnight. Cells were treated with 30 µM of EVT0185 for 3 h. At the end of the treatment period, medium was removed, cells were washed once with ice-cold PBS and were flash frozen in liquid $N_2$. For extraction, plates were placed on ice, and cells were scraped in 266 µl of PBS per well. The cell suspension from three wells were pooled (approximately 800 µl total volume) and added to 1 ml of ice-cold 2:1 dichloromethane:methanol in a glass vial. The samples were vortexed for $2 \times 12$ s and then centrifuged at 300g for 5 min at 4 °C. The upper aqueous phase was transferred to a new tube and filtered through a 0.45-µm PTFE filter. Samples were delivered to the Centre for Microbial Chemical Biology at McMaster University for LC–MS/MS analysis.

**LS–MS/MS analysis.** Samples were evaporated using a Turbovap LV (Biotage) evaporation system with the water bath set to 30 °C. Dried samples were resuspended in 100 µl of methanol, then analysed by injecting 5 µl into an Agilent 1290 Infinity series HPLC (Agilent) coupled to an LTQ-Orbitrap XL mass spectrometer (Thermo Fisher Scientific). Separation of the analytes was achieved on an Agilent Eclipse XDB-C18 (100 mm × 2.1 mm, internal diameter (i.d.), 3.5 µm) analytical column using mobile phases consisting of (A) 10 mM ammonium acetate in water, pH 8.5, and (B) acetonitrile and a constant flow of 0.2 ml min⁻¹. The analytical column was maintained at 30 °C. The separation of analytes was achieved using a gradient starting at 20% B, which was held for 1.5 min. The gradient increased to 95% B over the next 3.5 min and was held for 4.5 min. The column was then re-equilibrated back to the initial conditions over the final 5.5 min. The total run time was 16 min. The HPLC eluent was introduced into the mass spectrometer using electrospray ionization in positive mode with a spray voltage of 3.6 kV. The mass spectrometer was set to acquire spectra in data-dependent acquisition mode. The full MS scan was set to an $m/z$ range of 100–2,000 in the ion trap with a resolution of 30,000. The MS/MS was performed in the Orbitrap with a collision induced dissociation (CID) collision energy of 35. The activation time was set at 30 ms with the activation parameter $q = 0.250$ and an isolation window of 1.5 $m/z$. The resulting mass spectra were analysed using the Xcalibur 2.1 Qualitative software package.

## Clonogenic assay

Hep3B (HB-8064) human hepatocarcinoma and Hepa1-6 (CRL-1830) mouse hepatoma cell lines were purchased from the ATCC and grown in eagle's minimum essential medium (10-009-CV, Mediatech) and Dulbecco's modified eagle medium (319-005-CL, Wisent Bioproducts), respectively, supplemented with 10% FBS (098150, Wisent Bioproducts) and 1% penicillin–streptomycin (15140122, Gibco, Thermo Fisher Scientific) in a humidified incubator at 5% $CO_2$/95% air. Bempedoic acid was purchased from MedChemExpress. Hep3B and Hepa1-6 cells were seeded in 12-well plates at a cell density of 1,000 and 500 cells per well, respectively. On day 2, media in each well were aspirated and replaced with 1,000 µl of fresh complete media and cells were treated with or without the respective drugs in an increasing concentration followed by the incubation of the cells for 7 days in an incubator. On day 9, the media in each well were aspirated and cells were fixed with 10% formalin (500 µl) for 10 min at room temperature. Formalin was aspirated and the plates were washed with 1× PBS and stained with crystal violet for another 10 min. After 10 min of staining, crystal violet was poured over

the cells, rinsed with tap water for three times and the plates were dried overnight. The next day, colonies (more than 50 cells) were counted and analysed.

## Cryo-EM studies on ACLY–EVT0185-CoA

The pTrcHis2-hACLY expression construct[35] for full-length human ACLY (Uniprot ID P53396-2) in frame with C-terminal MYC and His tags was transformed in BL21(DE3) cells. Cultures were grown in Luria–Bertani medium at 28 °C and expression was induced by the addition of 1 mM IPTG at an optical density at 600 nm of 0.6–0.7. After overnight expression, the bacterial culture was collected by centrifugation and bacterial cells were resuspended in IMAC-binding buffer, consisting of 50 mM sodium phosphate, pH 7.4, and 150 mM NaCl supplemented with cOmplete protease inhibitor cocktail without EDTA (Roche). Bacterial cells were lysed by sonication and insoluble material was removed by centrifugation. The resulting supernatant was clarified using a 0.22-µm filter and loaded onto a Ni Sepharose column equilibrated with IMAC-binding buffer. The IMAC column was washed and His-tagged ACLY was eluted with binding buffer supplemented with increasing concentrations of imidazole. Elution fractions were pooled and concentrated using ultracentrifugation. ACLY was further purified by size-exclusion chromatography using HiLoad 16/600 Superdex 200 and Superose 6 (Increase) columns with 20 mM HEPES, pH 7.4, and 150 mM NaCl as a running buffer. The top fractions from the final size-exclusion chromatography elution peak were pooled and concentrated to 10 mg ml⁻¹, aliquoted, flash frozen and stored at −80 °C in a freezer until further use.

For cryo-EM grid preparation, purified ACLY (10 mg ml⁻¹ in HBS buffer) was supplemented with 0.5% CHAPSO, 1 mM Mg₂ATP and 4 mM EVT0185-CoA (freshly prepared from powder). Of the sample, 4 µl was applied to a glow-discharged C-Flat 1.2/1.3 300 mesh copper grid (Protochips), blotted for 4.5 s under 99% humidity at 22 °C and plunged into liquid ethane using a GP2 Leica grid plunger. Grids were screened using a JEOL 1400 Plus microscope equipped with a JEOL Ruby CCD camera at the VIB Bioimaging Core Ghent. Data collection was performed using a JEOL cryoARM 300 microscope equipped with a 6k × 4k GATAN K3 detector resulting in 7,263 movies with a raw pixel size of 0.72 Å (BECM). Movies were processed via patch-based motion correction and contrast transfer function estimation as implemented in cryoSPARC (v3.1.0)[79].

Particles were extracted with a box size of 480 pixels with 2× binning. Good-quality 2D classes were initially obtained via the Blob Picker job in cryoSPARC, followed by template-based picking and neural network-based particle picking via TOPAZ as implemented in cryoSPARC[80]. Ensuing 2D classification, 2D class selection and removal of potential duplicate particles within a distance of 150 Å resulted in a particle set of 210,900 particles. Ab initio 3D classification (number of classes = 5) followed by homogeneous and non-uniform refinement[81] without application of symmetry resulted in a 3D reconstruction that had a resolution of 3.7 Å following a gold-standard non-uniform refinement and represented the pseudo-D2-symmetric ACLY tetramer. In this cryo-EM map, the central, tetrameric CSH module was markedly better defined than the N-terminal CCS modules, indicating structural heterogeneity of the CCS modules with respect to the central CSH module.

To address this flexibility and possibly resolve the CCS–CSH assembly at a higher resolution, we applied symmetry expansion in combination with local refinement followed by 3D classification. Following homogeneous and non-uniform refinement in D2 symmetry, the associated particle set was re-extracted without binning (0.72 Å per pixel) and symmetry expanded around the D2 axes. Using the molmap function in Chimera, a volume blurred to 25 Å that contained one CCS arm and the central CSH module was generated and transformed into a mask with the Volume Tools job in cryoSPARC (map threshold = 0.09, dilation radius = 7.2 Å and soft padding width = 14.4 Å). Gold-standard local refinement was performed by limiting the rotation and shift search

extent around the original consensus refinement poses to 20° and 10 Å, respectively, in combination with a Gaussian prior over the pose or shift magnitudes (with SDROT = 15 Å and SDSHIFT = 7 Å). The centre of mass of the mask was used as a fulcrum point. Ensuing 3D classification without alignment in cryoSPARC using the 3D classification job followed by another round of gold-standard local refinement resulted in a cryo-EM volume with a $FSC_{0.143}$ resolution of 3.3 Å in which the atomic models for the CSS and CSH modules (extracted from Protein Data Bank (PDB) ID 6XHX) were fitted using Chimera and real-space refined in Phenix[82] using reference restraints to the starting model. Cryo-EM map regions representing ligand density in the ATP-grasp fold domain of the CCS module and in the CoA-binding domain were of the CSH module modelled as Mg.ADP and the adenosine 3′-phosphate 5′-diphosphate moiety of bound EVT0185-CoA, respectively. Restraints for EVT0185-CoA were generated via the de Grade Web Server (https://grade.globalphasing.org)[83].

A cryo-EM micrograph for the ACLY–EVT0185-CoA complex is provided in Extended Data Fig. 4b. Cryo-EM data and refinement statistics are summarized in Supplementary Table 2. Reported resolutions are based on the gold-standard $FSC_{0.143}$ criterion[84], and FSC curves were corrected for the effects of soft masking by high-resolution noise substitution[85]. A map-to-model correlation was calculated using phenix. mtriage using the independent half maps as input. Local resolution maps were computed using the blocres algorithm[86] as implemented in cryoSPARC with an FSC threshold of 0.5. All representations of cryo-EM Coulomb potential density maps and structural models were prepared with ChimeraX[87] and PyMol[88]. Cryo-EM map contour sigma levels have been reported based on map normalization in Coot.

**Materials availability.** The protein expression construct for full-length human ACLY is available via the BCCM/GeneCorner Plasmid Collection (http://bccm.belspo.be) through the following accession code: LMBP 11277 (pTrcHis2-hACLY).

Cryo-EM maps following global and local refinement and the real-space-refined model for the CCS–CSH assembly have been deposited in the Electron Microscopy Data Bank (EMDB) with the accession code EMD-53847 and in the PDB with the ID 9R90 (Extended PDB ID pdb_00009R90).

## Generation of stable ACLY knockdown cell line

**Cell culture.** Hep3B cells (ATCC HB-8064) were cultured adherently using Eagle's minimum essential medium (ATCC 30-2003) supplemented with 10% (v/v) FBS (10437-028, Gibco) and 1% (v/v) antibiotic–antimycotic solution (15240112, Gibco). HEK293T (ATCC CRL-3216) was cultured adherently using Dulbecco's modified eagle medium, high glucose (11965092, Gibco) supplemented with 10% (v/v) FBS and 1% (v/v) antibiotic–antimycotic solution as aforementioned along with 1% (v/v) L-glutamine (25030081, Gibco), 1% (v/v) MEM non-essential amino acids solution (11140050, Gibco) and 1% (v/v) sodium pyruvate (11360070, Gibco).

**Plasmid construction.** Dox-inducible shACLY plasmid vectors were generated using the inducible EZ-Tet-pLKO-Hygro vector (Addgene plasmid no. 85972). Two ACLY shRNA sequences—5′-CGTGAGA GCAATTCGAGATTA and 5′-GGCATGTCCAAGCTCAA—and one non-targeting shRNA sequence (5′-CCTAAGGTTAAGTCGCCCTCG) were ligated, transformed, amplified and validated using established protocol as previously described[89].

**Lentivirus production.** A second generation lentivirus production system was used to generate high-titre lentivirus. HEK293T cells were incubated with psPAX2 (Addgene plasmid no. 12260), pMD2.g (Addgene plasmid no. 12259) and the EZ-Tet-pLKO-Hygro (Addgene plasmid no. 85972) transfer plasmid at a ratio of 1:1:2 pmol in the presence of Lipofectamine 2000 in Opti-MEM. Sixteen hours post-transfection, media

were changed to 1% bovine serum albumin (A1470, Sigma-Aldrich) and 1 mM Na-butyrate (3850, Tocris) supplemented with antibiotic-free DMEM (11965092, Gibco) for HEK293T cell culture. Forty-eight hours post-transfection, lentivirus media were collected and mixed with Lenti-X concentrator (631232, Takara) before storing overnight at 4 °C. Next day, lentivirus was concentrated through centrifugation at 1,500$g$ for 45 min at 4 °C. The resulting pellet was resuspended at 1:10 of initial media volume, aliquoted and frozen in liquid nitrogen for storage.

**Lentivirus transduction.** An initial dilution of 1:50 v/v followed by serial dilutions of 1:4 v/v were used to transduce target cell lines at 60,000 cells per well in six-well plates using 5 µg ml$^{-1}$ polybrene (107689, Sigma-Aldrich) supplemented cell culture media. Cells were transduced for 24 h, followed by recovery in lentivirus-free media for another 24 h before selection of successfully transduced cells using a pre-determined, cell line-dependent concentration of hygromycin B (H3274, Sigma-Aldrich) or puromycin (P8833, Sigma-Aldrich) for 7 days. Surviving cells post-antibiotic selection were further propagated to establish stable cell line.

**DNL assay.** Cells were incubated in media supplemented with 1 µCi [$^{14}$C] glucose (NEC042V250UC, Perkin Elmer) and [$^3$H] acetate (NET003005MC, Perkin Elmer) for 4 h. Subsequently, cells were washed and scraped in PBS (10010023, Gibco). Lipids were extracted using a chloroform:methanol (1:2) solution, vortexed for 30 s and centrifuged at 13,000$g$ for 10 min to isolate cellular debris and protein fractions from lipid supernatant. Lipids were further purified with chloroform:water (1:1) solution via vortex and centrifugation as described. Peptide concentration was quantified using the Pierce BCA Protein Assay Kit (23225, Thermo Scientific) for normalization calculation. Of lipid supernatant, 100 µl was extracted from the non-polar chloroform phase and mixed with 5 ml of Ultima Gold Scintillation Fluid (6013329, Perkin Elmer) for radioactivity quantification.

**Fatty acid oxidation assay.** Cells were incubated with serum-free cell culture media supplemented with 2% bovine serum albumin (A1470, Sigma-Aldrich), 500 µM sodium palmitate (P9787, Sigma-Aldrich), 0.5 µCi ml$^{-1}$ [$^{14}$C] palmitic acid (NEC075H050UC, Perkin Elmer) and 1 mM L-carnitine (11242008001, Roche) for 4 h. Cell culture supernatant was collected at the end of incubation and mixed with 1 ml of 1 M acetic acid (34256-1L-R, Fluka). Solution was sealed in a glass vial and shaken at 75 rpm at room temperature for 2 h. Released $CO_2$ was absorbed in 450 µl of 1 M benzethonium hydroxide (B2156, Sigma-Aldrich) during this time. Subsequently, tube containing benzethonium hydroxide was mixed with 5 ml of Ultima Gold Scintillation Fluid (6013329, Perkin Elmer) for radioactivity quantification.

For western blotting, cells were washed with cold PBS (10010023, Gibco) and scraped with 100–200 µl of cold cell lysis buffer (50 mM HEPES, pH 7.4, 150 mM NaCl, 100 mM NaF, 10 mM Na-pyrophosphate, 5 mM EDTA, 250 mM sucrose, 1 mM dithiothreitol and 1 mM Na-orthovanadate, 1% Triton X-100 and cOmplete protease inhibitor cocktail; 11836153001, Roche). Solution samples were collected and centrifuged at 13,000$g$ for 10 min at 4 °C to isolate protein fraction from cell debris. Protein samples were quantified for peptide concentration using the Pierce BCA Protein Assay Kit (23225, Thermo Scientific) according to instruction. Samples were then diluted to 1 µg µl$^{-1}$ using 4× SDS sample buffer (40% glycerol, 240 mM, Tris-HCl, pH 6.8, 8% SDS, 0.04% bromophenol blue, 5% β-mercaptoethanol and 20 mM dithiothreitol) and cell lysis buffer. Samples were then heated for 5 min at 95 °C before immunoblotting. SDS polyacrylamide gels were prepared at 10% or 12% polyacrylamide concentration dependent on protein size. Of the protein sample, 25 µg was loaded into each lane. Gel electrophoresis was performed first at 90 V as protein travels through the stacking gel and then at 120 V, through the separating gel. Protein samples were then transferred onto a PVDF membrane (03010040001, Roche) at 4 °C for 90 min following the standard wet transfer protocol. Subsequently, membranes were rinsed and rocked gently with 1× TBS (50 mM Tris,

150 mM NaCl and 1 M HCl, pH 7.4) for three cycles of 5 min each then blocked using a TBST-BSA solution (1× TBS, 0.5% bovine serum albumin and 0.1% Tween-20) for 1 h at room temperature. Primary antibodies (ACLY (13390; 1:1,000 dilution); p-ACLY S455 (4331; 1:1,000 dilution); and β-actin HRP conjugate (12620; 1:5,000 dilution); all Cell Signaling Technology) were diluted at 1:1,000 in TBST-BSA solution. Membranes were then cut according to protein sizes and incubated in respective antibody solutions overnight at 4 °C with gentle rocking. The next day, primary antibody solutions were removed, and membranes were washed using TBST. Subsequently, membranes were incubated at room temperature with gentle rocking for 1 h in solutions containing 1:10,000 dilution of species-specific secondary antibody conjugated to HRP (anti-rabbit IgG HRP linked; 7074, Cell Signaling Technology). At the conclusion of incubation, membranes were washed again in TBST solution and kept in TBS solution until imaging and for storage.

## Human PBMC culture and T lymphocyte and B lymphocyte proliferation assay

Peripheral blood mononuclear cells (PBMCs) were isolated from donors using the Ficoll Paque density gradient method. PBMCs were then washed with RPMI1640 and pelleted. Cells were resuspended with 10 ml of 3.5 µM CellTrace CFSE (C34570, Thermo Fisher) and incubated at room temperature for 20 min. After incubation, 40 ml complete RPMI1640 was added for the next 5 min to absorb excess CFSE. PBMCs were then washed, resuspended in complete RPMI1640 (10% FBS, 1% penicillin–streptomycin and 1% L-glutamine) and plated at a cell density of $1 \times 10^6$ cells per well in 12-well plates.

For T cell proliferation, PBMCs were induced with 3 µg ml$^{-1}$ of concanavalin A (C5275, Sigma-Aldrich) and 10 ng ml$^{-1}$ of human recombinant IL-2 (200-02, Preprotech). For B cell proliferation, PBMCs were induced with 1 µg ml$^{-1}$ of R848 (36611, mAb Tech) and 10 ng ml$^{-1}$ of IL-2 (36611, mAb Tech). EVT0185 was diluted in DMSO and was added at the final concentration of 0.1 µM, 0.3 µM, 1 µM, 3 µM and 10 µM. Next, PBMCs were incubated at 37 °C and 5% $CO_2$ incubator for 5 days. Cells were then harvested, stained with E780 Live/Dead stain (C34570, Thermo Fisher), CD3 (300434, BioLegend) and CD19 (302241, BioLegend) antibody and analysed using flow cytometry. The results were further analysed using the FlowJo software. After 5 days, cells were then prepared for flow cytometry.

## Flow cytometry

For the B cell depletion study, the tail vein blood was collected from each animal after 15 days of single injection with 250 µg of anti-CD20 (152104, BioLegend) or isotype (400566, BioLegend) antibody. The red blood cells were lysed twice with 1× red blood cell lysis buffer. The samples were then centrifuged at 1,500 rpm for 5 min at 4 °C. The cells were washed and blocked with Fc block (1:200; 553142, BD Biosciences) and stained with CD45.2 BV510 (1:25; 109838, BioLegend), B220 (1:100; 563894, BD Biosciences), CD19 (1:100; 152409, BioLegend) and 7AAD (1:100; A1310, Thermo Fisher Scientific). CytoFlex Flow Cytometer (Beckman Coulter Life Sciences) was used for data acquisition and was further analysed using FlowJo (v10.5).

For PBMC analysis, the cells were harvested after 5 days of treatment and stained with E780 Live/Dead stain (C34570, Thermo Fisher), CD3 (300434, BioLegend) and CD19 (302241, BioLegend) antibody and analysed using flow cytometry. The results were further analysed using the FlowJo software.

## Statistical analysis and reproducibility

Independent $t$-test and analysis of variance (ANOVA) were performed using GraphPad Prism software (v9.5.1). Correlation and regression analysis were performed using R studio. All values are reported as mean ± s.e.m. unless stated otherwise. Differences were considered significant ($*$) when $P < 0.05$. Statistical significance between two independent groups was analysed by unpaired $t$-test. Representative

images shown in Figs. 1e and 5g and Extended Data Figs. 1n and 9d,e were repeated at least four times. Extended Data Fig. 3d,e was repeated two times. Extended Data Fig. 4b shows a representative cryo-EM micrograph from the dataset (7,263 micrographs in total), illustrating the particle distribution and image quality. Histological scores were analysed using the non-parametric Mann–Whitney test when comparing two independent groups. Wilcoxon's test was used to analyse cell type and regional prevalence in MIBI-TOF images. Descriptions of additional software and statistical analysis used for transcriptomic analysis are mentioned in their respective methods.

## Reporting summary

Further information on research design is available in the Nature Portfolio Reporting Summary linked to this article.

## Data availability

Data that support the findings of this study are available within the article and its Supplementary Information. The bulk RNA-seq data of the WD-DEN liver tumours, scRNA-seq data and spatial transcriptomic data of the WD-DEN and WD-CCl₄ livers have been deposited at the NCBI Gene Expression Omnibus (GEO) and are accessible under accession numbers GSE296668, GSE297190 and GSE297081, respectively. For the mouse HCC model comparison, raw RNA-seq files for tumour samples derived from control-DEN and WD-CCl₄ models were downloaded from the NCBI Sequence Read Archive under reference numbers PRJNA488497 and PRJNA386995, respectively. Single-nucleus sequencing of cells derived from healthy, MASLD, MASH-HCC tumour and tumour-adjacent tissue were obtained from GSE174748 and GSE189175. The gel source data are provided in Supplementary Fig. 1. Source data are provided with this paper.

## Code availability

The analysis codes used for bulk RNA-seq, scRNA-seq and spatial transcriptomics are publicly available on GitHub: https://github.com/jianhanwu/acly_project, https://github.com/russtafayyazi/Nature.scRNAseq.CodeDeposition.2025# and https://github.com/wddong-1988Mcmaster/SpatialLiverCancer2025, respectively. The analysis code for MIBI-TOF data has been deposited in Zenodo[90] (https://doi.org/10.5281/zenodo.15518613).

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

**Acknowledgements** Espervita Therapeutics provided financial support for the conduct of the research related to EVT0185 and EVT0185-CoA. G.R.S. acknowledges the support of a Diabetes Canada Investigator Award (DI-5-17-5302-GS), a Canadian Institutes of Health Research Foundation Grant (201709FDN-CEBA-116200), a Tier 1 Canada Research Chair in Metabolism and Obesity and a J. Bruce Duncan Endowed Chair in Metabolic Diseases. J.G., J.W. and F.D.P. acknowledge support by a MITACS postdoctoral fellowship sponsored by Espervita Therapeutics. K.V. acknowledges support from Research Foundation Flanders (FWO; grant 1524918N). S.N.S. is supported by a program grant from the VIB. S.L.F. acknowledges the support of US National Institutes of Health grants 5R01DK128289-03, 5R01DK121154-04 and 5P30CA196521-08. Y.H. acknowledges the support of the US National Institutes of Health grant 5R01CA233794-05. We thank N. Henriquez and T. Campbell at the Centre for Microbial and Chemical Biology at McMaster University for assistance with the LC–MS experiments. We thank the VIB-VUB facility for Biological Electron Cryogenic Microscopy (BECM; Belgium) and the VIB Bioimaging Core Ghent (Belgium) for infrastructural access and technical support.

**Author contributions** The overall conceptualization of studies related to model development, molecular and biochemical analysis, and characterization of *Acly*-KO mice were conducted by G.R.S., J.G., J.W. and J.S.V.L. D.C.O. designed the structure of EVT0185, and the synthesis of EVT0185 and its CoA derivative explained in the Supplementary Information. Representatives of Espervita Therapeutics were involved with the design and interpretation of studies involving EVT0185 (J.S.V.L., J.G., R.S.N., G.R.S., DCO. and S.H.). J.G. designed, performed and managed the cell and animal experiments corresponding to Figs. 1–3 and 5 and Extended Data Figs. 1 and 5, and conducted the biochemical, histological and molecular analyses related to Figs. 4 and 5 and Extended Data Figs. 1, 2, 6, 7, 9 and 11. J.W. designed and executed the Hep3B cell and animal experiments for Extended Data Fig. 10, and conducted analyses of human and mouse bulk RNA-seq analysis presented in Figs. 4 and 5 and Extended Data Figs. 1, 7 and 12. J.S.V.L. designed and performed the CoA detection experiment (Extended Data Fig. 3), an in vivo DNL experiment (Extended Data Fig. 6) and assisted J.G. with animal experiments for Figs. 1 and 3. J.D.M. performed the MIBI-TOF imaging analysis presented in Fig. 5 and Extended Data Fig. 9. R.F. performed scRNA-seq analysis shown in Extended Data Figs. 1, 3, 8 and 12, and assisted J.W. for the bulk RNA-seq analysis (Extended Data Fig. 1). E.A. carried out liver

histological assessments related to Fig. 1 and Extended Data Figs. 1, 2 and 9. S.R. performed the lipid area and immunofluorescence imaging analysis (Fig. 1 and Extended Data Figs. 6 and 10). D.B. and S.L.F. performed the WD-CCl$_4$ mouse experiment (Fig. 3h–j). F.D.P. performed the *SLC27A2* transfection experiment shown in Extended Data Fig. 3. B.N. performed the human T cell and B cell proliferation experiments as presented in Extended Data Fig. 3. L.K.T. and C.M.V. performed the DNL experiments in WT and *Acly*-KO hepatocytes (Extended Data Fig. 5). S.B. assisted J.G. in the validation of the B cell depletion experiments (Extended Data Fig. 11) in collaboration with B.B. and J.L. D.W., J.G., M.J.T.J., A.P. and S.B. prepared samples for spatial transcriptomics. D.W. performed the bioinformatic analysis presented in Fig. 4 and Extended Data Fig. 8. F.D.P. and E.M.D. assisted with viral injections used to generate *Acly*-KO mice (Fig. 1). N.K. performed histological analysis of MASH-HCC mouse and human MASH-HCC tissues (Extended Data Fig. 1n). E.E.T. assisted in CCl$_4$ mouse experiments with J.G. and J.S.V.L. (Fig. 3). Cryo-EM studies associated with Fig. 2 and Extended Data Fig. 4 were conducted by K.V., A.D., K.H.G.V. and S.N.S. W.D. measured the TCA metabolite levels in tumour tissues (Extended Data Fig. 7g). J.G., J.S.V.L., S.B., F.D.P., R.F., M.J.T.J., E.E.T., D.W., J.L., B.M., A.A., V.H., J.W., C.M.V. and L.K.T. assisted with animal euthanization and tissue collection. J.G. led and coordinated the study, managed data collection and organization, and contributed to the finalization of the manuscript under the supervision of G.R.S. G.R.S., K.V., S.L.F., T.T., L.J., Y.H., S.H., R.S.N., D.C.O., J.L.B., M.L., P.M., J.A.H. and K.B. contributed to the experimental design and scientific discussions throughout the study. G.R.S. and J.G. wrote and revised the manuscript with the assistance of J.W., J.D.M., D.W., J.S.V.L. and R.F. The final manuscript was reviewed and edited by all the authors.

**Competing interests** G.R.S., R.S.N., S.H., D.C.O., Y.H., J.S.V.L. and J.G. are shareholders of Espervita Therapeutics. The laboratories of S.L.F., K.V. and G.R.S. received funding from Espervita Therapeutics. The other authors declare no competing interests.

**Additional information**
**Correspondence and requests for materials** should be addressed to Gregory R. Steinberg.

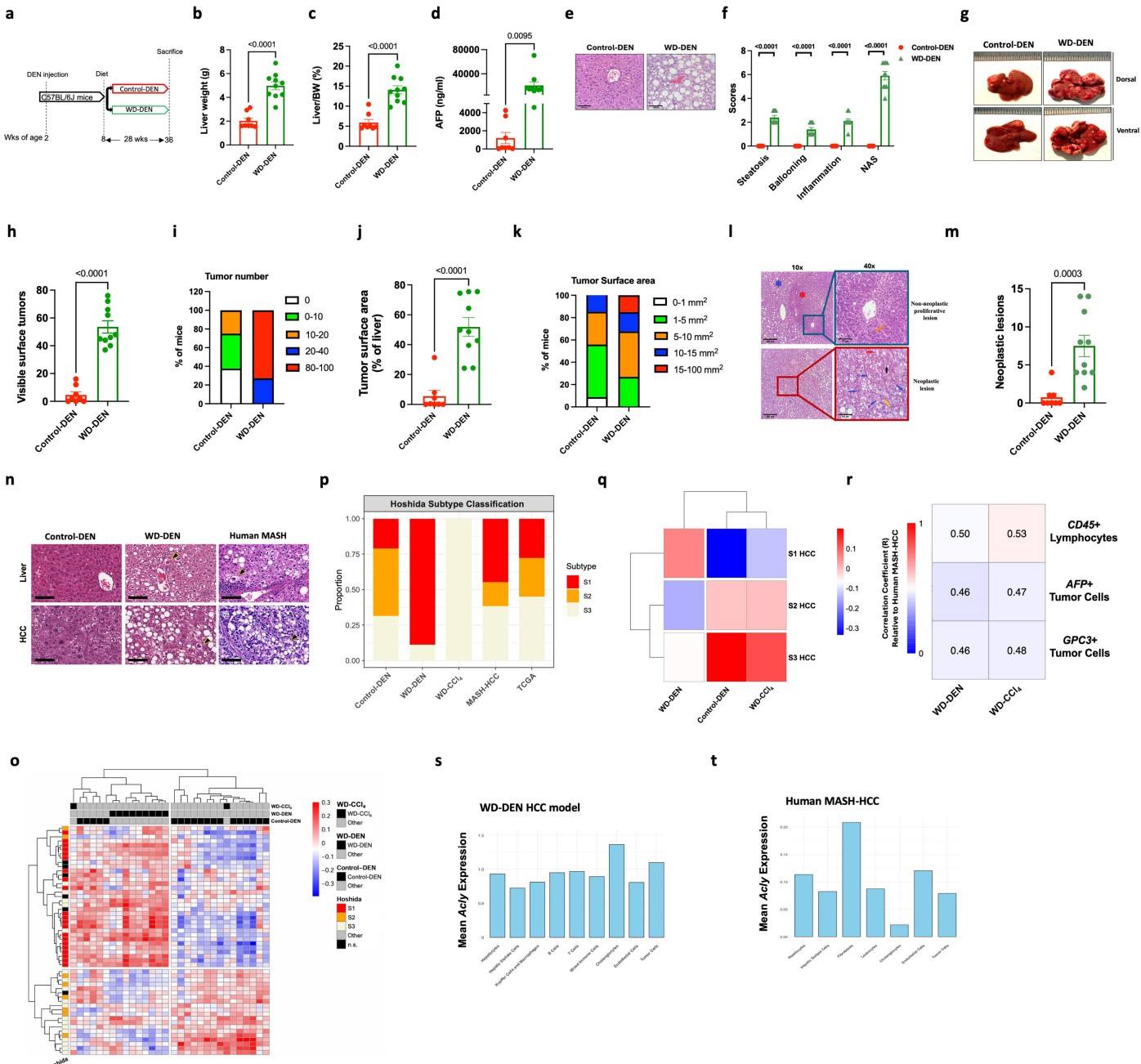

**Extended Data Fig. 1 | Feeding DEN-injected mice a high-fat and fructose (Western Diet (WD)) promotes MASH-driven HCC, which mimics human disease pathology. a**, Study design; DEN-injected animals fed Control Chow (Control-DEN) or Western Diet (WD-DEN). **b**, Liver mass, **c**, Liver mass/BW, and **d**, plasma AFP levels; mean ± SEM, WD-DEN (n = 10) vs Control-DEN (n = 8); unpaired two-tailed t-test: $P = 3.86 \times 10^{-6}$ (b), $P = 1.41 \times 10^{-5}$ (c). **e**, Representative H and E-stained liver sections. **f**, histological scores for steatosis, ballooning degeneration, inflammation, and NAS; mean ± SEM, WD-DEN (n = 10) vs Control-DEN (n = 8); unpaired Mann-Whitney two-tailed test: $P = 2.29 \times 10^{-5}$. **g**, Dorsal and ventral liver images. **h**, Tumor counts; mean ± SEM, WD-DEN (n = 10) vs Control-DEN (n = 8) mice; unpaired two-tailed t-test: $P = 9.47 \times 10^{-8}$. **i**, Tumor count distribution and **j**, tumor surface area; mean ± SEM, WD-DEN (n = 10) vs Control-DEN (n = 8); unpaired two-tailed t-test: $P = 2.18 \times 10^{-5}$. **k**, Distribution of tumor surface area per group. **l**, Histological feature of lesions (blue star: hepatic parenchyma, red star: Non-neoplastic proliferative lesion (FAH), orange arrow: slightly atypical hepatocyte (enlarged nuclei, higher N/C ratio, basophilic cytoplasm), black arrow: mitosis, blue arrows: atypical hepatocytes with

enlarged and occasional binucleated nuclei, irregular nuclear membrane, prominent nucleoli and cytoplasmic alterations, red arrow: balloon cells with Mallory Denk bodies, yellow arrow: inflammatory infiltration). **m**, Number of neoplastic lesions in Control-DEN (n = 8) and WD-DEN (n = 10); mean ± SEM; unpaired two-tailed t-test. **n**, Representative H and E-stained liver images showing similar macro/microvesicular steatosis in hepatocytes and HCC cells and ballooning (arrowhead) in livers from WD-DEN mice and MASH HCC samples from humans. Scale bar = 100 μm. **o**, Average pairwise correlation on a per-model basis for mouse samples (column) and per-tissue or tissue molecular subtype basis for human samples (row). **p**, Proportion of mouse and human tumor tissue classification based on Hoshida molecular subtypes. **q**, Pairwise Pearson correlation between mouse models (columns) for cell types of interest (rows), using normalized gene expressions for all 1-to-1 orthologous genes between the mouse models (WD-DEN, WD-CCl$_4$) and humans with MASH-HCC. **r**, Individual sample pairwise mouse to human correlation matrix. **s-t**, Mean *Acly* mRNA expression by cell types in livers from WD-DEN mice (**s**) and human MASH-HCC (**t**).

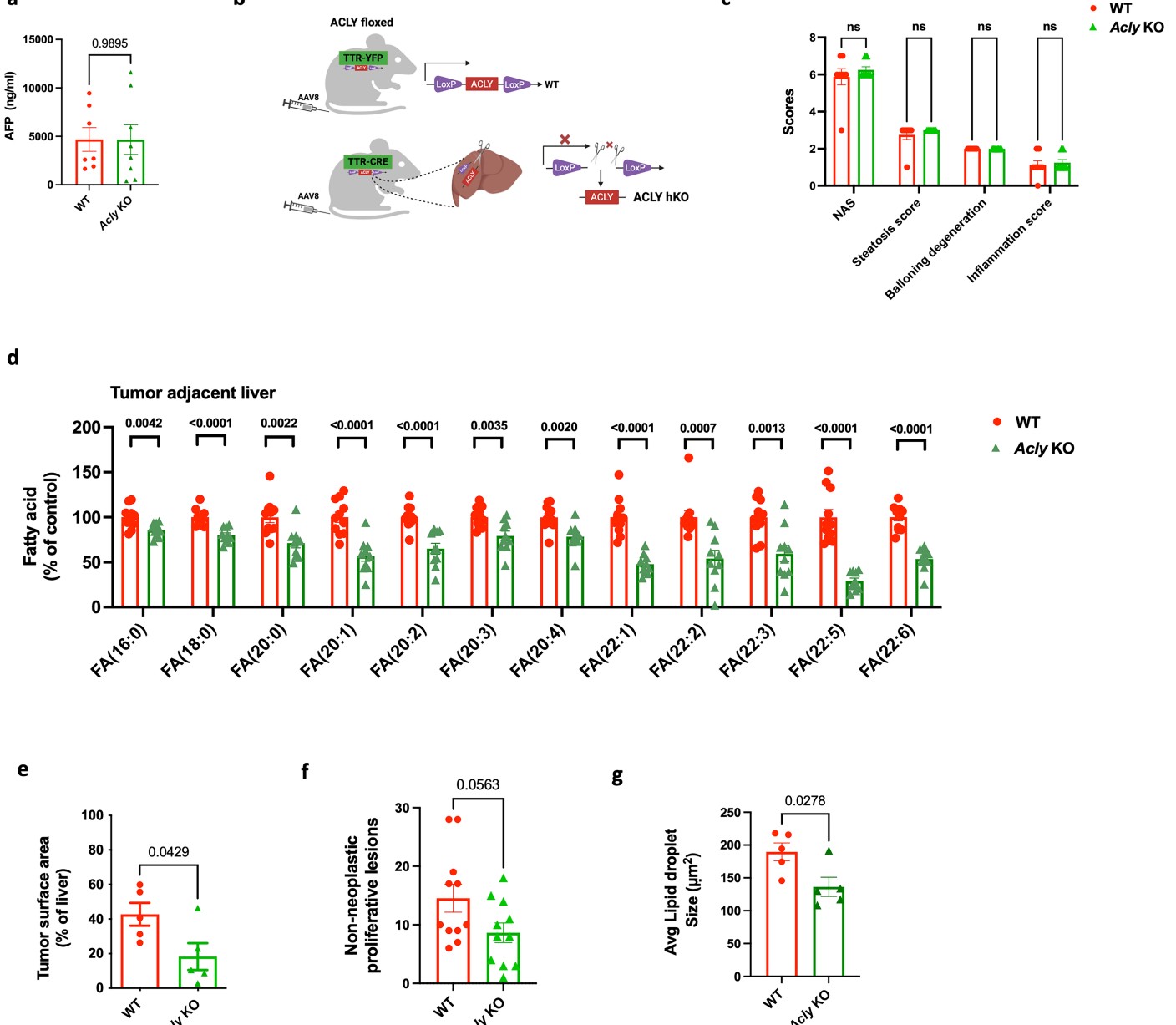

**Extended Data Fig. 2 | Genetic deletion of ACLY does not affect hepatic ballooning, liver steatosis, lobular inflammation, and NAFLD activity score (NAS) but reduces tumor burden. a**, Baseline serum AFP levels prior to AAV injection in WD-DEN treated mice. Data are presented as mean ± SEM. WT (n = 8) and *Acly* KO (n = 8) mice and statistical comparison was analyzed by two-tailed unpaired t-test. **b**, Representative diagram of genetic deletion of ACLY using hepatocyte-specific AAV. The diagram was created using BioRender (https://biorender.com). **c**, Pathological scoring of livers for steatosis, ballooning, inflammation, and NAFLD activity in WT (n = 8) and *Acly* KO (n = 8) mice. Data are presented as mean ± SEM. Statistical comparison for histological scores was analyzed by an unpaired Mann-Whitney two-tailed test. **d**, Fatty acid levels in tumor adjacent liver sections from WT and *Acly* KO mice (FA, Fatty acid; C:D, the

total number of carbon atoms to the number of carbon-carbon double bonds). Data are presented as mean ± SEM, *Acly* KO (n = 10) vs WT (n = 11) mice by unpaired two-tailed t-test. Significant differences observed for FA(18:0): $P = 3.7 \times 10^{-5}$, FA(20:1): $P = 5.09 \times 10^{-5}$, FA(20:2): $P = 8.35 \times 10^{-5}$, FA(22:1): $P = 9.46 \times 10^{-7}$, FA(22:5): $P = 4.39 \times 10^{-7}$ and FA(22:6): $P = 1.08 \times 10^{-7}$. **e**, Percentage of total liver area covered by tumors. Data are presented as mean ± SEM. WT (n = 5) and *Acly* KO (n = 5) mice analyzed by unpaired two-tailed t-test. **f**, Non-neoplastic proliferative lesions in livers from WT (n = 11) and *Acly* KO (n = 11) mice. Data are presented as mean ± SEM. P-values by unpaired two-tailed t-test. **g**, Average lipid droplet size in liver tumors from WT and *Acly* KO mice. Data are presented as mean ± SEM, n = 5/group. Statistical significance within each genotype condition was analyzed by an unpaired two-tailed t-test.

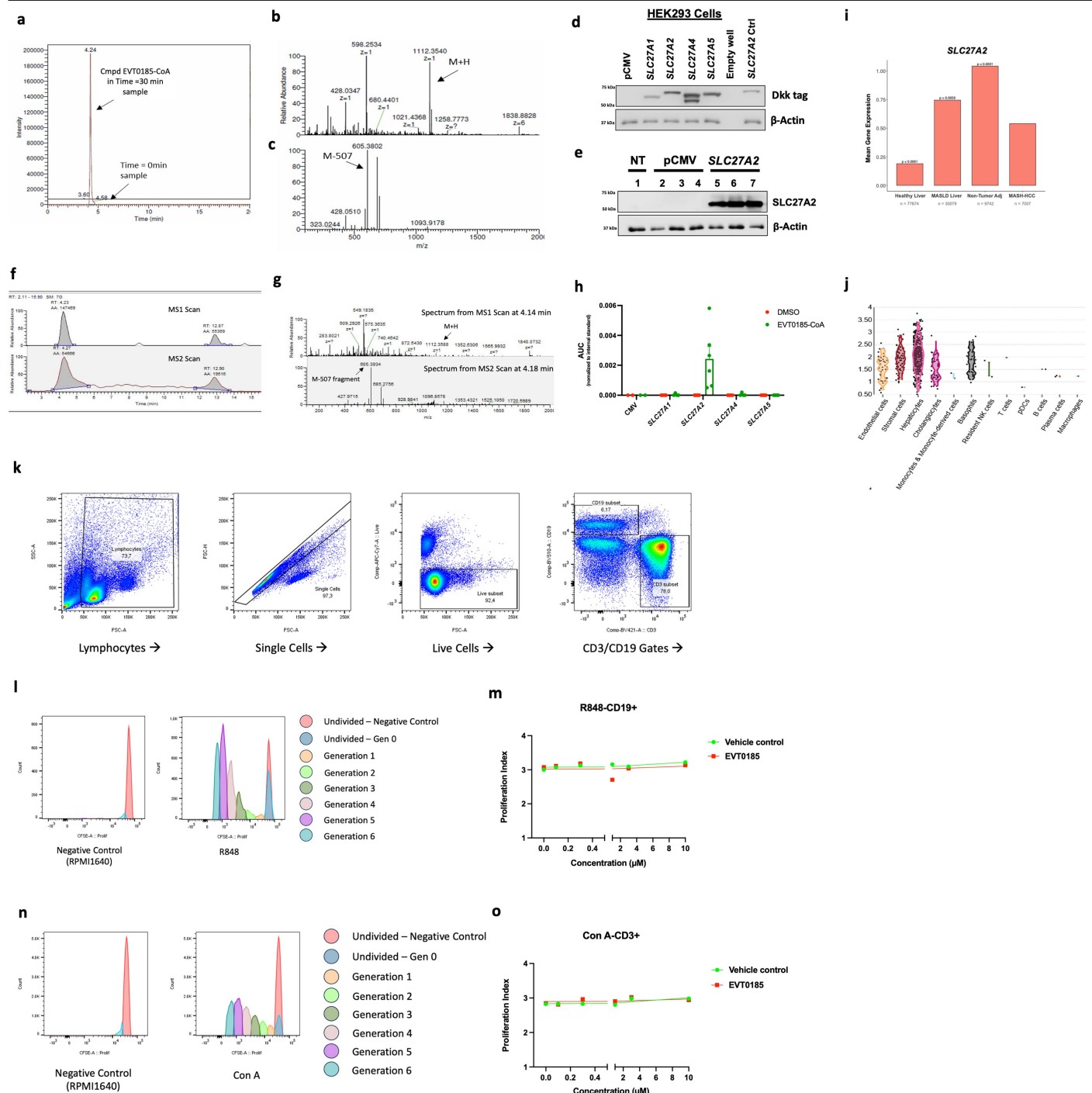

**Extended Data Fig. 3 | HEK-293 cells expressing SLC27A2 generated the CoA thioester. a**, Overlay of the extracted ion chromatograms for EVT0185 reaction mixture at 0 and 30 min. **b**, Mass spectrum of EVT0185-CoA at 30-min. **c**, Fragment ions of EVT0185-CoA at 30-min sample showing the fragment ion 605 m/z produced following the characteristic neutral loss of 3'-phosphonucleoside diphosphate (507 Da). Western blot showing **d**, Dkk tag expression in empty vector control (pCMV), *SLC27A1*, *SLC27A2*, *SLC27A4*, and *SLC27A5* transfected HEK293 cells **e**, SLC27A2 (ACSVL1) protein expression in non-transfected cells (lane 1), pCMV controls (lanes 2–4) and SLC27A2 (lanes 5–7) transfected cells; β-actin as a loading control. For gel source data, see Supplementary Fig. 1. **f**, Extracted ion chromatogram from the MS1 and MS2 scans of HEK293 cells transfected with SLC27A2 and treated with 30 μM EVT0185. **g**, Mass spectrum from the MS1 and MS2 scans which shows the parent ion (1112 m/z) and the neutral loss of 3'-phosphonucleoside diphosphate (507 Da),

respectively. **h**, EVT0185-CoA detected only in extracts from HEK293 cells overexpressing SLC27A2 (ACSVL1); mean ± SEM, n = 6 biologically independent samples/group. **i**, Proportion of human liver cells expressing SLC27A2 mRNA at different stages of disease progression. **j**, Human SLC27A2 protein expression in different cell types. **k**, Flow cytometry gating strategy: lymphocyte population identified via FSC vs. SSC, singlets via FSC-A vs. FSC-H, live cells via FSC vs. e780 viability dye, and CD19+ B cells and CD3+ T cells via CD19 vs. CD3 plot. **l**, PBMCs stimulated with R848 and IL-2 showing CD19+ B cell proliferation compared to negative control. **m**, Average proliferation index of R848-induced CD19+ B cells with EVT0185 or vehicle, n = 2 biologically independent PBMC donors. **n**, PBMCs stimulated with Concanavalin A (Con A) and IL-2 showing CD3+ T cell proliferation compared to negative control. **o**, Average proliferation index of Con A-induced CD3+ T cells with EVT0185 or vehicle, n = 2 biologically independent PBMC donors.

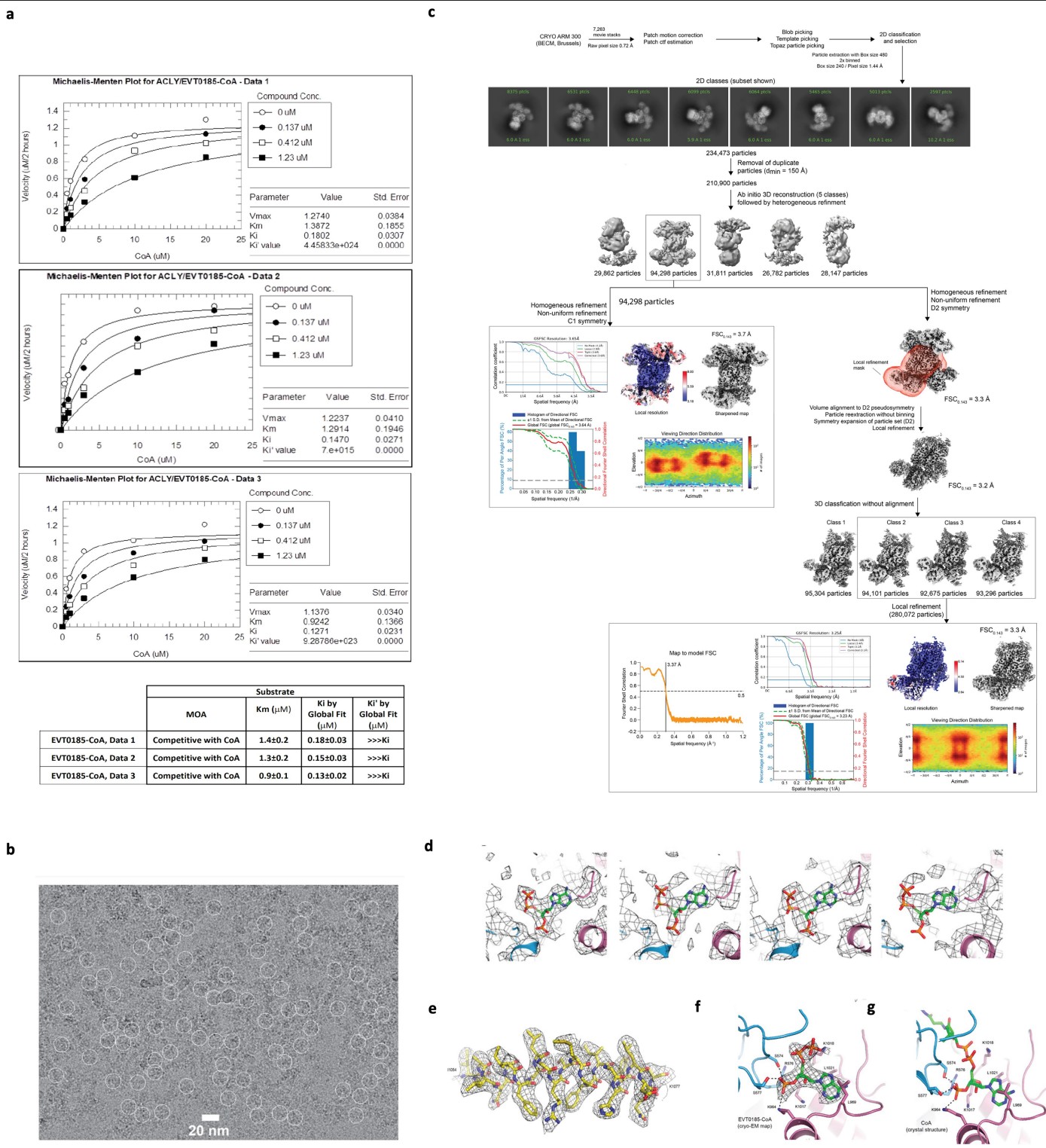

**Extended Data Fig. 4 | Competitive assay and cryo-EM analysis of the ACLY:(R, S)-EVT0185-CoA complex. a**, Global fit of Michaelis-Menten Plots by GraFit software for EVT0185-CoA effects on ACLY activity. **b**, Motion-corrected micrograph with picked particles encircled. The bottom scale bar is 20 nm. **c**, Cryo-EM data processing workflow in cryoSPARC for the ACLY:(R,S)-EVT0185-CoA complex. **d**, Sharpened cryo-EM map for ACLY:(R, S)-EVT0185-CoA complex following refinement without symmetry applied illustrating the ligand density at the four CoA-binding binding pockets of ACLY. The density is overlaid with

the final real-space refined model molecular model for adenosine 3'-phosphate 5'-diphosphate (shown in Fig. 2i) based on the structural superposition of the CSH domains. **e**, Segment of the sharpened cryo-EM map following symmetry expansion, 3D classification, and local refinement and carved around residues ACLY residues 1054–1077 (carve radius = 2 Å). **f-g**, Comparison between the binding modes of (R, S)-EVT0185-CoA (panel f) and CoA bound to human ACLY in pdb 6hxh (panel g).

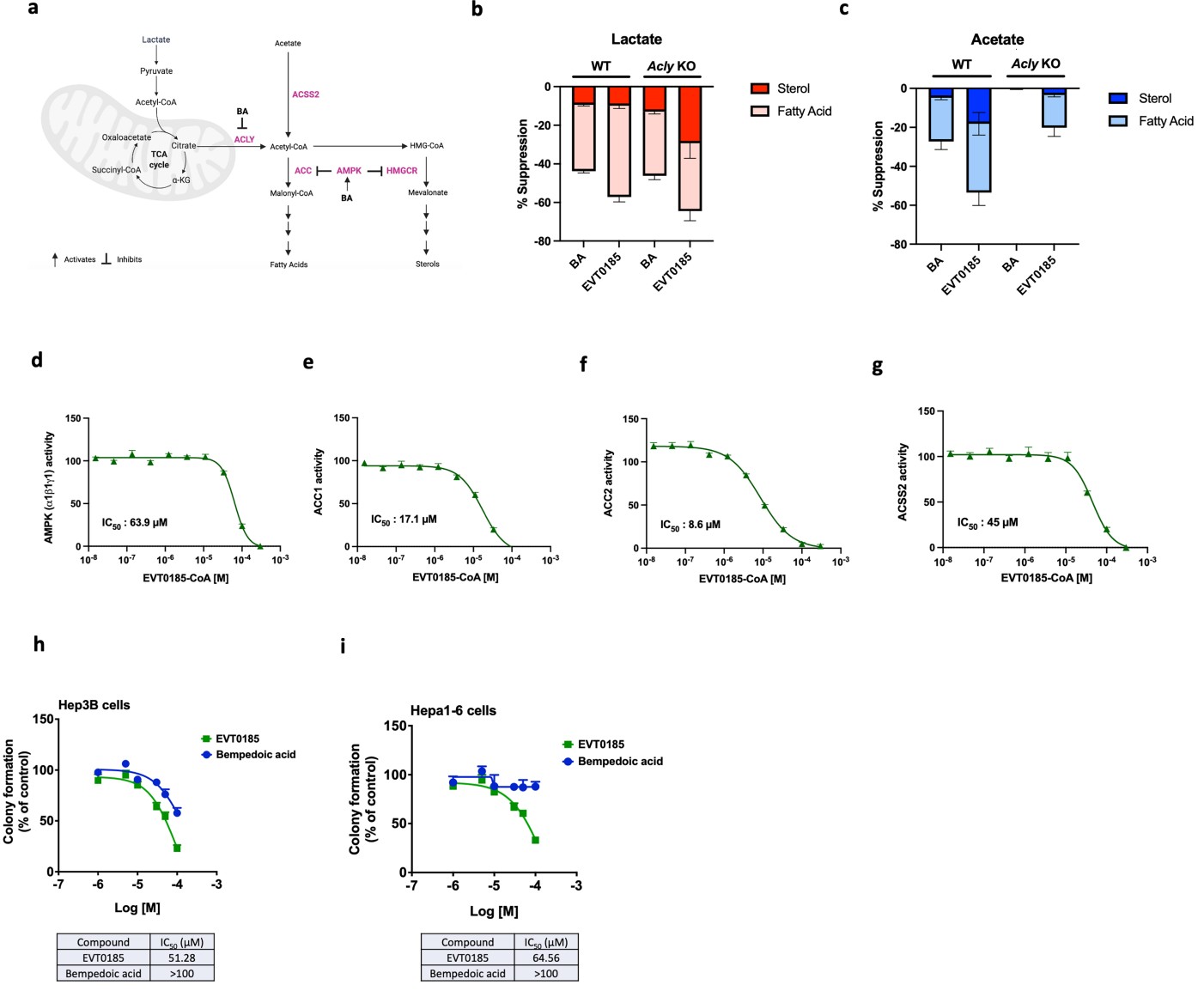

**Extended Data Fig. 5 | EVT0185 inhibits key metabolic enzymes involved in the regulation of sterol and fatty acid synthesis. a**, Graphical representation showing metabolic enzymes in the regulation of sterol and fatty acid synthesis. The graphical representation was created using BioRender (https://biorender.com). (**b-c**), Effect of BA (Bempedoic acid) and EVT0185 on **b**, lactate and **c**, acetate incorporation into fatty acids and sterols in WT and *Acly* KO mouse hepatocytes. Each stacked bar represents the mean ± SEM, % suppression with BA or EVT0185 in WT (n = 3) and *Acly* KO (n = 3) mice. (**d-g**) Inhibitory effect of

EVT0185-CoA on the activity of **d**, AMPK; **e**, ACC1; **f**, ACC2; and **g**, ACSS2. Each line graph represents the mean ± SEM, enzyme activity inhibition by EVT0185-CoA (n = 3 samples/group). (**h** and **i**) Effect of Bempedoic acid and EVT0185 on clonogenic survival/colony formation in **h**, Hep3B human, and **i**, Hepa1-6 mouse HCC cell lines. Each line graph represents the mean ± SEM colony formation in cells treated with EVT0185 or Bempedoic acid (n = 4 biologically independent samples/treatment groups).

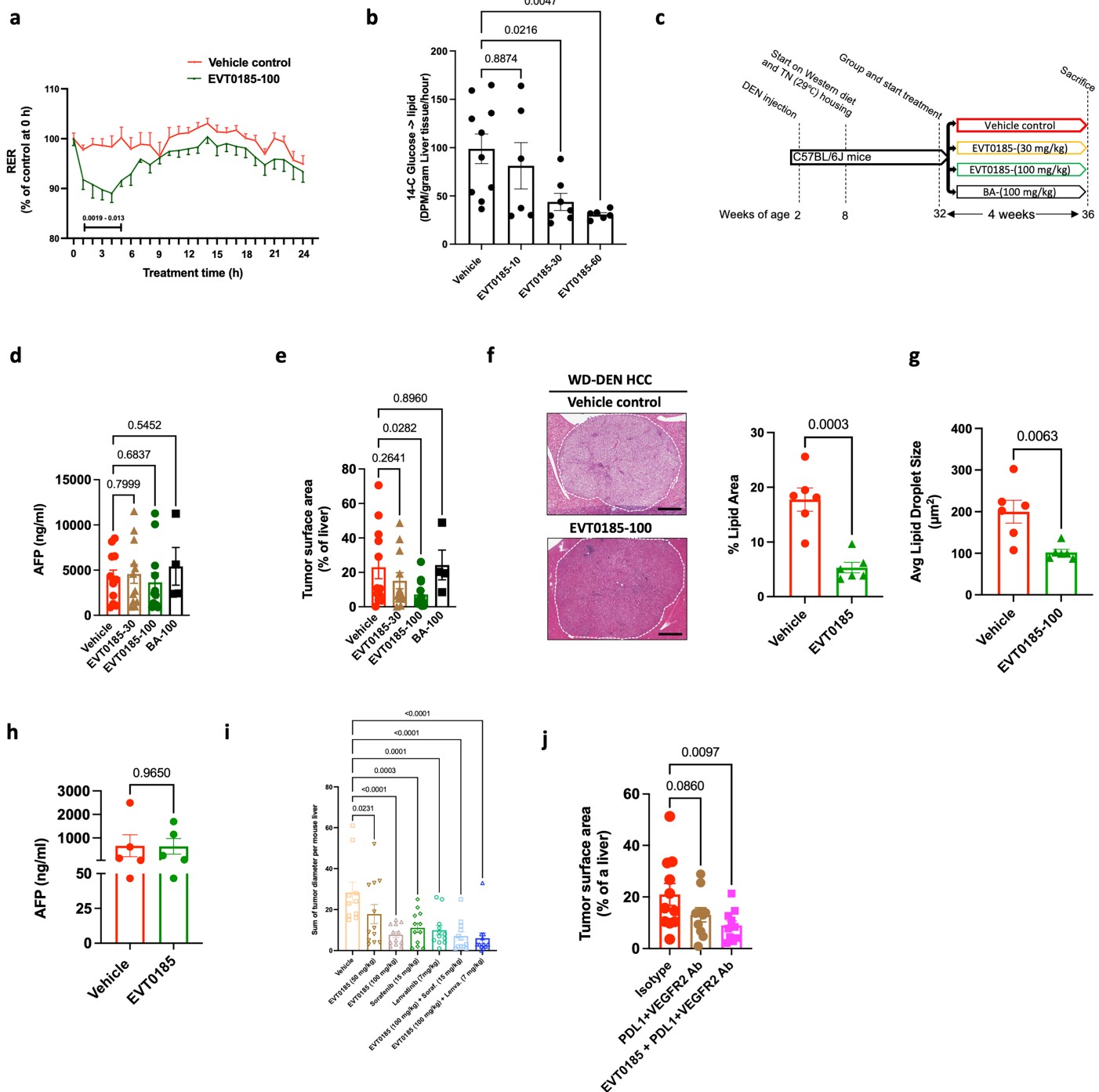

**Extended Data Fig. 6 | EVT0185 decreases RER, [14]C-glucose incorporation into lipid and tumor burden in mouse liver. a**, Effect of a single dose of EVT0185 on RER (Respiratory Exchange Ratio) in C57BL-6 mice. Each line graph represents the mean ± SEM, P values by unpaired two-tailed t-test, EVT0185 (n = 8) vs. vehicle-treated (n = 9) mice. **b**, [14]C-glucose incorporation into fatty acids and cholesterol in the liver isolated from C57BL-6 mice treated with EVT0185 for 7 days. Each bar represents the mean ± SEM, Vehicle (n = 10), EVT0185-10 (n = 6), 30 (n = 7), and 60 (n = 6) mg/kg-treated mice, one-way ANOVA followed by Tukey's multiple comparisons. **c**, Experimental Scheme of WD-DEN HCC model. **d**, Serum-AFP levels before starting treatment, and **e**, tumor surface area after treatments. Data as mean ± SEM, Vehicle (n = 12), EVT0185-30 and 100 mg/kg (n = 12) and Bempedoic acid (n = 4)-treated mice; one-way ANOVA with Fisher's LSD. (**f-g**) **f**, Representative images showing tumor lipid droplets (Scale bars are 600 μm) and bar diagram showing

percentage area of a lipid and **g**, average lipid droplet size in tumor, each bar represents the mean ± SEM, n = 6 mice/group, P values by unpaired two-tailed t-test. **h**, WD-fed mice with elevations in AFP at 18 months before starting treatment. Each bar represents the mean ± SEM, n = 5 mice/group, P values by unpaired two-tailed t-test. **i**, Sum of tumor diameter per mouse liver. Each bar represents the mean ± SEM, Treatment groups (n = 12) vs. vehicle-treated mice (n = 10), P values by one-way ANOVA followed by Fisher's LSD multiple comparisons: $P = 1.99 \times 10^{-5}$ (EVT0185-100 vs. vehicle), $P = 1.14 \times 10^{-5}$ (EVT0185+Soraf vs. vehicle) and $P = 4.57 \times 10^{-6}$ (EVT0185+Lenva vs. vehicle). **j**, Tumor surface area. Each bar represents the mean ± SEM, Isotype control (n = 11), PDL1 + VEGFR2 Ab (n = 10), and EVT0185 + PDL1 + VEGFR2Ab (n = 11) injected mice, P values by one-way ANOVA followed by Fisher's LSD multiple comparisons.

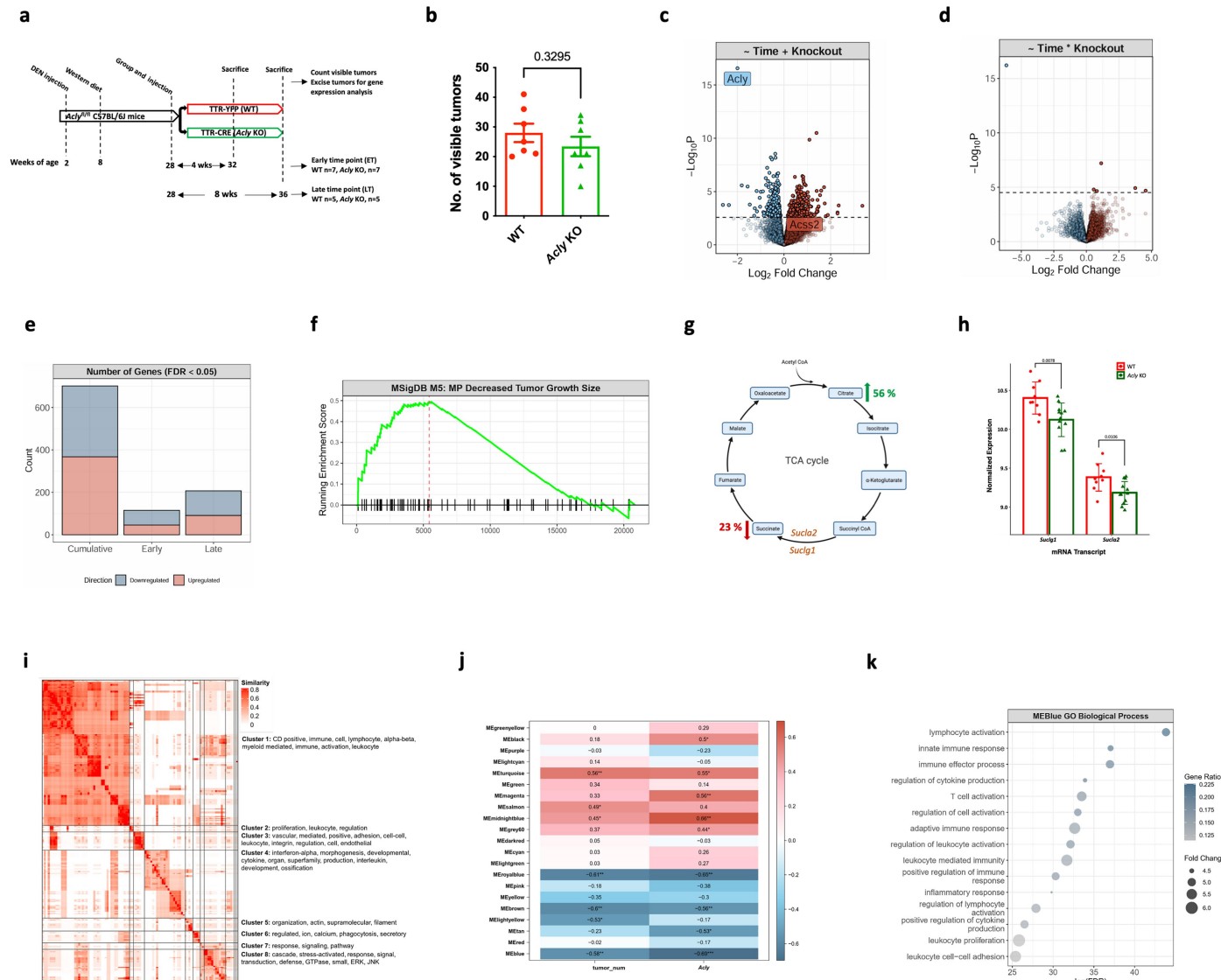

**Extended Data Fig. 7 | Transcriptomic profiling of *Acly* KO tumors vs WT tumors. a**, Experimental scheme. **b**, No. of visible tumors on the liver surface in mice 4 weeks after adenovirus injection. Data are presented as mean ± SEM, WT (n = 7) vs. *Acly* KO (n = 7). Statistical analysis was performed using a two-tailed unpaired t-test. **c**, Differential expression analysis of *Acly* KO using an additive model adjusting for timepoint. Significance was determined by Wald test with a false discovery adjusted threshold of 5% as implemented in DESeq2. **d**, Modifying effect of time on *Acly* KO using an interaction model. Significance of interaction was determined by Wald test with a false discovery adjusted threshold of 5% as implemented in DESeq2. **e**, Comparison between the number of significant genes in each module negatively associated with both *Acly* expression and

surface tumor burden. **f**, GSEA of decreased tumor growth size. **g**, TCA cycle showing a percentage change in tumor citrate and succinate levels in *Acly* KO mice compared to WT mice. The illustration of the TCA cycle was created using BioRender (https://biorender.com). **h**, Tumor *Suclg1* and *Sucla2* mRNA expression. Each bar represents the mean ± SEM, *Acly* KO (n = 12) vs WT (n = 9) mice, P values by two-tailed unpaired t-test. **i**, Hierarchical clustering of upregulated biological processes based on semantic similarity. **j**, Identification of co-expression modules significantly associated with *Acly* expression and surface tumor burden. **k**, Gene ontology analysis of co-expressed genes within module blue, *Acly* KO (n = 12) vs WT (n = 9) mice.

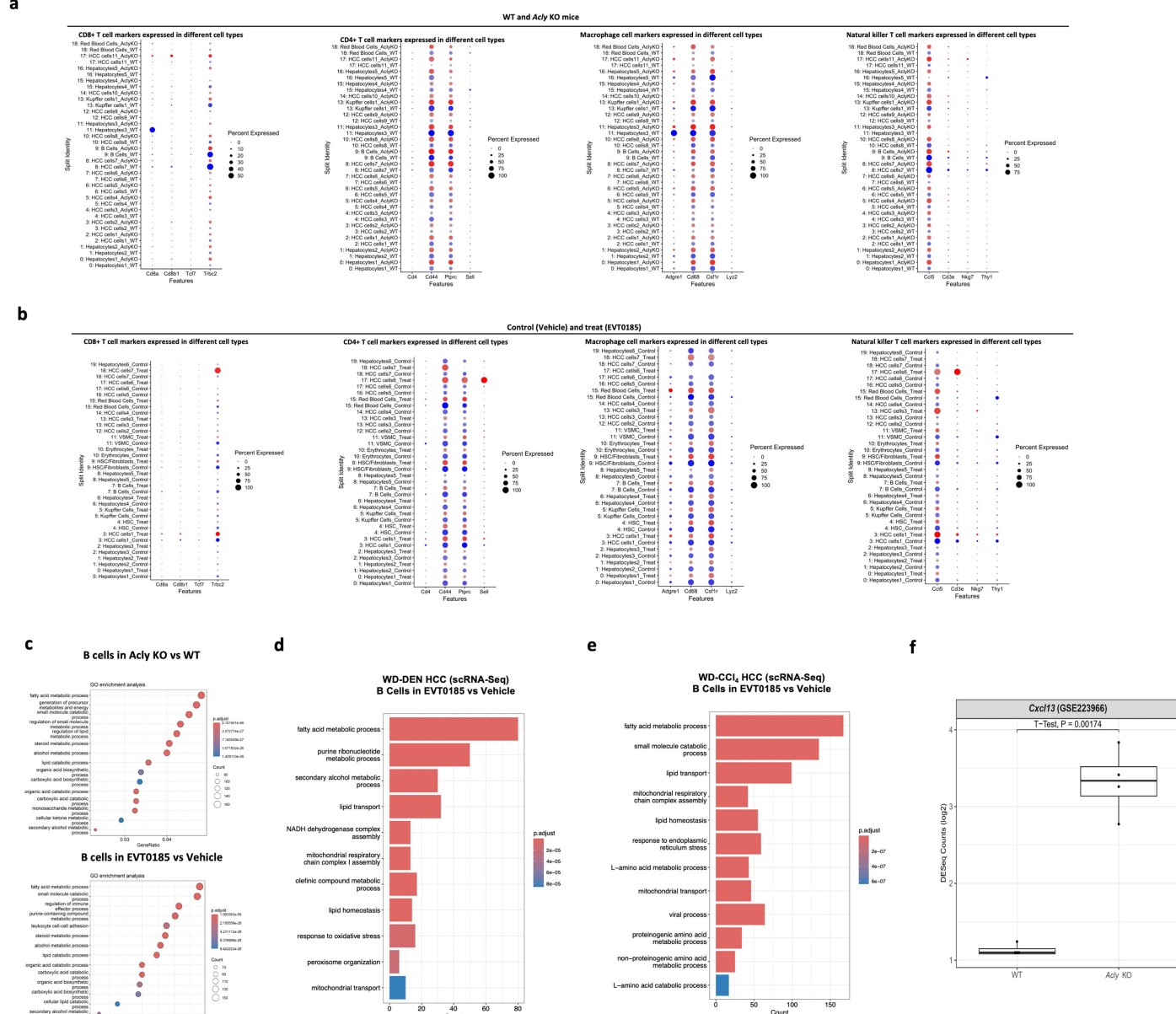

**Extended Data Fig. 8 | Genetic inhibition of ACLY or treatment with EVT0185 in MASH-driven HCC selectively enhances tumor-infiltrating B cell populations with minimal impact on other immune cells. a-b,** Immune cell markers expressed in different cell types in **a**, WT and *Acly* KO and **b**, Vehicle and EVT0185-treated mice. **c**, Top upregulated pathways in B cells in *Acly* KO or EVT0185-treated mice (spatial transcriptomics analysis). Statistical analysis was performed using Fisher's Exact test. **d** and **e**, Single seq analysis of **d**, WD-DEN and **e**, WD-CCl₄ mouse livers showing top upregulated pathways in B cells.

Statistical analysis was performed using a one-sided hypergeometric test (enrichGO); p-values adjusted by Benjamini-Hochberg. **f**, *Cxcl13* mRNA expression analyzed from publicly available RNA-seq dataset in WT and *Acly* KO DEN tumors cultured in vitro (GSE223966)[10]. Boxplot lines represent the first quartile, median, and third quartile. Whiskers connect the minimum and maximum values. Significance was ascertained by an unpaired two-tailed t-test between *Acly* KO vs WT (n = 4 hepatocellular carcinoma cell lines derived from DEN-induced tumors in *Acly^(f/f)* mice).

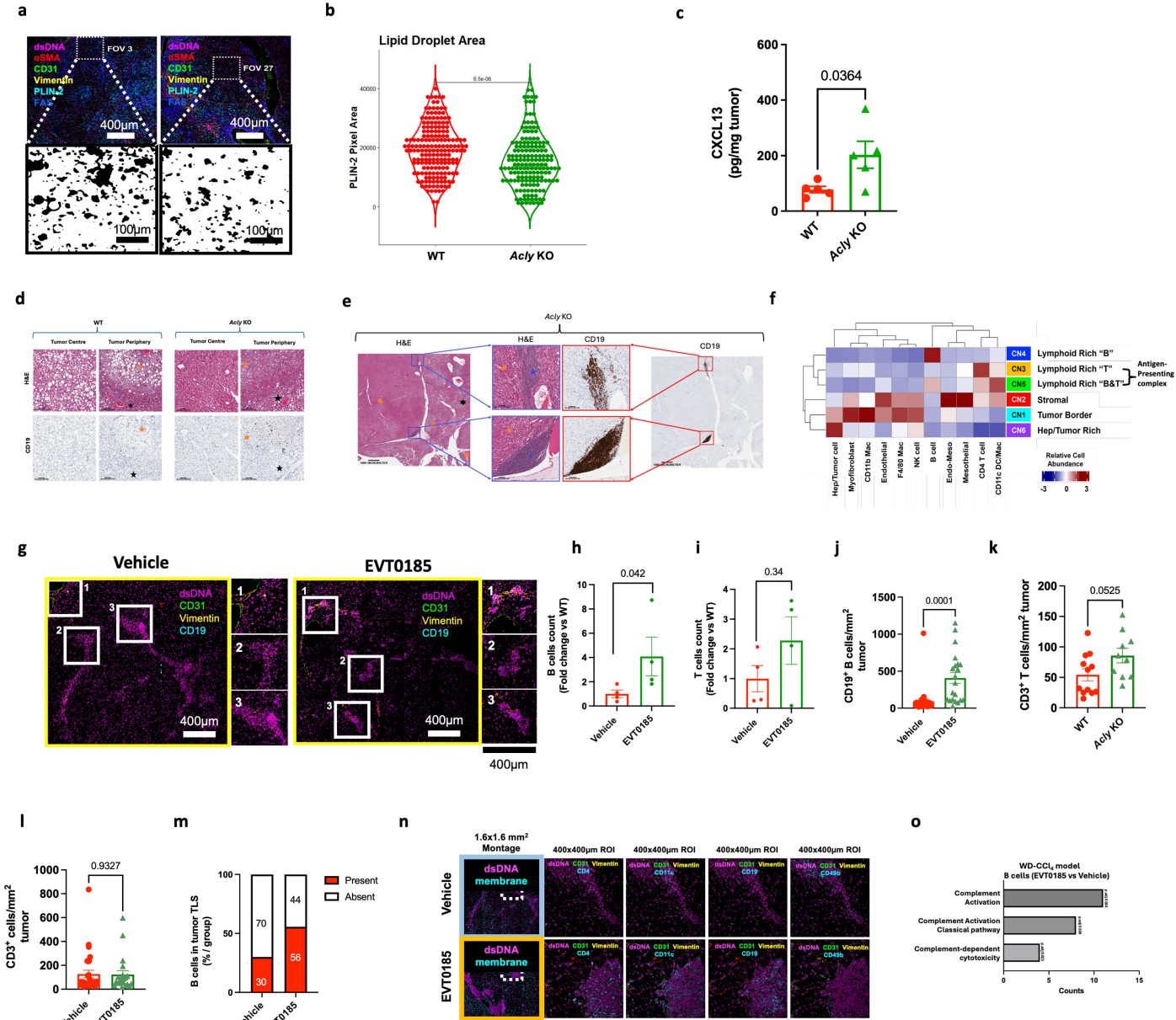

**Extended Data Fig. 9 | B cells in the tumor periphery were in close proximity to cells involved in antigen presentation. a** and **b**, Quantification of lipid droplet area was extrapolated by converting the PLIN-2 signal into a mask and quantifying the area covered by this mask per 400 × 400 micron field of view (FOV). These calculations were performed in ImageJ using custom scripts. Each dot represents a single FOV, Wilcoxon non-parametric unpaired two-sided test. *Acly* KO (n = 188) vs WT (n = 170) FOVs from n = 4 mice/group. **c**, CXCL13 protein level in tumors from WT and *Acly* KO mice; mean ± SEM, n = 5 mice/group; unpaired two-tailed t-test. **d**, Representative images showing B cells at tumor centre and periphery (Orange star: Tumor front (border-periphery), Black star: Surrounding non-tumoral hepatic tissue). **e**, B cell aggregations in *Acly* KO tumor (Orange star: Tumor, Black star: Non-tumoral liver tissue, Blue star: B Cell aggregations). **f**, Heatmap of cell neighborhoods hierarchically sorted by cell phenotype and showing relative cell abundances. (**g-j, l-n**) WD-DEN mice treated with Vehicle or EVT0185. **g**, Representative MIBI images of liver tumors. **h**, B cell, and **i**, T cell counts from regions representing the tumor lesion interface with the liver; mean ± SEM, n = 4 mice/group; unpaired Wilcoxon non-parametric unpaired two-sided test. **j**, CD19+ positive cells count/mm² tumor area; mean ± SEM, Vehicle (n = 29) and EVT0185 (n = 21) lesions; unpaired two-tailed t-test. **k**, CD3+ positive cells count/mm² tumor area; mean ± SEM, WT (n = 12) and *Acly* KO (n = 10) lesions; unpaired two-tailed t-test. **l**, CD3+ positive cells count/mm² tumor area; mean ± SEM, Vehicle (n = 29) and EVT0185 (n = 21) lesions; unpaired two-tailed t-test. **m**, Percentage of mice with/without B cells in TLS. **n**, TLS in EVT-treated mouse compared with vehicle control. **o**, GO enrichment analysis showing upregulated pathways related to complement activation in B cells in the EVT0185-treated group relative to Vehicle. Statistical analysis was performed using one-sided hypergeometric test (enrichGO); p-values adjusted by Benjamini-Hochberg.

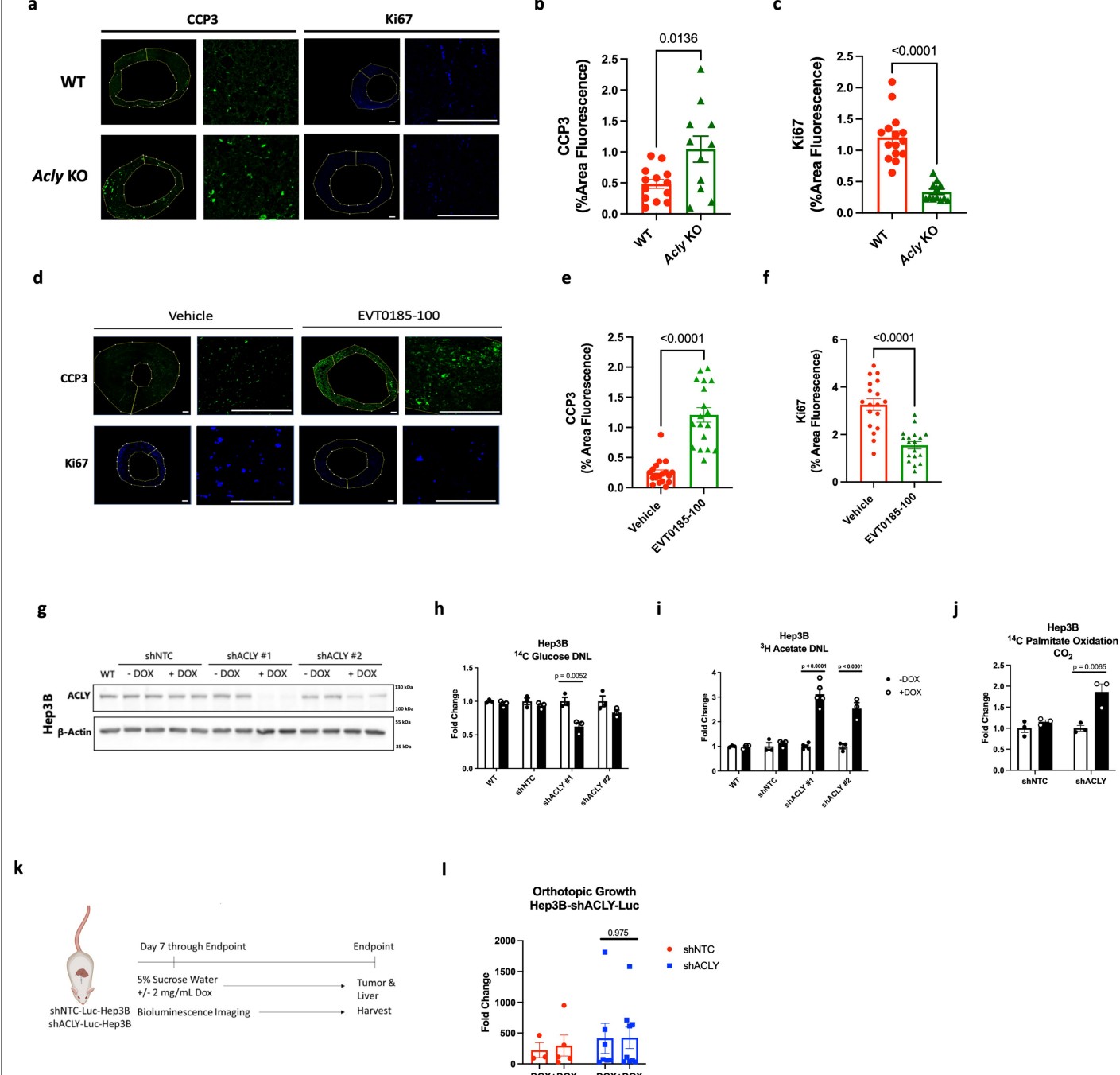

**Extended Data Fig. 10 | Reduction in tumor burden due to ACLY inhibition is mediated by the induction of an immunogenic response.**
**a-c, a,** Representative images with percentage area fluorescence level of **b**, CCP3; mean ± SEM, *Acly* KO (n = 11) vs WT (n = 13) tumors by unpaired two-tailed t-test, and **c**, Ki67 protein expression in tumor leading edge; mean ± SEM, *Acly* KO (n = 12) vs WT (n = 15) tumors by unpaired two-tailed t-test: $P = 8.74 \times 10^{-8}$. **d-f, d,** Representative images and percentage area fluorescence level of **e**, CCP3 ($P = 1.34 \times 10^{-8}$) and **f**, Ki67 ($P = 1.27 \times 10^{-6}$) protein expression in the tumor leading edge. Data are presented as mean ± SEM. EVT0185 vs Vehicle control by unpaired two-tailed t-test, (n = 18 tumors per group). **g,** DOX inducible shRNA suppresses ACLY expression and phosphorylation in Hep3B cells. For gel source data, see Supplementary Fig. 1. (**h-j**) Inducible knockdown led to **h**, reduced [14C] glucose-mediated DNL and **i**, increased [3H] acetate mediated DNL in Hep3B cells and **j**, increased fatty acid oxidation based on the sampling of [$^{14}$C] incorporation in gaseous carbon dioxide obtained from cell culture media. Data are presented as mean ± SEM. Number of cell culture replicates per condition were WT (n = 3), shNTC (n = 3), shACLY #1 (n = 3), and shACLY #2 (n = 3). Significance was determined by an unpaired one-tailed t-test. **k,** Outline of in vivo orthotopic experiment. The illustration of the mouse was created using BioRender (https://biorender.com). **l,** Bioluminescence at 6 weeks was not significantly different between control and ACLY-deficient tumor. Each bar represents the mean ± SEM. Number of mice per condition at the time of sacrifice were shNTC -Dox (n = 3), shNTC +Dox (n = 5), shACLY -Dox (n = 7), shACLY +Dox (n = 9). Significance was determined by unpaired one-tailed t-test.

**WT-Iso**

**WT-Anti CD20**

**KO-Iso**

**KO-Anti CD20**

Lymphocytes → Single cells → Live cells → CD45+ → B220+CD19+

**b**

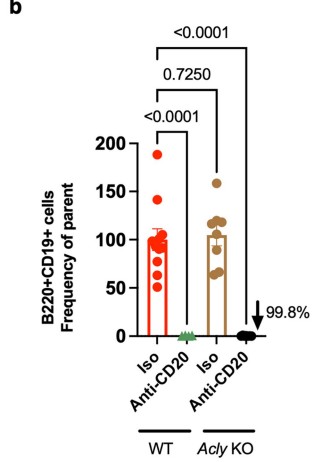

**Extended Data Fig. 11 | Confirmation of B cell depletion. a**, Gating strategy to identify B220⁺CD19⁺ cells in Isotype control, and Anti-CD20 injected WT and *Acly* KO mice. Briefly, cells were gated as follows: SSC vs FSC plot of lymphocytes population, FSC-H vs FSC-A to identify single cell population, FSC vs 7-AAD to identify live cells population, B220 vs CD19 to identify B220⁺CD19⁺ B cell population. **b**, B220⁺CD19⁺ cells in Iso (Isotype control) and Anti-CD20

(B cell-depleted)-injected WT or *Acly* KO mice. Data are presented as mean ± SEM, Significant reductions were observed in WT-Anti-CD20 (n = 4; P = 2.37 × 10⁻⁶) and in *Acly* KO-Anti-CD20 (n = 7; P = 9.76 × 10⁻⁸), compared to WT-Isotype control (n = 11)-injected mice. The *Acly* KO-Isotype control group included n = 8 mice. Statistical analysis was performed using one-way ANOVA followed by Fisher's LSD test.

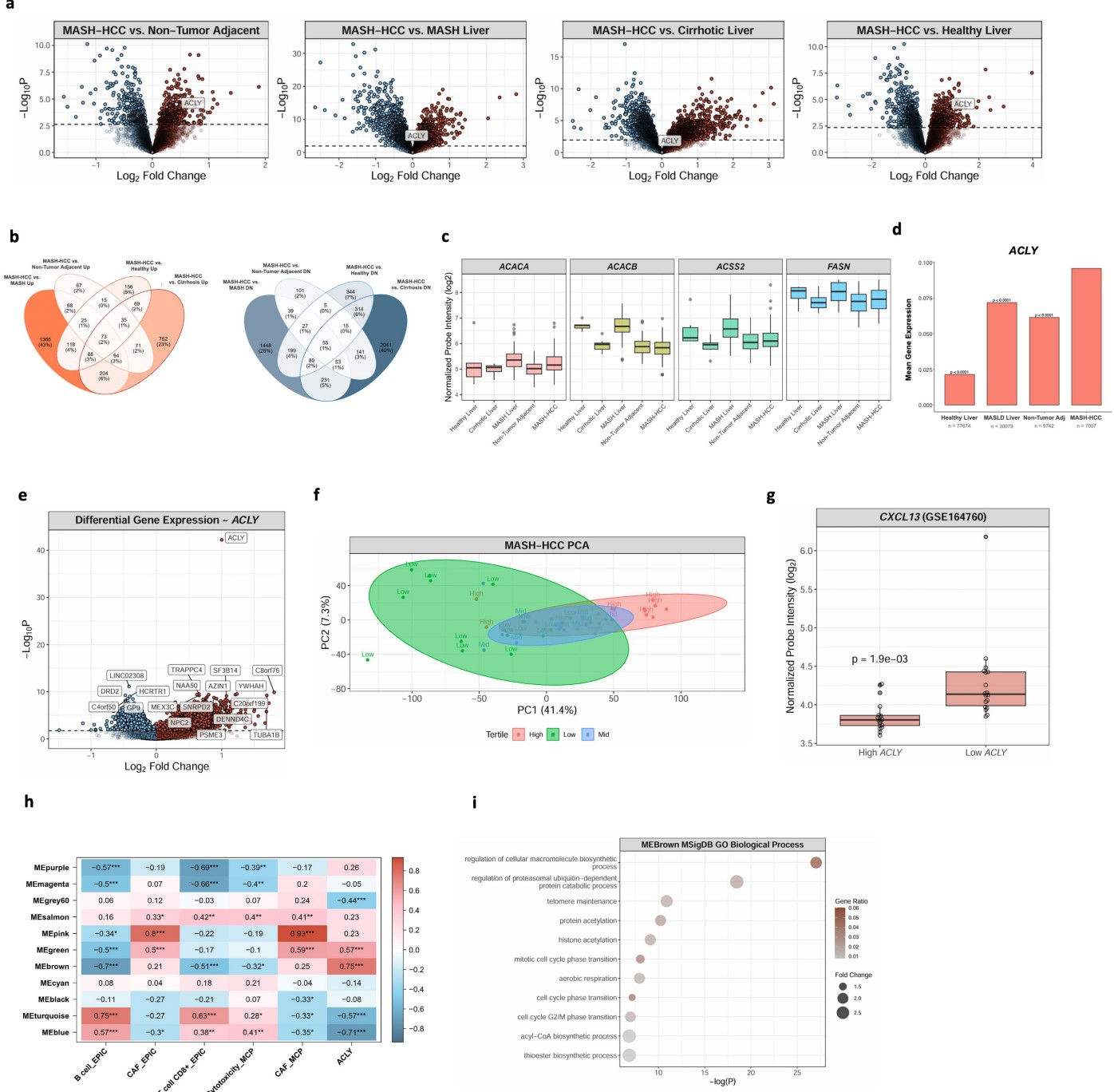

**Extended Data Fig. 12 | Differential expression analysis in human MASH-HCC. a**, Differential expression analysis comparing MASH-HCC (n = 53) and non-tumor adjacent liver (n = 29), MASH liver (n = 74), cirrhotic liver (n = 8), and healthy liver tissue (n = 6); two-tailed t-test; 5% FDR as implemented in limma. **b**, Overlap between significantly upregulated and downregulated genes in each pairwise comparison. **c**, Gene expression of lipogenic enzymes acetyl-CoA carboxylase (*ACACA* or *ACACB*), *ACSS2*, and *FASN* across disease states (MASH-HCC, n = 53; non-tumor adjacent, n = 29; MASH liver, n = 74; cirrhotic liver, n = 8; healthy liver, n = 6). Boxplot lines represent the first quartile, median, and third quartile. Whiskers connect the minimum and maximum values. **d**, *ACLY* upregulation in human MASH-HCC compared to all other tissue types using scRNAseq samples. Significance assessed using Wilcoxon rank-sum tests comparing each condition to MASH-HCC, with p-values adjusted using the Benjamini-Hochberg method. Exact p-adj values: vs healthy

$P_{adj} = 6.9 \times 10^{-247}$, vs MALSD $P_{adj} = 1.0 \times 10^{-16}$, vs non-tumor adjacent $P_{adj} = 1.6 \times 10^{-18}$. **e**, Differential expression analysis of genes associated with *ACLY* expression in MASH-HCC; two-tailed t-test; 5% FDR as implemented in limma. **f**, Principal component embeddings of genes differentially expressed with respect to ACLY expression validate stratification of MASH-HCC tissues by high, medium, and low ACLY expression levels. **g**, Upregulation of *CXCL13* among human MASH-HCC samples with reduced (bottom tertile, n = 18) relative to elevated (top tertile, n = 18) *ACLY* expression. Significance determined by two-tailed t-test with a false discovery adjusted threshold of 5% as implemented in limma. Boxplot lines represent the first quartile, median, and third quartile. Whiskers connect the minimum and maximum values. **h**, Identification of co-expression modules associated with *ACLY* expression and immune features. **i**, Significant biological processes associated with the immune and *ACLY* associated gene co-expression module.

# Reporting Summary

## Statistics

For all statistical analyses, confirm that the following items are present in the figure legend, table legend, main text, or Methods section.

| n/a | Confirmed | |
|---|---|---|
| ☐ | ☒ | The exact sample size (*n*) for each experimental group/condition, given as a discrete number and unit of measurement |
| ☐ | ☒ | A statement on whether measurements were taken from distinct samples or whether the same sample was measured repeatedly |
| ☐ | ☒ | The statistical test(s) used AND whether they are one- or two-sided *Only common tests should be described solely by name; describe more complex techniques in the Methods section.* |
| ☐ | ☒ | A description of all covariates tested |
| ☐ | ☒ | A description of any assumptions or corrections, such as tests of normality and adjustment for multiple comparisons |
| ☐ | ☒ | A full description of the statistical parameters including central tendency (e.g. means) or other basic estimates (e.g. regression coefficient) AND variation (e.g. standard deviation) or associated estimates of uncertainty (e.g. confidence intervals) |
| ☐ | ☒ | For null hypothesis testing, the test statistic (e.g. *F*, *t*, *r*) with confidence intervals, effect sizes, degrees of freedom and *P* value noted *Give P values as exact values whenever suitable.* |
| ☒ | ☐ | For Bayesian analysis, information on the choice of priors and Markov chain Monte Carlo settings |
| ☒ | ☐ | For hierarchical and complex designs, identification of the appropriate level for tests and full reporting of outcomes |
| ☐ | ☒ | Estimates of effect sizes (e.g. Cohen's *d*, Pearson's *r*), indicating how they were calculated |

*Our web collection on statistics for biologists contains articles on many of the points above.*

## Software and code

Policy information about availability of computer code

| Data collection | Flow cytometry data were collected using CytoFlex (Beckman Coulter Life Sciences) and BD LSR Fortessa (BD Biosciences). |
|---|---|
| | Immunofluorescence data were collected on an inverted confocal microscope (Leica Microscope Systems). |
| | MIBI : Spectral images of stained liver lesions were collected using an Ionpath MIBIscope with Multiplexed Ion-Beam Imaging technology. For Acly Ko cohort, thirty-five 400x400μm regions for each experimental group, representing 43,974 cell objects were acquired while for EVT0185 cohort, twenty-four 400x400μm regions for each experimental group were collected for a total of 30,191 cell objects. |
| | Bulk RNA seq : Next-generation sequencing was conducted at the McMaster Genomics Facility, Farncombe Institute, McMaster University, using Illumina HiSeq 1500 (Illumina; San Diego, CA, USA). Samples were randomly distributed across lanes of a HiSeq Rapid v2 flow cell to eliminate lane-specific effects and single-end 50 bp reads were generated at 12.5 million clusters per sample. Microarray CEL files were downloaded from GSE164760. Raw RNA-Seq files for tumor samples derived from DEN and CCl4 models from PRJNA488497 and PRJNA386995 respectively. |
| | ScRNA seq: Sequencing was performed using the Illumina Novaseq 6000 sequencing system (S4, 2x150, 2-2.5 billion reads, targeting 5000 cells per sample at a depth of 25000 read pairs per cell). |
| | Single nuclei sequencing: Sequencing of cells derived from Healthy, MASLD, MASH-HCC tumor and tumor adjacent tissue were obtained from GSE174748 and GSE189175 |
| | Spatial Transcriptomics: Slides with 5 μm of tissue sections were processed with deparaffinization, H&E staining, imaging, decrosslinking, |

hybridization, ligation, probe extension, pre-amplification and probe-based library construction to generate gene expression libraries for each tissue for following sequencing. Images were taken using the Nikon 90i Eclipse upright microscope. RNA-seq was performed using the Illumina NextSeq 2000 (P2 Flow cell, 2×50bp configuration) system.

Grids were screened using a JEOL 1400 Plus microscope equipped with a JEOL Ruby CCD camera at the VIB Bioimaging Core Ghent. Cryo-electron microscopy data were collected at the VIB-VUB facility for Bio Electron Cryogenic Microscopy (BECM, Brussels, Belgium)

| Data analysis | Flow cytometry: FlowJo v10.8.1 |

Immunofluorescence and lipid droplet analysis: Image J.

CD3 and CD19 positive cells counting: Halo software.

MIBI analysis: Segmentation mask : Mesmer, Single cell phenotyping:'FlowSOM' R package
            Cellular Neighborhood analyiss: ImcRtools package

RNA seq analysis: Sequence quality: FastQC , Removal of low-quality reads and adapter sequences: Cutadapt, Genome alignment: HISAT2, Quantification of reads: Feature Counts, Surrogate variable analysis: sva v3.44.0, Differential gene expression analysis: DESeq2 package v1.36.0, Differential expression analysis for microarray data: limma v3.52.3, Over-representation and GSEA : clusterProfiler v4.4.4, Semantic similarity and binary cut method: simplifyEnrichment v1.6.1, Murine cell-type deconvolution: transcript-per-million (TPM) normalized gene expression data and mMCP-counter v1.1.0., Human cell-type deconvolution: TIMER2.0 and ESTIMATE package v1.0.13.

ScRNA seq analysis: Demultiplexing-Cell Ranger, Integration analysis-Seurat v5,Mouse-human correlation analysis-biomaRt package, Human gene-specific mean expression data-Seurat v5 in R studio.

Single nuclei sequencing: Processed using the Seurat package version 5.1.0 in R, Cell type identification-sc-type package in R.

Spatial Transcriptomics: Sequence data from FFPE tissues- Space Ranger count pipeline from 10X genomics, Quality control, normalization, dimensional reduction and clustering, variable gene selection, spatially-variable feature detection, annotation, differential expression, integration with multiple samples- Seurat, Spatial Transcriptomics data analyses- Linux system, R, RStudio software and Python programming language

Single-particle cryo-electron microscopy data analysis- CryoSparc v3.1.0. Neural-network based particle picking- TOPAZ. The atomic model for human ACLY (pdb 6hxh)- fitted ChimeraX and Coot and real-space refined in Phenix. Structural analysis-Pymol. Restraints for EVT0185-CoA -de Grade Web Server ( https://grade.globalphasing.org).

For manuscripts utilizing custom algorithms or software that are central to the research but not yet described in published literature, software must be made available to editors and reviewers. We strongly encourage code deposition in a community repository (e.g. GitHub). See the Nature Portfolio guidelines for submitting code & software for further information.

# Data

Policy information about availability of data

All manuscripts must include a data availability statement. This statement should provide the following information, where applicable:
- Accession codes, unique identifiers, or web links for publicly available datasets
- A description of any restrictions on data availability
- For clinical datasets or third party data, please ensure that the statement adheres to our policy

Data that support the findings of this study are available within the article and its Supplementary information. The bulk RNA-seq data of the WD-DEN liver tumors, and spatial transcriptomic data of the WD-DEN and WD-CCl4 livers have been deposited at NCBI Gene Expression Omnibus (GEO) and are accessible under accession numbers GSE296668 and GSE297081, respectively. For the mouse HCC model comparision, raw RNA-Seq files for tumor samples derived from Control-DEN and WD-CCl4 models were downloaded from the NCBI Sequence Read Archive under reference numbers PRJNA488497 and PRJNA386995 respectively. Single-nuclei sequencing of cells derived from Healthy, MASLD, MASH-HCC tumor and tumor adjacent tissue were obtained from GSE174748 and GSE189175. Gel source data are provided in Supplementary Figure 1. Source data are provided with this paper.

Cryo-EM maps following global and local refinement and the real-space refined model for the CCS/CSH assembly have been deposited in the Electron Microscopy Data Bank (EMDB) and Protein Data Bank with accession codes EMDB 17148 and PDB 8OS4. Detail info is in Supplementary Table S2.

# Research involving human participants, their data, or biological material

Policy information about studies with human participants or human data. See also policy information about sex, gender (identity/presentation), and sexual orientation and race, ethnicity and racism.

| Reporting on sex and gender | Not Applicable |

| Reporting on race, ethnicity, or other socially relevant groupings | Not Applicable |

| Population characteristics | Not Applicable |

| | |
|---|---|
| Recruitment | Not Applicable |
| Ethics oversight | Not Applicable |

Note that full information on the approval of the study protocol must also be provided in the manuscript.

# Field-specific reporting

Please select the one below that is the best fit for your research. If you are not sure, read the appropriate sections before making your selection.

☒ Life sciences    ☐ Behavioural & social sciences    ☐ Ecological, evolutionary & environmental sciences

For a reference copy of the document with all sections, see nature.com/documents/nr-reporting-summary-flat.pdf

# Life sciences study design

All studies must disclose on these points even when the disclosure is negative.

| | |
|---|---|
| Sample size | Sample size was calculated based on the animal availability after around 7 months of DEN injection. The sample size were considered as adequate based on the previous experiments which would be sufficient to provide meaningful conclusions (Cell metabolism. 2019 Jan 8;29(1)). No statistical method was used to predetermine sample size. |
| Data exclusions | No data were excluded during the analysis |
| Replication | Acly KO Animal experiments were performed three times and replication attempt was successful. |
| Randomization | Mice were randomized based on their serum AFP (Alpha Feto protein) levels prior to the treatment/ AAV injection |
| Blinding | Animal experiments were not blinded. However, tissue collection, histological, biochemical analysis and other analytic tests were blinded. |

# Reporting for specific materials, systems and methods

We require information from authors about some types of materials, experimental systems and methods used in many studies. Here, indicate whether each material, system or method listed is relevant to your study. If you are not sure if a list item applies to your research, read the appropriate section before selecting a response.

## Materials & experimental systems

| n/a | Involved in the study |
|---|---|
| ☐ | ☒ Antibodies |
| ☐ | ☒ Eukaryotic cell lines |
| ☒ | ☐ Palaeontology and archaeology |
| ☐ | ☒ Animals and other organisms |
| ☒ | ☐ Clinical data |
| ☒ | ☐ Dual use research of concern |
| ☒ | ☐ Plants |

## Methods

| n/a | Involved in the study |
|---|---|
| ☒ | ☐ ChIP-seq |
| ☐ | ☒ Flow cytometry |
| ☒ | ☐ MRI-based neuroimaging |

# Antibodies

| | |
|---|---|
| Antibodies used | Western Blot: SLC27A2 (Invitrogen, # PA5-30420), B-Actin (Cell Signaling Technology, #X5125S)<br><br>Immunofluorescense microscopy: CCP3 (Cell Signaling Technology, #9664S), Ki67 (ThermoFisher, #MA5-14520)<br><br>MIBI imaging: dsDNA nucleus DNA (IonPath, # 708901-100), CD19 (Invitrogen, #14019482), CD4 (Ionpath, #714304-100), CD11c ( IonPath, #714402-100), Arginase-1 (IonPath, # 715001-100), CD49b (IonPath, # 715102-100), CD31 e (IonPath, # 715202-100), Ki-67 (IonPath, # 715302-100), CD11b (IonPath, #715504-100), F4/80 (IonPath, # 715603-100), CD8 (IonPath, # 715803-100), CD3e (IonPath, # 715904-100), FAS (CST, #66058SF) , ACC (IonPath, #52923SF), Vimentin (IonPath, #716301-100), alphaSMA (IonPath, #716401-100), PLIN2 (Novus, # NB110-40877), B220 (IonPath, #716702-100), HNF4A (Invitrogen, MA1-199), CD45 (IonPath, # 715503-100), Na-K-ATPase membrane (IonPath, #717603-100).The antibodies are listed in Supplementary Table S3 as well.<br><br>Flow cytometry: Human recombinant IL-2 (Preprotech, #200-02), R848 (mAb tech, #36611), E780 Live/Dead stain (Thermofisher, #C34570), Anti-human CD3 (Biolegend, #300434), Anti-human CD19 (Biolegend, #302241), CD45.2 BV510 (BioLegend, #109838), B220 ( BD Biosciences, #563894), CD19 ( Biolegend, #152409) and 7AAD (Thermo Fisher Scientific, #A1310).<br><br>B cell depletion study: Isotype control (Biolegend, #400566) and Anti-CD20 antibody (Biolegend, #152104) |

| Validation | All antibodies applicable for western blot, Immunofluorescence microscopy, Flow cytometry and MIBI imaging were validated by manufacturers and validation report is available in the corresponding website. |

# Eukaryotic cell lines

Policy information about cell lines and Sex and Gender in Research

| Cell line source(s) | Mouse primary hepatocytes- Male C57BL-6/J mouse<br>HEK-293-ATCC<br>Hep3B-ATCC<br>Hepa1-6-ATCC |
| Authentication | Cell lines have been authenticated by original source and are authenticated in-house by observation of cell morphology |
| Mycoplasma contamination | Cell lines were not tested for mycoplasma contamination |
| Commonly misidentified lines<br>(See ICLAC register) | No commonly misidentified cell lines were used |

# Animals and other research organisms

Policy information about studies involving animals; ARRIVE guidelines recommended for reporting animal research, and Sex and Gender in Research

| Laboratory animals | C57BL-6/Acly f/f, C57BL-6 mice were used for animal experiments |
| Wild animals | None |
| Reporting on sex | Male mice were used for animal experiments. Sex was not considered during the experiments. |
| Field-collected samples | The study did not involve samples collected from the field. |
| Ethics oversight | Animal experiments were carried out using the guidelines approved by the Animal Research Ethics Board at McMaster University, Canada (Steinberg Laboratory Animal Utilization Protocol #16-12-42, 21-01-04) or the Institutional Animal Care and Use Committee (IACUC) at Icahn School of Medicine at Mount Sinai, NY (IACUC approval# PROTO202100080). |

Note that full information on the approval of the study protocol must also be provided in the manuscript.

# Plants

| Seed stocks | Not applicable |
| Novel plant genotypes | Not applicable |
| Authentication | Not applicable |

# Flow Cytometry

## Plots

Confirm that:

☒ The axis labels state the marker and fluorochrome used (e.g. CD4-FITC).

☒ The axis scales are clearly visible. Include numbers along axes only for bottom left plot of group (a 'group' is an analysis of identical markers).

☒ All plots are contour plots with outliers or pseudocolor plots.

☒ A numerical value for number of cells or percentage (with statistics) is provided.

## Methodology

| Sample preparation | Peripheral Blood Mononuclear Cells (PBMCs) were isolated from donors using the Ficoll Paque density gradient method. |

| | |
|---|---|
| Sample preparation | PBMCs were then washed, resuspended in complete RPMI1640 (10% FBS, 1% P/S, 1% L-Glutamine), and plated  in 12 well plates.  Next, PBMCs were incubated at 37oC, 5% CO2 incubator for 5 days. Cells were then harvested, stained with E780 Live/Dead stain (Thermofisher, #C34570), CD3 (Biolegend, #300434), CD19 (Biolegend, #302241) antibody and analyzed using flow cytometry.<br>For B cell depletion confirmation, the tail vein blood was collected from each animal after 15 days of single injection with 250 μg of anti-CD-20 (Biolegend, 152104) or isotype (Biolegend, 400566) antibody. The Red blood cells were lysed twice with 1X RBC lysis buffer. The samples were then centrifuged at 1500 rpm for 5 min at 4 °C. The cells were washed and blocked with Fc block (BD Biosciences, 553142) and stained with CD45.2 BV510 ( BioLegend, 109838), B220 (BD Biosciences, 563894), CD19 (Biolegend, 152409) and 7AAD (Thermo Fisher Scientific, A1310). |
| Instrument | CytoFlex  (Beckman Coulter Life Sciences) and BD LSR Fortessa (BD Biosciences). |
| Software | FlowJo v10.8.1 |
| Cell population abundance | Cells were not sorted in the present study. |
| Gating strategy | For PBMC experiment,FSC vs SSC plot of lymphocytes population, FSC-A vs FSC-H to identify single cell population, FSC vs e780 Viability Dye to identify live cells population, CD19 vs CD3 to identify CD19+ B cells and CD3+ T cells.<br><br>For B cell depletion study, SSC vs FSC plot of lymphocytes population, FSC-H vs FSC-A to identify single cell population, FSC vs 7-AAD to identify live cells population, B220 vs CD19 to identify B220+CD19+ B cell population |

☒ Tick this box to confirm that a figure exemplifying the gating strategy is provided in the Supplementary Information.

