## [Peer Review file · Nature]

ACLY inhibition enhances tumor immunogenicity and resolves MASH-driven HCC

Corresponding Author: Professor Gregory Steinberg

This file contains all reviewer reports in order by version, followed by all author rebuttals in order by version. Parts of this Peer Review File have been redacted as indicated to maintain the confidentiality of unpublished data.

Version 0:

Reviewer comments:

Referee #1

(Remarks to the Author)

In the manuscript by Jaya Gautam and colleagues "Genetic or Pharmacologic inhibition of ATP citrate lyase (ACLY) enhances tumor immunogenicity and resolves NASH-driven hepatocellular carcinoma" the authors investigate the role of the ATP citrate lyase (ACLY) in liver cancer formation in the context of NASH. This is an innovative manuscript that describes a possible role of ACLY involved in metabolic and immune modulation in the context of liver cancer.

The manuscript is - in part - well performed, however, in several points this manuscript falls short and draws conclusions that currently lack appropriate models, methods or controls.

Moreover, the manuscript neglects published literature that do not fit quite well the data or the conclusions of the authors – derived from NASH models that do not involve DEN or CCL4 – but that are just diet related - this should be taken into account by the authors, discussed and functionally described (what is the role of CXCR5+ B-cells – or also T cells involved):

(1) Importantly, the major model applied by the authors in this study does not reflect liver tumorigenesis through NASH and naturally linked fibrosis – but rather reflects DEN-induced liver cancer (or CCL4 triggered fibrosis) – which is then accelerated and exacerbated in the context of a high fat diet: Their NASH model is based on a chemical carcinogen in combination with a NASH diet (thus already the controls carry tumors). This referee does not criticize the model per se – but in the context of this manuscript there are massive effects on the immune system in case DEN is used for examples. One effect of DEN is the induction of enhanced expression of tumor neo-antigens (as a consequence of DEN-related mutagenesis – plus there is also damage during this process) – thus affecting an increased antigenicity of the liver tumor cells and an "artificial" immune response to antigens. It is not clear how the immune system reacts in a classical NASH-HCC model (e.g. induced by WD) – maybe – at least in the WT situation and for ACLY expression the authors could do this respectively.

(2) Moreover, the B-cell depletion data goes opposite to several NASH-HCC papers recently published that indicated that B-cell depletion in NASH-HCC actually reduces liver cancer (even in a therapeutic setting) (Deng CJ, et al., Int J Mol Sci. 2022; Weng et al., EBioMedicine 2018; Kotsiliti et al., J Hep 2023) - this is in contrast to what is shown in the manuscript (B-cell depletion (only in part) reverts the liver cancer reducing aspects (thus makes more tumors) of ACLY ko mice. What do the B-cells do functionally?

(3) The paper would profit from more detailed analyses of immune cells in mice (e.g. Where do the B-cells sit in the livers upon ACLY ko? What is the altered metabolic phenotype of immune cells (e.g. T and B cells) in ACLY ko mice? What do the B-cells do in order to enable better tumor surveillance in the Ko mice – is it B-cells only or CXCR5+ T cells? Who is really executing tumor immune surveillance?) as well as a better characterization of human data (NASH/NASH-HCC).

Still, overall, the concept of this manuscript is interesting but it necessitates several experiments, methods and controls to clearly decipher the mechanistic underpinning proposed.

Figure 1:

It is important that the authors apply a model of carcinogen independent induced NASH to HCC transition and test this in hepatocyte specific ACLY knock-out mice (or AAV-related as performed). In which cells parenchymal and non-parenchymal expressed is ACLY expressed in NASH induced HCC models (carcinogen independent)? Where is ACLY expressed in human NASH and human NASH-HCC? How does expression correlate with survival upon partial hepatectomy in human

liver cancer patients? The effect on the lipid distribution within liver cancer is interesting but also not too strong. How does this look in non-liver cancer affected tissue? When is the expression started – is it highly expressed in B-cells? A better characterization of the tumors in terms of genetics should be done. DNA damage, location of cell death, senescence, metabolic assays could/should be done.

In general, throughout the whole manuscript it would also be nice to show the tumor nodule size and to characterize the tumors better.

Figure 2: I would be important to get an overview on the metabolomic and proteomic phenotype in liver cancer and non-tumor affected tissue in ACYL ko mice and to see over time how the latter change. Protein expression changes of ACLY in liver tissue (tumor, non tumor?) – mouse and human?

Where are the B-cells located in the liver – and which B-cell subtypes are to be found increased in the ACYL ko livers? How efficient was the depletion of B cells?

Figure 3: This is an impressive figure and I like very much this analysis. The identification of a novel dicarboxylic-acid prodrug that is converted to CoA thioester and inhibits ACLY through its CoA-binding site is very impressive and should get more focus in the paper – effects on immune cells? Effects on hepatocytes? Effects on hepatoma cells/ liver cancer cells?

Figure 4: This is an interesting and in principle well performed figure. The prevention group shows a very impressive result. Also the combinatorial approach (F and G) is well done and interesting. Here I would propose again a classical diet related NASH model (like WD) and a combination of immunotherapy and anti-VEGFa – which is the current standard of care in HCC. ACLY expression and immunogenicity in human tissue - the correlation is clear – but what does it really mean and what is it based on ? Could this be done more convincingly in another assay? What are the proposed underpinnings here? Stratification for immunotherapy of HCC possible (responder non-responder)?

Referee #2

(Remarks to the Author)

In this study by Gautam and colleagues, a new inhibitor of ATP-citrate lyase (EVT0185) is reported and found to suppress HCC in the context of NASH. Studies in a genetic model of liver ACLY KO confirm that loss of ACLY suppresses NASH-HCC tumor growth. Both genetic ACLY KO and treatment with EVT0185 promote increased B cell infiltration of tumors, possibly due to increased Cxcl13 expression. New treatments for HCC are urgently needed, and it is encouraging that EVT0185 has similar anti-cancer effects as TKI used clinically to treat HCC in at least one mouse model. Thus the study has potential for high impact and translational relevance. However, there are several concerns with the study in its present form.

1. It is not clear whether ACLY is the relevant molecular target of EVT0185 in suppressing tumor growth. While it is shown that EVT0185 inhibits ACLY and that, similar to genetic ACLY KO, it reduces tumor growth, it is possible that it may be acting through additional targets to exert its anti-tumor effects. Indeed, in figure 4A, it is shown that the drug suppresses acetate-dependent lipogenesis. Since acetate is converted to acetyl-CoA for lipogenesis via ACSS2, not ACLY, this suggests that the drug's effects on lipid synthesis are not through ACLY. Perhaps it is also inhibiting ACSS2 or other enzymes in the DNL pathway. ACSS2 has previously been shown to be important for HCC tumor development, so this seems plausible (PMID: 25525877). Moreover, ACSS2 targeting has also been shown to impact anti-tumor immunity (PMID: 37723305). A key experiment is thus to treat the tumor-bearing liver ACLY KO mice with EVT0185 to determine if the drug may have additional relevant targets in this context.

2. Relationship of EVT0185 to the FDA-approved drug bempedoic acid is not tested or even discussed as far as I can tell- this would seem to be a key point for understanding the potential translational importance and novelty of this study. Looking at the structures of the molecules screened, including EVT0185, it is evident that they are very similar to bempedoic acid, which the authors have previously used in many prior studies, including in a related study to test efficacy in NASH treatment (PMID: 35675800). Many of the analogs tested are also very similar to those described in a 2004 study, which also impaired lipid synthesis (<https://doi.org/10.1021/jm040006p>). Furthermore the mechanism of action of EVT0185 appears to be similar if not identical to that of bempedoic acid, and both are prodrugs that require SLC27A2 for activation. Comparisons are warranted, both in terms of IC50 for inhibition of ACLY and in terms of in vivo effects on tumor growth. Bempedoic acid is already in clinical use- what advantage does EVT0185 provide over bempedoic acid?

3. Related to the two points above, it has been shown that bempedoic acid activates AMPK (PMID: 23118444), which suppresses ACC, and the authors previously showed that ACC inhibition can suppress HCC (PMID: 30244972). It seems plausible that AMPK activation accounts for part of the effect of EVT0185 on tumor growth. Is AMPK activated by EVT0185, similar to that reported for bempedoic acid?

4. B cell role is only correlative with the presented data. Figure 2Q is the missing key control to clarify this- the effect of anti-CD20 on tumor growth in WT animals. From current data, it is unclear if the depleting B cells will equally effect WT and KO tumors or whether it specifically reverses the anti-tumor effects of ACLY KO as claimed. In addition, B cell depletion experiments should be carried out in the context of the drug treatment studies.

5. It is shown that Cxcl13 expression increases with ACLY KO, suggesting a plausible mechanism linking ACLY deficiency to B cell infiltration. However, this mechanism has not been explored further, either in terms of a potential functional role of Cxcl13 or in terms of how ACLY might be regulating Cxcl13 expression. Additionally, the relationship between Cxcl13 and

Acly should also be examined in the human dataset.

6. T cell response has been previously reported to be linked to B cells in the context of HCC (PMID: 26669617). Figure 2K suggests that T cell infiltration also increases with ACLY KO. Is this important in the mechanism of tumor growth suppression? T cell depletion experiments could also be carried out.

7. Proposed mechanism of inhibition needs biochemical support. The authors claim that this inhibitor is CoA competitive by structural analysis. This claim should also be supported by biochemical analysis, i.e. mode of inhibition assays. The authors also suggest that the terminal carboxyl group of EVT0185-CoA could anchor to the citrate binding pocket of ACLY. However, the inhibitor is only partially resolved where it contacts the CoA binding pocket while the rest of the molecule is disordered, suggesting that the carboxyl end is not bound. Again, mode of inhibition studies are necessary to validate the proposed model.

Minor points:

8. Fig 1L: Please clarify what is measured here- free fatty acids or total saponified fatty acids

9. Figure 2A-C- test is too small to read

10. In figure 2/extended figure 3, the authors perform RNA-seq on multiple tumor timepoints. While the authors did demonstrate the experimental rationale for using multiple time points, the computational modeling strategy is not adequately explained in the text or methods.

11. Fig 1K: clarify what is meant by % lipid area? Does that mean within the circle drawn, the % of area that is lipid droplets? How are the lipid droplets defined- just by white area? It may be more appropriate to carry out neutral lipid staining.

12. Please clarify the statement on line 202: Acly KO tumors were depleted of unassigned cell types.

13. Fig 4A- please clarify timeline- how long were mice on diet before the 1 month drug treatment?

14. Further discussion about what is known about the role of B cells in HCC would be appreciated.

Referee #3

(Remarks to the Author)

Gautam, Wu, Lally, et al. present a study of the antiproliferative effects of ATP citrate lysase (ACLY) inhibition by both genetic and chemical means. The structure of ACLY with a newly identified inhibitor, EVT0185, was determined by cryo-EM. This review focuses on the structure determination.

The cryo-EM maps are of high quality and I have no concerns about their generation or interpretation.

I commend the authors for showing model-in-map fit in figures where necessary to support their interpretation.

Line 281-2 should be re-written for appropriate use of terminology and rounding of resolution (using the odd-even rule) to one decimal place. E.g.:

“A 3D reconstruction without the application of symmetry had a resolution of 3.6 Å following a gold-standard refinement”

Note the use of “gold” not “golden”. Also, it is the refinement that is gold standard, which means keeping the half maps separate, not the FSC (although people often say “gold standard FSC, as in line 817 here).

The phrase “in C1 symmetry” used throughout the manuscript is strange. Maps are not in a symmetry but have symmetry applied. Further, C1 symmetry means no symmetry was applied.

Line 791-792 should be improved to address the same points as line 281-2.

A reference should be added for non-uniform refinement in the methods section.

Figure ED10 is useful but the image quality should be improved: it is hard to read the text when zoomed in.

Something does not look right with the model-to-map FSC. The falloff in FSC is far more abrupt than I would expect. It looks as if either the map or the model-derived-map was filtered to 3.37 Å before calculation of the FSC (or perhaps this appearance is a result of how the FSC is plotted). The authors should double check that everything is as it should be.

Resolutions should be generally stated to one decimal place throughout the manuscript and in figures, although it is not necessary to change the FSC output from cryoSPARC in Fig. S10.

I do not know what “contoured at 0.176 V”, “contoured at 0.12 V”, or “contoured at 0.2 V” means in the figure captions (Figure 3, Figure ED9) and in the text (lines 952-957). What is “V”? Is it a fraction of the highest density value in the map? Please define.

Version 1:

Reviewer comments:

Referee #1

(Remarks to the Author)

In the revised version of this manuscript the authors have tried to answer most of the questions raised by the Editors and the referees. Most of the responses were convincing to this referee and indicate the mode of action of ATP citrate lyase inhibition - however, in my point of view there are several points not understood.

What is not convincing to me at that point is whether this therapy will work in pathophysiological state of the art model on MASH HCC - like WD induced liver cancer (without the use of chemical carcinogens).

DEN-WD is not a mouse model in which MASH is causally linked to liver cancer (like in the human situation). It is a mouse model in which cancer is induced by DEN and then accelerated by WD (due to its immunosuppressive, proproliferative function of the diet).

The reason why I am pertinent on this is the fact that DEN (used in their models) is a genotoxic and alkylating agent, that will induce a lot of tumor neoantigens - thus driving an immune response against tumor cells. Thus, the effects on anti-tumor surveillance are artificial and hyperreactive.

The authors state that due to time reasons they were not able to do these experiments, however I am not sure whether this is an argument not to do the most important experiment of the manuscript when it comes to application in human liver cancer caused by MASH - according to the standard of the field (see also PMID: PMC11199145 DOI: 10.1038/s42255-024-01043-6).

This also relates to their findings on B-cells - it is not unexpected/novel (B cell might prime T cells; they might also potentially actually contribute to killing) but novel and not clear to me would rather be to understand what the role of B cells exactly is. Will they interact/collaborate with T cells (CD8 killers) or even NK cells? Would the authors see an effect when combining anti CD8 with anti B-cell depletion.

It is also interesting to see that in their models the combination of immunotherapy and anti VEGFR2 is not made better by the addition of ACLy inhibition, when it comes to tumor load.

Also the human data are not clear to me - why did the authors not look into immunotherapy trials or cohorts of HCC patients with and without MASH and Atezo/Bev- when it comes to their markers?

In summary, this is an interesting manuscript, that in my view lacks one important experiment and does not address several obvious points (human cohorts, role of B cells and T cells) - for the reasons discussed above I believe that the latter experiments are needed to make a sufficiently important claim - as this manuscript intends to do.

Referee #2

(Remarks to the Author)

The authors have addressed my concerns. The manuscript is greatly improved and I believe will be impactful. I congratulate the authors on an excellent and exciting study.

One recommended additional citation for the discussion is that aligning with this study, a prior study also showed that ACLY targeting by shRNA or bempedoic acid in liver cancer cells sensitized to anti-PDL1 (PMID: 38055816).

Referee #3

(Remarks to the Author)

The reviewers have addressed my concerns about the cryo-EM aspects of the manuscript.

The only exception is related to stating the contour level of the map.

It is not strictly necessary to state the contour level each time one shows the map. However, if it is to be stated, it shouldn't be as an arbitrary value. If the authors wish to state the contour level they could normalize the map (e.g. in Chimera) so that the levels correspond to the standard deviation of the noise in the map. They could then specify the contour level in terms of sigma.

Version 2:

Reviewer comments:

Referee #1

(Remarks to the Author)

The authors have now fully replied to my points. This has become a very nice study, that might alter the landscape of therapy. Moreover, it demonstrates a novel mechanism. I have no further comments.

Referee #3

(Remarks to the Author)

The authors have addressed all of my concerns about the manuscript related to cryo-EM.

Referee #4

(Remarks to the Author)

Steinberg and coworkers describe a small molecule inhibitor of ATP citrate lyase (ACLY), which specifically exerts its activity in hepatocellular cancer cells, promoting the production of the chemoattractant CXCL13, thereby increasing B cell infiltration and suppressing tumour burden. The selective effect of the inhibitor is due to the expression of SLC27A2 required to generate the corresponding CoA thioester, exclusively in cancer cells but not in immune cells. The small molecule EVT0185 has been identified using an unbiased phenotypic screen, binding to the ACLY CoA binding site is supported by CryEM experiments and inhibition of ACLY is shown in vitro validating the work hypothesis. Conversion of EVT0185 into active EVT0185-CoA using rat liver microsomes is also shown. This is an interesting study, and the dataset is most compelling. The authors indicated 'sufficient half-life for chronic dosing', however half-life is not technically measured. While this statement is technically correct, I would recommend that the authors perform the measurement of rephrase. It could be that the half-life is extremely short but sufficient for the compound to exert its activity. The text may therefore be misleading at it stands. Also, toxicity in chronic dosing is not quantified at effective doses. A basic tolerability study may be useful. The compound EVT0185 contains two neopentyl centers next to the carboxylates making the latter poorly reactive. It is surprising that a thioester can be biosynthesized from this small molecule. It would be interesting in future work to investigate the effect of least hindered analogues or other methylated isomers. Also, it would be interesting to look at the status of permissive acetylated histone marks by ChIP-seq upon treatment with the compound. It is possible that expression of CXCL13 is simply the result of AcetylCoA levels being affected on inhibition of ACLY, directly impacting the acetylation status of specific histone marks impacting transcription of this gene (and others).

Response to the Editor's Comments

1. I think there is sufficient support to take this forward, though I think show superiority of EVT0185 over current ACLY inhibitors should be the bar, including in an additional model.

In the revised manuscript, we have shown the superiority of EVT0185 over the FDA-approved ACLY inhibitor, bempedoic acid, and completed further biochemical characterization as described below:

1. Isolated hepatocytes from WT and *Acly* KO mice comparing effects on *de novo* lipogenesis from acetate (ACLY-independent) and lactate (ACLY-dependent) (Extended Data Figure 7B, C)
2. Clonogenic survival assays in cultured Hep3B (human) and Hepa1-6 (mouse) HCC cell lines (Extended Data Figure 7H and 7I).
3. Established that EVT0185 has superior efficacy compared to bempedoic acid *in vivo* (Figure 4A and 4B) (Extended Data Figure 8D).

Point 1,2 and Point 3 are described below for the editor and in lines 245-265 and 277-282 of the revised manuscript.

Text in blue below addresses (Point 1 and Point 2) and can be found in Lines 245-265 in the manuscript.

In addition to inhibiting ACLY, long-chain fatty acyl-CoAs can also activate AMPK which then inhibits fatty acid synthesis through the phosphorylation and inhibition of acetyl-CoA carboxylase (ACC)⁴⁷ (Extended Data Figure 7A). Therefore, to examine whether EVT0185 might have additional activities beyond ACLY, we assessed the incorporation of lactate into fatty acids and sterols in hepatocytes from WT and *Acly* KO mice and contrasted these effects to equimolar concentrations of bempedoic acid (Extended Data Figure 7B). We also examined the effects of EVT0185 and bempedoic acid to inhibit acetate incorporation into fatty acids and sterols (Extended Data Figure 7C) which is not dependent on ACLY activity and can be blocked through inhibition of ACC and/or acetate CoA synthetase 2 (ACSS2). In WT hepatocytes EVT0185 exerted greater potency for inhibiting fatty acid and cholesterol synthesis from lactate or acetate than bempedoic acid, a difference which became more dramatic in the absence of ACLY suggesting additional targets might be important (Extended Data Figure 7B, C). We subsequently completed additional cell-free assays and found that in contrast to studies with bempedoic acid³⁶, EVT0185-CoA inhibited, not activated, AMPK β 1 containing heterotrimers (Extended Data Figure 7D). It also inhibited both acetyl-CoA carboxylase (ACC) isozymes (ACC1) and (ACC2) as well as

ACSS2 (Extended Data Figure 7E-G). EVT0185 also displayed greater inhibition of clonogenic survival of cultured human (Hep3B) and mouse (Hepa1-6) HCC cells lines than bempedoic acid (Extended Data Figure 7H, I). As activation of AMPK has been implicated in pro-survival pathways (reviewed in⁴⁸) and inhibition of ACLY often leads to compensatory upregulation of ACSS2^{8,9,12}, these data indicate important differences between EVT0185 and bempedoic acid which supported further development for HCC.

Extended Data Figure 7

Text in blue below addresses Point 3 (Line no: 277-282)

Using the WD-DEN mouse model described in Figure 1 and 2, mice were randomized based on AFP to daily gavage with vehicle, EVT0185 at 30 or 100 mg/kg or Bempedoic Acid at 100 mg/kg for one month (Extended Data Figure 8C, 8D). Compared to Vehicle, EVT0185 at 30 and 100 mg/kg reduced tumor burden by 58% and 67%, respectively, while bempedoic acid (100 mg/kg) had limited efficacy (Figure 4A and 4B).

Extended Data Figure 8

Figure 4

In summary, as requested we have completed multiple studies: in vitro biochemistry, cell-based assays (hepatocytes and cancer cells) and in vivo efficacy, which clearly differentiate EVT0185 and show superiority compared to bempedoic acid.

PARAGRAPH REDACTED

FIGURE REDACTED

We believe the necessity to use the WD is unreasonable as the WD-CCl₄ model and DEN models of MASH-HCC have both been recently utilized extensively in high-impact publications including recently in Nature (PMID: 36198802). Importantly, the WD-CCl₄ model has also been directly compared to pure diet-induced models of MASH-HCC and found to be very similar to these models and the immunosuppressive microenvironment of human disease (PMID: 33992698). In addition, compared to other mouse models, the WD-CCl₄ model has similar histological and transcriptional characteristics that closely align with human MASH and fibrosis (PMID: 38867022). Therefore, we believe the use of this model is well validated and as indicated in the above-cited paper is comparable to “pure diet-induced” and not subject to “artificial” immune response to antigens.

In the revised version of the paper, we have also further characterized the genetic features of our new WD-DEN model which is used in the *Acly* KO and EVT0185 studies. It should be noted that this model involves only a single injection of DEN at 2 weeks of age, not repeated injections as we and others have used previously to induce HCC in rats (PMID: 30244972). In the initial manuscript version, we showed that this model had similar pathological features to humans with MASH-HCC. In the revised version of this manuscript, we have now further characterized the genetic features of the model by completing bulk and single cell RNAseq and compared it directly with the well-validated WD-CCl₄ HCC model and human MASH-HCCs described below and in lines no: 125-147) of the manuscript.

To further validate whether the WD-DEN model might be a suitable model for studying MASH-HCC, we completed bulk RNA sequencing of tumors and explored the correlation in gene expression between tumor from patients with MASH-HCC (Extended Data Figure 1J). We also compared these findings with the Control-DEN model and the well-established WD-CCl₄ (also known as FAT-MASH) model^{28,32,33} (Extended Data Figure 1J). Consistent with a previous report²⁸, we validated the unique transcriptomic enrichment of the Hoshida S1 subtype³⁴ and depletion of the S2 subtype among patients with MASH-HCC relative to HCC of mixed etiology (Figure 1F). We then selected orthologous genes featured among Hoshida subtypes to evaluate the similarity between mouse models and human MASH-HCC samples. Correlation analysis revealed a unique similarity between the WD-DEN model and human S1 subtype MASH-HCC samples. In contrast, WD-CCl₄ (PRJNA386995)³² and Control-DEN (PRJNA488497)³⁵ models were both more associated with S2 and S3 subtypes of MASH-HCC (Figure 1G). Classification of mouse samples based on molecular subtype using Nearest Template Prediction also identified substantial enrichment of S1 subtype among samples derived from the WD-DEN mouse model whereby 8 out of 9 samples were classified as S1, in contrast to Control-DEN and WD-CCl₄ samples which were classified as S2 and S3 subtypes respectively (Figure 1F).

To further define whether there were similarities with immune cell and tumor cell populations in our newly developed WD-DEN model and the WD-CCl₄ model we conducted single-cell RNA sequencing (scRNAseq) of livers from both models and compared them to a recently published dataset of humans with MASH-HCC³⁶. We found that lymphocytes and tumor cells (AFP⁺ and GPC3⁺) of the WD-DEN model correlated in a similar manner to that of the previously validated WD-CCl₄ mouse model²⁸ and humans with MASH-HCC (Figure 1H).

Extended Data Figure 1

J

Figure 1

F

G

H

In addition to the above experiments highlighting the relevance of the WD-DEN and WD-CCL4 models we have also conducted studies in an additional model. To determine whether the immunogenic response is important for reducing tumor burden, we have generated an inducible system for knocking down ACLY expression in the human Hep3B cell line for orthotopic implantation into immunodeficient NRG mice (Extended Data Figure 13A). The results from this model described below indicate the importance of a functional adaptive immune system for reducing the tumor burden with ACLY inhibition.

Lines no: 398-407

To determine whether the immunogenic response was important for reducing tumor burden in response to genetic inhibition of ACLY, we generated an inducible system for knocking down ACLY expression in the human Hep3B cell line (Extended Data Figure 13A) for orthotopic implantation in immunodeficient NRG mice. As anticipated, ACLY knockdown reduced glucose incorporation into triglycerides and led to compensatory upregulation of acetate incorporation into triglycerides (Extended Data Figure 13B, C). In addition, inducible knockdown increased fatty acid oxidation (Extended Data Figure 13D). However, despite these anticipated changes in metabolic activity ACLY knockdown did not impair orthotopic tumor growth in immunodeficient mice (Extended Data Figure 13E, F). This suggests that alteration in lipid metabolism without a functional adaptive immune system was not sufficient for reducing tumor burden.

Extended Data Figure 13

Lastly, to further substantiate the immune contribution and to enhance the clinical relevance of our findings (as suggested by reviewer 1), we now also show the combination of EVT0185 + anti-PDL1/VEGFR is highly effective at reducing tumor burden in the well-characterized WD-CCl₄ model. These results are described in lines 291-297 and below for the editor.

To evaluate whether EVT0185 might also enhance response rates to the current standard of care immunotherapy Atezolizumab plus Bevacizumab¹, we completed a combination study with anti-PDL-1/VEGFR antibodies in the WD-CCl₄ model. Consistent with previous reports¹, we found that anti-PDL-1/VEGFR antibodies had minimal effects on tumor burden (Figure 4J-4L). However, the triple combination of EVT0185 + anti-PDL-1/VEGFR dramatically reduced tumor burden (Figure 4J, 4K) (Extended Data Figure 8H) and the number of animals with over 25 tumors indicating significant therapeutic benefit (Figure 4L).

Figure 4

Extended Data Figure 8

2. A revision will in particular need to address the mechanism by which EVT0185 acts, and provide further data with EVT0185, including biochemistry and benchmarking, as well as an additional model.

In addition to the biochemistry and benchmarking data provided above, we also now show that the inhibition of ACLY by EVT0185-CoA is competitive with Coenzyme A (Figure 3C, 3D) (Extended Data Figure 5A). The drastic shift of K_i' compared to K_i by global fit means the compound has almost no affinity to ACLY/ CoA complex compared to free ACLY and indicates that EVT0185-CoA competes with CoA to bind ACLY, thus, the compound shows competitive inhibition for ACLY with respect to Coenzyme A consistent with our cryo-EM structure. It should be noted that it is not possible to complete these assays with bempedoic acid-CoA as this compound is not commercially available and would require complex synthesis which we are not able to complete.

Lines no: 214-217

ACLY is inhibited by palmitoyl-CoA⁵, therefore we hypothesized that the conversion of EVT0185 to its CoA-thioester may drive the inhibition of ACLY activity. In cell-free assays EVT0185-CoA (not the unconjugated diacid) inhibited recombinant human ACLY activity (Figure 3B) and this effect was competitive with CoA (Figure 3C, D) (Extended Data Figure 5A).

Figure 3

C

D

Figure 3. C, Michaelis-Menten Plot and D, Lineweaver-Burk Plot for ACLY with EVT0185-CoA using GraphPad Prism

Extended Data Figure 5

A

	MOA	Substrate		
		Km (μM)	Ki by Global Fit (μM)	Ki' by Global Fit (μM)
EVT0185-CoA, Data 1	Competitive with CoA	1.4 \pm 0.2	0.18 \pm 0.03	>>>Ki
EVT0185-CoA, Data 2	Competitive with CoA	1.3 \pm 0.2	0.15 \pm 0.03	>>>Ki
EVT0185-CoA, Data 3	Competitive with CoA	0.9 \pm 0.1	0.13 \pm 0.02	>>>Ki

Kinetic constants of EVT0185-CoA determined by Global Fit

Extended Data Figure 5. A, Global fit of Michaelis-Menten Plots by GraFit software for EVT0185-CoA in ACLY.

With respect to mechanism of action of EVT0185 we now utilize spatial transcriptomics, single-cell RNAseq, single-cell proteomics (MIBI-TOF) and immunohistochemistry and show that like ACLY KO mice, tumors from EVT0185 treated mice have an increase in plasma B cells and tertiary lymphoid structures. The results of these new experiments are described below.

Spatial Transcriptomics and scRNAseq (line no: 336-361)

To examine with greater fidelity these tumor, immune interactions we completed spatial transcriptomics on livers from WT and *Acly* KO mice (Figure 5E, F) and mice treated with Vehicle or EVT0185 (100 mg/kg) (Figure 5K, L) in the WD-CCl₄ model of HCC. GO enrichment analysis of the spatially resolved HCC cells revealed that there were increases in fatty acid and lipid

metabolism in both *Acly* KO (Figure 5G) and EVT0185-treated (Figure 5M) mice, a finding consistent with previous studies in cultured cells¹⁰. This included upregulation of *Acss2*, *Acacb*, and *Acat2* (fatty acid oxidation), *Gpat3* (TG synthesis), and *Hmgcs1* (cholesterol synthesis) (Figure 5H, 5N). Spatial integration analysis of tumors revealed an increase in the number of B-cells (Figure 5F), but not other immune cell populations (Extended Data Figure 10A), in *Acly* KO mice compared to WT controls. A similar increase in the number of B-cells but no other immune cell types (T cells, macrophages, NKT cells) was also observed with EVT0185 compared to vehicle control (Figure 5L, Extended Data Figure 10B). We further interrogated the B-cell subtypes within the tumors using established markers⁵⁴ and found that in both *Acly* KO and EVT0185-treated mice there was a much higher proportion of plasma cells, which are critical for humoral immunity through the production of antibodies (Figure 5I, O). Consistent with an important role for fatty acid metabolism in the differentiation of plasma cells⁵⁵, GO enrichment analysis of the B-cells found that this pathway was upregulated in both *Acly* KO and EVT0185-treated mice (Extended Data Figure 10C). Similar observations were observed using scRNAseq from WD-DEN and WD-CCl₄ mice treated with EVT0185 (Extended Data Figure 10D, E). CXCL13 is a B-cell chemoattractant that has recently been shown in clinical populations to be reduced in MASH-driven HCC³⁰. Consistent with increases in B-cell infiltration, spatially resolved tumors of both *Acly* KO and EVT0185 treated mice had increased levels of *Cxcl13* (Figure 5J, P), a finding replicated in publicly available RNA-seq data from WT and *Acly* KO DEN-induced tumors cultured in vitro (GSE223966)¹⁰ (Extended Data Figure 10F). These data indicate that genetic and pharmacological inhibition of ACLY in HCC cells leads to increases in tumor CXCL13 and plasma B-cells.

Figure 5

E

F

K

L

G

I

J

H

M

O

P

N

Extended Data Figure 10

A

B

C

D

E

F

IHC and Single Cell Proteomic (MIBI-TOF) analysis

Results line no: 379-388, Discussion line no: 470-475:

Further analysis of the IHC images by a pathologist blinded to the treatments found diffuse and single B Cell infiltration with accentuation in the tumor periphery in both WT and KO groups (Figure 6G). However, the B-cell aggregations resembling tertiary lymphoid structures (TLSs) were identified mostly in KO group (80%) (Figures 6H) which have been associated with beneficial clinical outcomes⁵⁷ and are linked to reduced risk of HCC recurrence following resection⁵⁸. MIBI and IHC revealed very similar changes in mice treated with EVT0185 compared to vehicle controls (Extended Data Figure 11C-J). There were minimal or no changes in the expression of the T-cell marker CD3 (Extended Data Figure 11G, H). Consistent with the *Acl* KO, we found the number of TLSs was also higher in EVT-treated mice (Extended Data Figure 11I, J).

Intratumoral TLS has been associated with favourable clinical outcome in several cancer types including HCC^{57,58,66-68}. B cells in TLS are known to communicate with follicular T-cells which help them to proliferate and differentiate into long-lived plasma cells that can produce antibodies and induce antitumor immunity through antibody-dependent cytotoxicity (ADCC), activation of the complement system and complement-dependent cytotoxicity⁶⁹.

Figure 6

G

H

Extended Data Figure 11

C

D

E

F

G

H

I

J

3. Please also clarify the library used for the screen, and if there was any novel synthesis involve to generate EVT0185, in which case we need the full synthesis details and characterization of intermediates (see <https://www.nature.com/nature/for-authors/formatting-guide>).

The library used for the screening of the compounds was originally listed in Supplementary Table 1 and has been disclosed in the US patent number US 11,730,712 (2023), columns 1333-1336, Table 7 (EVT0185 as “I-1”) (Activity of Illustrative Compounds of the Invention and of Reference Compounds for Inhibition of Incorporation of ¹⁴C-Acetate in Primary Mouse Hepatocyte Total Lipids over a 4-Hour Period).

This has been described on lines no: 188-195 of the revised manuscript and below for the editor.

To identify novel ACLY inhibitors that may have greater potency and cellular permeability than bempedoic acid and allosteric inhibitors, respectively, we conducted a phenotypic screen in primary mouse hepatocytes examining fatty acid and cholesterol synthesis from a library of novel molecules with variations in hydrocarbon chain length, di-carboxy or -hydroxyl terminal groups, gem-dimethyl groups, and central substitutions in sulfide, ester, keto, hydroxyl, and benzene. While several compounds inhibited lipogenesis (Supplementary Table S1 and originally disclosed in⁴⁴), 6-[4-(5-carboxy-5-methyl-hexyl)-phenyl]-2,2-dimethylhexanoic acid (EVT0185) was selected for further characterization (-84% inhibition of DNL at 100 μM, IC₅₀=0.46 μM)

Supplementary Table S1. Effect of compounds on lipogenesis in mouse primary hepatocytes

Compound Name	Structure	% Change from control at 100 μM ±SEM(n)	Statistical difference vs control	p- value	IC ₅₀ in μM
EVT0019		-30 ± 2.1 (n=4)	Yes	0.0020	ND
EVT0024		-41 ± 4.8 (n=2)	Yes	0.0229	73.23
EVT0025		-16 ± 4.4 (n=4)	No	ns	ND

Compound Name	Structure	% Change from control at 100 μ M \pm SEM(n)	Statistical difference vs control	p-value	IC ₅₀ in μ M
EVT0026		-52 \pm 5.5 (n=4)	Yes	0.0073	ND
EVT0054		-69 \pm 2.3 (n=4)	Yes	<0.0001	25.58
EVT0139		-33 \pm 6.7 (n=4)	Yes	0.0151	ND
EVT0146		-76 \pm 4.7 (n=4)	Yes	0.0014	0.46
EVT0149		-90 \pm 1.5 (n=4)	Yes	<0.0001	5.39
EVT0165		-20 \pm 5.4 (n=4)	No	ns	ND
EVT0173		-82 \pm 2.8 (n=6)	Yes	<0.0001	0.34
EVT0174		+44 \pm 8.1 (n=4)	Yes	<0.0001	ND
EVT0175		-62 \pm 4.5 (n=4)	Yes	<0.0001	0.35
EVT0185		-84 \pm 1.0 (n=6)	Yes	<0.0001	0.46
EVT0186		-70 \pm 2.2 (n=4)	Yes	<0.0001	27.66

Compound Name	Structure	% Change from control at 100 μ M \pm SEM(n)	Statistical difference vs control	p- value	IC ₅₀ in μ M
EVT0187		-91 \pm 3.6 (n=4)	Yes	<0.0001	<0.3
EVT0199		-43 \pm 7.3 (n=4)	Yes	<0.0001	ND
EVT0203		-38 \pm 8.6 (n=4)	Yes	0.0091	ND
EVT0210		-12 \pm 3.3 (n=4)	No	ns	ND

The compounds were synthesized as described in patents US 11,098,002 (2021) and US 11,730,712 (2023). The synthetic pathway for EVT0185 (as “Compound I-1”) and the characterization of the intermediates (B1, B2, and B3) are disclosed in the US 11,098,002 (2023) (Compound I-1, Example 10, columns 314-317).

This has been described in the method section of the revised manuscript in lines no: 523-524.

Synthesis of EVT0185:

EVT0185 (6-[4-(5-carboxy-5-methyl-hexyl)-phenyl]-2,2-dimethylhexanoic acid) was prepared according to the methods disclosed in Compound I-1, Example 10⁷⁸.

Structure of EVT0185

Referee #1 (Remarks to the Author):

In the manuscript by Jaya Gautam and colleagues “Genetic or Pharmacologic inhibition of ATP citrate lyase (ACLY) enhances tumor immunogenicity and resolves NASH-driven hepatocellular carcinoma” the authors investigate the role of the ATP citrate lyase (ACLY) in liver cancer formation in the context of NASH. This is an innovative manuscript that describes a possible role of ACLY involved in metabolic and immune modulation in the context of liver cancer.

The manuscript is - in part - well performed, however, in several points this manuscript falls short and draws conclusions that currently lack appropriate models, methods or controls.

Moreover, the manuscript neglects published literature that do not fit quite well the data or the conclusions of the authors – derived from NASH models that do not involve DEN or CCL4 – but that are just diet related - this should be taken into account by the authors, discussed and functionally described (what is the role of CXCR5+ B-cells – or also T cells involved):

Reviewer #1:

(1) Importantly, the major model applied by the authors in this study does not reflect liver tumorigenesis through NASH and naturally linked fibrosis – but rather reflects DEN-induced liver cancer (or CCL4 triggered fibrosis) – which is then accelerated and exacerbated in the context of a high fat diet: Their NASH model is based on a chemical carcinogen in combination with a NASH diet (thus already the controls carry tumors). This referee does not criticize the model per se – but in the context of this manuscript there are massive effects on the immune system in case DEN is used for examples. One effect of DEN is the induction of enhanced expression of tumor neo-antigens (as a consequence of DEN-related mutagenesis – plus there is also damage during this process) – thus affecting an increased antigenicity of the liver tumor cells and an “artificial” immune response to antigens. It is not clear how the immune system reacts in a classical NASH-HCC model (e.g. induced by WD) – maybe – at least in the WT situation and for ACLY expression the authors could do this respectively.

No single mouse model can capture all aspects of human MASH-HCC.

In the current study we have utilized two distinct mouse models involving a western diet in combination with either DEN or CCL₄ to demonstrate the effects of EVT0185.

The WD-CCL₄ model and DEN models of MASH-HCC have both been recently utilized extensively in high-impact publications including recently in Nature (PMID: 36198802). Importantly, the WD-CCL₄ model has also been directly compared to pure diet-induced models of MASH-HCC and found to be very similar to these models and the immunosuppressive microenvironment of human disease (PMID: 33992698). In addition, compared to other mouse models, the WD-CCL₄ model has similar histological and transcriptional characteristics that closely align with human MASH and fibrosis (PMID: 38867022). Therefore, we believe the use of this model is well validated and as indicated in the above-cited paper is comparable to “pure diet-induced” and not subject to “artificial” immune response to antigens.

In the revised version of the paper, we have also now further characterized our new WD-DEN model which is used in the ACLY KO and EVT0185 studies. It should be noted that this model involves only a single injection of DEN at 2 weeks of age, not repeated injections as we and others have used previously to induce

HCC in rats (PMID: 30244972). In the initial version of the manuscript, we showed that this model had similar pathological features to humans with MASH-HCC. In the revised version of this manuscript, we have now further characterized the genetic features of the model by completing bulk and single-cell RNAseq (scRNAseq) as described below (line no: 125-147) and compared it directly with the well-validated WD-CCl₄ HCC models described above. Remarkably using multiple lines of unbiased technologies, we observed very similar responses between models.

Lines no:125-147

To further validate whether the WD-DEN model might be a suitable model for studying MASH-HCC, we completed bulk RNA sequencing of tumors and explored the correlation in gene expression between tumor from patients with MASH-HCC (Extended Data Figure 1J). We also compared these findings with the Control-DEN model and the well-established WD-CCl₄ (also known as FAT-MASH) model^{28,32,33} (Extended Data Figure 1J). Consistent with a previous report²⁸, we validated the unique transcriptomic enrichment of the Hoshida S1 subtype³⁴ and depletion of the S2 subtype among patients with MASH-HCC relative to HCC of mixed etiology (Figure 1F). We then selected orthologous genes featured among Hoshida subtypes to evaluate the similarity between mouse models and human MASH-HCC samples. Correlation analysis revealed a unique similarity between the WD-DEN model and human S1 subtype MASH-HCC samples. In contrast, WD-CCl₄ (PRJNA386995)³² and Control-DEN (PRJNA488497)³⁵ models were both more associated with S2 and S3 subtypes of MASH-HCC (Figure 1G). Classification of mouse samples based on molecular subtype using Nearest Template Prediction also identified substantial enrichment of S1 subtype among samples derived from the WD-DEN mouse model whereby 8 out of 9 samples were classified as S1, in contrast to Control-DEN and WD-CCl₄ samples which were classified as S2 and S3 subtypes respectively (Figure 1F).

To further define whether there were similarities with immune cell and tumor cell populations in our newly developed WD-DEN model and the WD-CCl₄ model we conducted single-cell RNA sequencing (scRNAseq) of livers from both models and compared them to a recently published dataset of humans with MASH-HCC³⁶. We found that lymphocytes and tumor cells (AFP⁺ and GPC3⁺) of the WD-DEN model correlated in a similar manner to that of the previously validated WD-CCl₄ mouse model²⁸ and humans with MASH-HCC (Figure 1H).

Extended Data Figure 1

Figure 1

PARAGRAPH REDACTED

FIGURE REDACTED

-In summary: 1) Given the challenges with generating a WD model of HCC in the absence of a chemical carcinogen, 2) our new transcriptional characterization of the WD-DEN model showing it closely represents the genetic and immune features of advanced MASH-driven HCC 3) Similar phenotypic, spatial transcriptomic and single-cell proteomic data showing that EVT0185 elicits the same effect in both WD-CCl₄ and WD-DEN mouse models and 4) our clinical data showing strong associations between ACLY and B-cells in people with HCC we believe the current data strongly supports the clinical relevance of our findings.

(2) Moreover, the B-cell depletion data goes opposite to several NASH-HCC papers recently published that indicated that B-cell depletion in NASH-HCC actually reduces liver cancer (even in a therapeutic setting) (Deng CJ, et al., *Int J Mol Sci.* 2022; Weng et al., *EBioMedicine* 2018; Kotsiliti et al., *J Hep* 2023) - this is in contrast to what is shown in the manuscript (B-cell depletion (only in part) reverts the liver cancer reducing aspects (thus makes more tumors) of ACLY ko mice. What do the B-cells do functionally?

We appreciate the reviewer's perspective that the data may be opposing to some publications in the area. However, as we are sure the reviewer appreciates context and timing are critical to understand the tumor immune microenvironment.

For example, in addition to the publications cited in the manuscript, there have been several recent studies in humans which have shown a protective role for B-cells in clinical populations with HCC (PMIDs: 36092319, 29050338, 34796337, 37723590).

With respect to what the B-cells may be doing, we have now carefully characterized these populations using spatial transcriptomics, IHC, and multi-ion beam time of flight tomography (MIBI-TOF) single-cell proteomics as described below.

Spatial Transcriptomics (line no: 336-361)

To examine with greater fidelity these tumor, immune interactions we completed spatial transcriptomics on livers from WT and *Acly* KO mice (Figure 5E, F) and mice treated with Vehicle or EVT0185 (100 mg/kg) (Figure 5K, L) in the WD-CCl₄ model of HCC. GO enrichment analysis of the spatially resolved HCC cells revealed that there were increases in fatty acid and lipid metabolism in both *Acly* KO (Figure 5G) and EVT0185-treated (Figure 5M) mice, a finding consistent with previous studies in cultured cells¹⁰. This included upregulation of *Acss2*, *Acacb*, and *Acat2* (fatty acid oxidation), *Gpat3* (TG synthesis), and *Hmgcs1* (cholesterol synthesis) (Figure 5H, 5N). Spatial integration analysis of tumors revealed an increase in the number of B-cells (Figure 5F), but not other immune cell populations (Extended Data Figure 10A), in *Acly* KO mice compared to WT controls. A similar increase in the number of B-cells but no other immune cell types (T cells, macrophages, NKT cells) was also observed with EVT0185 compared to vehicle control (Figure 5L, Extended Data Figure 10B). We further interrogated the B-cell subtypes within the tumors using established markers⁵⁴ and found that in both *Acly* KO and EVT0185-treated mice there was a much higher proportion of plasma cells, which are critical for humoral immunity through the production of antibodies (Figure 5I, O). Consistent with an important role for fatty acid metabolism in the differentiation of plasma cells⁵⁵, GO enrichment analysis of the B-cells found that this pathway was upregulated in both *Acly* KO and EVT0185-treated mice (Extended Data Figure 10C). Similar observations were observed using scRNAseq from WD-DEN and WD-CCl₄ mice treated with EVT0185 (Extended Data Figure 10D, E). CXCL13 is a B-cell chemoattractant that has recently been shown in clinical populations to be reduced in MASH-driven HCC³⁰. Consistent with increases in B-cell infiltration, spatially resolved tumors of both *Acly* KO and EVT0185 treated mice had increased levels of *Cxcl13* (Figure 5J, P), a finding replicated in publicly available RNA-seq data from WT and *Acly* KO DEN-induced tumors cultured in vitro (GSE223966)¹⁰ (Extended Data Figure 10F). These data indicate that genetic and pharmacological inhibition of ACLY in HCC cells leads to increases in tumor CXCL13 and plasma B-cells.

Figure 5

E

F

K

L

G

I

J

H

M

O

P

N

Extended Data Figure 10

A

B

C

B cells in Acly KO vs WT

D

WD-DEN HCC (scRNA-Seq) B Cells in EVT0185 vs Vehicle

E

WD-CCI1 HCC (scRNA-Seq) B Cells in EVT0185 vs Vehicle

F

IHC and MIBI-TOF analysis

Results line no: 379-388, Discussion line no: 470-475

Further analysis of the IHC images by a pathologist blinded to the treatments found diffuse and single B Cell infiltration with accentuation in the tumor periphery in both WT and KO groups (Figure 6G). However, the B-cell aggregations resembling tertiary lymphoid structures (TLSs) were identified mostly in KO group (80%) (Figures 6H) which have been associated with beneficial clinical outcomes⁵⁷ and are linked to reduced risk of HCC recurrence following resection⁵⁸. MIBI and IHC revealed very similar changes in mice treated with EVT0185 compared to vehicle controls (Extended Data Figure 11C-J). There were minimal or no changes in the expression of the T-cell marker CD3 (Extended Data Figure 11G, H). Consistent with the *Acly* KO, we found the number of TLSs was also higher in EVT-treated mice (Extended Data Figure 11I, J).

Intratumoral TLS has been associated with favourable clinical outcome in several cancer types including HCC^{57,58,66-68}. B cells in TLS are known to communicate with follicular T-cells which help them to proliferate and differentiate into long-lived plasma cells that can produce antibodies and induce antitumor immunity through antibody-dependent cytotoxicity (ADCC), activation of the complement system and complement-dependent cytotoxicity⁶⁹.

Figure 6

Extended Data Figure 11

(3) The paper would profit from more detailed analyses of immune cells in mice (e.g. Where do the B-cells sit in the livers upon ACLY ko? What is the altered metabolic phenotype of immune cells (e.g. T and B cells) in ACLY ko mice? What do the B-cells do in order to enable better tumor surveillance in the Ko mice – is it B-cells only or CXCR5+ T cells? Who is really executing tumor immune surveillance?) as well as a better characterization of human data (NASH/NASH-HCC).

-To characterize the potential importance of the immunogenic response, we generated an inducible system for knocking down ACLY expression in the human Hep3B cell line for orthotopic implantation and showed that alteration in lipid metabolism without a functional adaptive immune system was not sufficient for reducing tumor burden.

Line no: 398-407

To determine whether the immunogenic response was important for reducing tumor burden in response to genetic inhibition of ACLY, we generated an inducible system for knocking down ACLY expression in the human Hep3B cell line (Extended Data Figure 13A) for orthotopic implantation in immunodeficient NRG mice. As anticipated, ACLY knockdown reduced glucose incorporation into triglycerides and led to compensatory upregulation of acetate incorporation into triglycerides (Extended Data Figure 13B, C). In addition, inducible knockdown increased fatty acid oxidation (Extended Data Figure 13D). However, despite these anticipated changes in metabolic activity ACLY knockdown did not impair orthotopic tumor growth in immunodeficient mice (Extended Data Figure 13E, F). This suggests that alteration in lipid metabolism without a functional adaptive immune system was not sufficient for reducing tumor burden.

Extended Data Figure 13

A) The paper would profit from more detailed analyses of immune cells in mice (e.g. Where do the B-cells sit in the livers upon ACLY ko?)

-As detailed using spatial transcriptomics (Figure 5E-P and Extended Data Figure 10A-F, described above in question no. 2), IHC and MIBI-TOF analysis (Figure 6A-H, Extended Data Figure 11C-J, described below), we perform detailed analyses of immune cells and show that genetic deletion of ACLY or treatment with EVT0185 increases B-cells within the tumor periphery.

MIBI TOF and IHC analysis (Lines no: 368-388)

Using a panel of 21 mono-isotopic Lanthanide metal-conjugated antibodies (Supplementary Table S3), we performed automated clustering with the Bioconductor R package 'FlowSOM' and revealed 9 types of cells present in the tissues. These were: B cells, CD4 T cells, CD11b+ macrophages, CD11c+ dendritic cell/macrophages, F4/80+ macrophages, myofibroblasts, endothelial cells, endothelial-mesothelial cells, and tumor cells (Figure 6A). Consistent with the transcriptomics, analysis from fields of view representing the leading edge of the tumor lesion of *Acly* KO mice compared to WT controls revealed a significant increase in B-cells (Figure 6B, C) but no other immune cells (Figure 6A, D). Consistent with a reduction in the lipid droplet area, lipid droplet-associated protein (Plin2) pixel area was reduced in *Acly* KO compared to WT mice (Extended Data Figure 11B). Histological staining of tumors of *Acly* KO mice confirmed large increases in CD19 ($p=0.0004$) (Figure 6E) along with an increase in CXCL13 protein expression (Figure 6F). Further analysis of the IHC images by a pathologist blinded to the treatments found diffuse and single B Cell infiltration with accentuation in the tumor periphery in both WT and KO groups (Figure 6G). However, the B-cell aggregations resembling tertiary lymphoid structures (TLSs) were identified mostly in KO group (80%) (Figures 6H) which have been associated with beneficial clinical outcomes⁵⁷ and are linked to reduced risk of HCC recurrence following resection⁵⁸. MIBI and IHC revealed very similar changes in mice treated with EVT0185 compared to vehicle controls (Extended Data Figure 11C-J). There were minimal or no changes in the expression of the T-cell marker CD3 (Extended Data Figure 11G, H). Consistent with the *Acly* KO, we found the number of TLSs was also higher in EVT-treated mice (Extended Data Figure 11I, J).

Figure 6

Figure 6

Extended Data Figure 11

B) What is the altered metabolic phenotype of immune cells (e.g. T and B cells) in *ACLY* ko mice?

-As described above in reviewer's question no. 2, we provide the altered metabolic phenotype of immune cells by *ACLY* inhibition and discuss in lines: 350-354.

Consistent with an important role for fatty acid metabolism in the differentiation of plasma cells⁵⁵, GO enrichment analysis of the B-cells found that this pathway was upregulated in both *Acly* KO and EVT0185-treated mice (Extended Data Figure 10C). Similar observations were observed using scRNAseq from WD-DEN and WD-CCl₄ mice treated with EVT0185 (Extended Data Figure 10D, E).

Extended Data Figure 10

C

D

E

C) What do the B-cells do in order to enable better tumor surveillance in the Ko mice – is it B-cells only or CXCR5+ T cells?

-As described above in reviewer's question no. 2, using spatial transcriptomics (Figure 5I and 5O), IHC and single-cell proteomics (Figure 6H) (Extended Data Figure 11I, J), we find that the B-cells form terminal lymphoid structures (TLSs) which have been linked to increased antibody production.

-We could find no evidence for the role of CXCR5+T cells in any of our analysis (Extended Data Figure 10 A and B) (Extended Data Figure 11 E, G, and H) (Figure 6D).

D) Who is really executing tumor immune surveillance?) as well as a better characterization of human data (NASH/NASH-HCC).

-Our data in which B-cells are depleted using an antibody to CD20 (Figure 6J-L) indicate that B-cells are critical for executing tumor immune surveillance. These data are consistent with recent studies described above in reviewer question no. 2, which have revealed connections between B cells and HCC in clinical populations (PMIDs: 36092319, 29050338, 34796337, 37723590).

Figure 6

-We complete further characterization of the human data as requested in lines 433-440 and described below for the reviewer.

In addition to the bulk transcriptome analysis, analysis of scRNA seq data found significant upregulation of *ACLY* expression in human MASH-HCC tumor (Extended Data Figure 15D) with significant overexpression of *ACLY* in malignant hepatocytes compared to normal hepatocytes using single-nuclei sequencing of human MASH-HCC³⁶ (Figure 6N). Next, we investigated the association between *ACLY* and immune infiltration in MASH-HCC tumor to assess if a similar association between *ACLY* and immunogenicity identified in our experimental *Acly* KO model could be observed in human tissue samples⁶¹. We observed consistent inverse correlation between *ACLY* and B cell markers identified from the human liver milieu (Figure 6O).

Extended Data Figure 15

Figure 6

Figure 6

Still, overall, the concept of this manuscript is interesting but it necessitates several experiments, methods and controls to clearly decipher the mechanistic underpinning proposed.

Reviewer Question 4: Figure 1. It is important that the authors apply a model of carcinogen independent induced NASH to HCC transition and test this in hepatocyte specific ACLY knock-out mice (or AAV-related as performed). In which cells parenchymal and non-parenchymal expressed is ACLY expressed in NASH induced HCC models (carcinogen independent)? Where is ACLY expressed in human NASH and human NASH-HCC? How does expression correlate with survival upon partial hepatectomy in human liver cancer patients? The effect on the lipid distribution within liver cancer is interesting but also not too strong. How does this look in non-liver cancer affected tissue? When is the expression started – is it highly expressed in B-cells? A better characterization of the tumors in terms of genetics should be done. DNA damage, location of cell death, senescence, metabolic assays could/should be done.

In general, throughout the whole manuscript it would also be nice to show the tumor nodule size and to characterize the tumors better.

A) It is important that the authors apply a model of carcinogen independent induced NASH to HCC transition and test this in hepatocyte specific ACLY knock-out mice (or AAV-related as performed).

-As discussed above (in question no. 1) the WD-CCL4 is a well-established model of MASH-HCC that closely mimics human disease. We have also completed further genetic characterized the WD-DEN

model and show it also closely replicates human disease development. Lastly, we did try the WD experiment but had very low penetrance after over 1 year on diet making this unfeasible.

B) In which cells parenchymal and non-parenchymal expressed is ACLY expressed in NASH induced HCC models (carcinogen independent)? Where is ACLY expressed in human NASH and human NASH-HCC?

-Using single-cell RNA seq we show that ACLY is expressed in both parenchymal and non-parenchymal cells of WD-DEN HCC mice liver (Extended Data Figure 1K) and genetic deletion only affects ACLY in the parenchymal tissue (Figure 2E).

Extended Data Figure 1

K

Figure 2

E

Figure 2E, Representative *Acly* KO mice liver image stained with ACLY antibody. Black star: Inflammatory cell aggregation-positive for ACLY staining. Blue arrows: Mesenchymal cells (endothelial/kupffer cells)-positive for ACLY staining. Orange arrows: Hepatocytes-Negative for ACLY staining.

-Similarly, using **WORDS REDACTED** single nuclei sequencing, we show that **SENTENCES REDACTED** ACLY is significantly higher in malignant hepatocytes compared to normal hepatocytes (Figure 6N).

FIGURES REDACTED

Figure 6N

C) How does expression correlate with survival upon partial hepatectomy in human liver cancer patients?

-We do not have access to this data to examine the relationship between ACLY and survival upon hepatectomy. We do show that like individuals who have reduced recurrence following resection (PMID: 30213589), genetic inhibition of ACLY increases the number of B-cells and TLSs in the liver.

D) The effect on the lipid distribution within liver cancer is interesting but also not too strong. How does this look in non-liver cancer affected tissue?

As shown below, biochemical assessment of liver fatty acids found that **WORDS REDACTED** generally lowered liver lipids in non-liver cancer affected tissues.

Extended Data Figure 2

D

FIGURE REDACTED

E) When is the expression started – is it highly expressed in B-cells?

We are unclear of what the reviewer means by this question.

F) A better characterization of the tumors in terms of genetics should be done. DNA damage, location of cell death, senescence, metabolic assays could/should be done.

As detailed above we have completed spatial transcriptomics of the tumors of WT and *Acly* KO and Vehicle and EVT0185-treated mice and carefully describe the genetics of the tumors, cell death, senescence and metabolic pathways that are altered.

G) In general, throughout the whole manuscript it would also be nice to show the tumor nodule size and to characterize the tumors better.

As requested, we have now assessed tumor nodule size and inserted the data into Figure 6L, Extended Data Figure 1H, 2E, 8E, 8G, and 8H to characterize the tumors better.

Figure 6

Extended Data Figure 1

Extended Data Figure 2

Extended Data Figure 8

Reviewer Question 5-Figure 2: I would be important to get an overview on the metabolomic and proteomic phenotype in liver cancer and non-tumor affected tissue in ACYL ko mice and to see over time how the latter change. Protein expression changes of ACLY in liver tissue (tumor, non tumor?) – mouse

and human? Where are the B-cells located in the liver – and which B-cell subtypes are to be found increased in the ACYL ko livers? How efficient was the depletion of B cells?

A) It would be important to get an overview on the metabolomic and proteomic phenotype in liver cancer and non-tumor affected tissue in ACYL ko mice and to see over time how the latter change.

- We have completed biochemical, **WORDS REDACTED**, and IHC analysis to provide detailed metabolic and proteomic phenotypes in liver cancer and non-tumor affected tissue of the mice with *Acly* inhibition, as shown below.

-While we have done bulk RNA sequencing at 2 time points (early and late), both of which showed similar effects, it is not feasible for us to complete this resource-intensive spatial transcriptomics and single-cell proteomics at multiple time points.

Biochemical analysis

Figure 2L

Extended Data Figure 2

D

FIGURES REDACTED

IHC analysis

Figure 2

B

C

D

B) Protein expression changes of ACLY in liver tissue (tumor, non-tumor?) – mouse and human?

As shown above (Question 5A), we show protein changes of ACLY in liver tissue (tumor, non-tumor) using IHC (Figure 2B-2D). As shown above (Question 3D), we show the ACLY mRNA expression in human non-tumor adjacent and MASH HCC samples. (Extended Figure 15D)

C) Where are the B-cells located in the liver?

As described above (Question 2 and 3A) using spatial transcriptomics, MIBI-TOF, and IHC we now describe where the B-cells are located. (Figures 5 and 6) (Extended Data Figure 11).

D) Which B-cell subtypes are to be found increased in the ACYL ko livers?

As described above using (Question 2 and 3A) spatial transcriptomics and MIBI-TOF, we demonstrate that the B-cell subtype found in the livers of *Acly* KO mice are plasma cells (Figure 5I) with further analysis using IHC and MIBI TOF highlighting a potential role of TLS (Figure 6 and Extended Data Figure 11).

E) How efficient was the depletion of B cells?

As shown below and in Extended Data Figure 14 of the revised manuscript, the depletion efficiency of B-Cell was over 99%.

Extended Data Figure 14

Figure 3: This is an impressive figure and I like very much this analysis. The identification of a novel dicarboxylic-acid prodrug that is converted to CoA thioester and inhibits ACLY through its CoA-binding site is very impressive and should get more focus in the paper – effects on immune cells? Effects on hepatocytes? Effects on hepatoma cells/ liver cancer cells?

Thank you for the positive comments and we have tried to increase the focus on the novelty of these studies in the revised manuscript.

As the activity of EVT0185 is dependent on SLC27A2, it should have no activity in the immune cells (Extended Data Figure 3J), and consistent with this hypothesis we showed EVT0185 does not affect the proliferation of B-cells or T-cells cultured in vitro (Extended Data Figure 4A-4E). In the original draft of the paper, we described the effects of EVT0185 on inhibiting lipogenesis in hepatocytes. In the revised version of the paper, we compare the effect of EVT0185 with bempedoic acid in acetate or lactate-induced denovo lipogenesis and cholesterol synthesis. (Extended Data Figure 7A-7C). As requested, we now show that EVT0185 inhibits the clonogenic survival of cultured human (Hep3B) and mouse (Hepa1-6) HCC cell lines more effectively than the ACLY inhibitor bempedoic acid (Extended Data Figure 7H, I).

Effect on hepatocytes (Line no: 245-256)

In addition to inhibiting ACLY, long-chain fatty acyl-CoAs can also activate AMPK which then inhibits fatty acid synthesis through the phosphorylation and inhibition of acetyl-CoA carboxylase (ACC)⁴⁸ (Extended Data Figure 7A). Therefore, to examine whether EVT0185 might have additional activities beyond ACLY, we assessed the incorporation of lactate into fatty acids and sterols in hepatocytes from WT and *Acly* KO mice and contrasted these effects to equimolar concentrations of bempedoic acid (Extended Data Figure 7B). We also examined the effects of EVT0185 and bempedoic acid to inhibit acetate incorporation into fatty acids and sterols (Extended Data Figure 7C) which is not dependent on ACLY activity and can be blocked through inhibition of ACC and/or acetate CoA synthetase 2 (ACSS2). In WT hepatocytes EVT0185 exerted greater potency for inhibiting fatty acid and cholesterol synthesis from lactate or acetate than bempedoic acid, a difference which became more dramatic in the absence of ACLY suggesting additional targets might be important (Extended Data Figure 7B, C).

Extended Data Figure 7

Effect on human and murine hepatoma cells (Line no: 260-262)

EVT0185 also displayed greater inhibition of clonogenic survival of cultured human (Hep3B) and mouse (Hepa1-6) HCC cells lines than bempedoic acid (Extended Data Figure 7H, I).

Figure 4: This is an interesting and in principle well performed figure. The prevention group shows a very impressive result. Also the combinatorial approach (F and G) is well done and interesting. Here I would propose again a classical diet related NASH model (like WD) and a combination of immunotherapy and anti-VEGFa – which is the current standard of care in HCC. ACLY expression and immunogenicity in human tissue - the correlation is clear – but what does it really mean and what is it based on ? Could this be done more convincingly in another assay? What are the proposed underpinnings here? Stratification for immunotherapy of HCC possible (responder non-responder)?

-Thank you for the positive comments. We intended to do this study in mice fed a WD but as described above the long duration of the experiment and low penetrance make this unfeasible, so we have now completed it in the well-characterized WD-CCl₄ model. Consistent with previous studies (PMID: 33762733), we showed that the combination of anti-PDL1/VEGFa is ineffective in mice with MASH-HCC however, a combination of EVT0185+ anti-PDL1/VEGFa is highly effective at reducing tumor burden. These results are described in lines: 291-297 and below for the reviewer.

To evaluate whether EVT0185 might also enhance response rates to the current standard of care immunotherapy Atezolizumab plus Bevacizumab¹, we completed a combination study with anti-PDL-1/VEGFR antibodies in the WD-CCl₄ model. Consistent with previous reports¹, we found that anti-PDL-1/VEGFR antibodies had minimal effects on tumor burden (Figure 4J-4L). However, the triple combination of EVT0185 + anti-PDL-1/VEGFR dramatically reduced tumor burden (Figure 4J, 4K) (Extended Data Figure 8H) and the number of animals with over 25 tumors indicating significant therapeutic benefit (Figure 4L).

Figure 4

J

K

L

Extended Data Figure 8

H

ACLY expression and immunogenicity in human tissue - the correlation is clear – but what does it really mean and what is it based on?

-We agree with the reviewer that the correlation between ACLY expression and immunogenicity in human tissue is clear. It is unclear to us what the reviewer is asking when he/she says, “but what does it really mean and what is it based on?”

-We originally evaluated the association of ACLY expression and immunogenicity by inferring immune cell type proportions within the bulk sample by using previously established methods for cell type deconvolution (PMID:32442275). Deconvolution methods are based on mathematical modelling whereby

bulk transcriptome can be decomposed to the weighted sum of cell type-specific expression profiles. These methods intake bulk transcriptome data and previously defined cell type-specific gene expression signatures and output estimates of cell type quantity.

Could this be done more convincingly in another assay?

-We now provide an additional analysis based on the co-expression pattern of *ACLY* and liver-specific B cell markers to clarify this finding. This analysis is more intuitive, based entirely on Pearson's correlation coefficient, whereby a negative coefficient suggests a negative correlation. We selected the top 10 genes that were specifically upregulated in liver-specific B cells compared to all other cell types (e.g. hepatocytes, other immune cells). These genes were identified by a separate research group and have made the data accessible to the public through the GepLiver data portal (PMID: 37301898). Results from this analysis are consistent with that of deconvolution analyses. All 9 of the 10 genes that were profiled in the human MASH-HCC microarray dataset were negatively associated with *ACLY* expression. The results are presented in Figure 6O in the revised manuscript.

Figure 6

O

What are the proposed underpinnings here? Stratification for immunotherapy of HCC possible (responder non-responder)?

-The proposed underpinning is that we have shown for the first time that metabolic reprogramming of the tumor by inhibiting *ACLY* enhanced tumor immunogenicity and specifically B-cell infiltration and reducing tumor burden. We also show that the inhibition of *ACLY* enhances the effects of Atezo-Bev combination therapy and that tumors from individuals with MASH-HCC have the highest *ACLY* and lowest b-cell infiltration.

The aims of our study do not include stratification of patient subgroups for immunotherapy treatment as in our opinion, existing clinical trials of immunotherapy in MASH-HCC lack sufficient sample size; thus underscoring the need for future research. If and when such data becomes available, we believe our

discovery that ACLY is a key gene controlling tumor immunogenicity could be used to allow stratification for immunotherapy and if high ACLY is present, as is common in MASH-HCC, could represent a novel approach to enhance immune responses to immunotherapy.

Referee #2 (Remarks to the Author):

In this study by Gautam and colleagues, a new inhibitor of ATP-citrate lyase (EVT0185) is reported and found to suppress HCC in the context of NASH. Studies in a genetic model of liver ACLY KO confirm that loss of ACLY suppresses NASH-HCC tumor growth. Both genetic ACLY KO and treatment with EVT0185 promote increased B cell infiltration of tumors, possibly due to increased Cxcl13 expression. New treatments for HCC are urgently needed, and it is encouraging that EVT0185 has similar anti-cancer effects as TKI used clinically to treat HCC in at least one mouse model. Thus the study has potential for high impact and translational relevance. However, there are several concerns with the study in its present form.

-We thank the reviewer for their insightful comments and suggestions that the study has potential high impact and translational relevance.

1. It is not clear whether ACLY is the relevant molecular target of EVT0185 in suppressing tumor growth. While it is shown that EVT0185 inhibits ACLY and that, similar to genetic ACLY KO, it reduces tumor growth, it is possible that it may be acting through additional targets to exert its anti-tumor effects. Indeed, in figure 4A, it is shown that the drug suppresses acetate-dependent lipogenesis. Since acetate is converted to acetyl-CoA for lipogenesis via ACSS2, not ACLY, this suggests that the drug's effects on lipid synthesis are not through ACLY. Perhaps it is also inhibiting ACSS2 or other enzymes in the DNL pathway. ACSS2 has previously been shown to be important for HCC tumor development, so this seems plausible (PMID: 25525877). Moreover, ACSS2 targeting has also been shown to impact anti-tumor immunity (PMID: 37723305). A key experiment is thus to treat the tumor-bearing liver ACLY KO mice with EVT0185 to determine if the drug may have additional relevant targets in this context.

-The question about what the key target of EVT0185 is interesting and potentially important, however, we do not believe that treating *Acly* KO mice with EVT0185 will answer the question as both genetic deletion and EVT0185-treated mice had virtually no tumors; thus making it unfeasible to observe further significant reductions in tumor burden.

-To address the potential additional activities of EVT0185 to ACLY we have now completed a complex experiment in which lactate and acetate tracers are delivered to WT and *Acly* KO hepatocytes (n=3 independent experiments) and treated with EVT0185 or bempedoic acid (to also address reviewer question #2). We subsequently conducted cell-free assays in vitro assay with EVT0185-CoA against AMPK, ACSS2, and ACC1/2.

The results of these experiments are described below (lines no: 245-265 in the revised manuscript).

Lines no: 245-265

In addition to inhibiting ACLY, long-chain fatty acyl-CoAs can also activate AMPK which then inhibits fatty acid synthesis through the phosphorylation and inhibition of acetyl-CoA carboxylase (ACC)⁴⁸ (Extended Data Figure 7A). Therefore, to examine whether EVT0185 might have additional activities beyond ACLY, we assessed the incorporation of lactate into fatty acids and sterols in hepatocytes from WT and *Acly* KO mice and contrasted these effects to equimolar

concentrations of bempedoic acid (Extended Data Figure 7B). We also examined the effects of EVT0185 and bempedoic acid to inhibit acetate incorporation into fatty acids and sterols (Extended Data Figure 7C) which is not dependent on ACLY activity and can be blocked through inhibition of ACC and/or acetate CoA synthetase 2 (ACSS2). In WT hepatocytes EVT0185 exerted greater potency for inhibiting fatty acid and cholesterol synthesis from lactate or acetate than bempedoic acid, a difference which became more dramatic in the absence of ACLY suggesting additional targets might be important (Extended Data Figure 7B, C). We subsequently completed additional cell-free assays and found that in contrast to studies with bempedoic acid³⁷, EVT0185-CoA inhibited, not activated, AMPK β 1 containing heterotrimers (Extended Data Figure 7D). It also inhibited both acetyl-CoA carboxylase (ACC) isozymes (ACC1) and (ACC2) as well as ACSS2 (Extended Data Figure 7E-G). EVT0185 also displayed greater inhibition of clonogenic survival of cultured human (Hep3B) and mouse (Hepa1-6) HCC cells lines than bempedoic acid (Extended Data Figure 7H, I). As activation of AMPK has been implicated in pro-survival pathways (reviewed in⁴⁹) and inhibition of ACLY often leads to compensatory upregulation of ACSS2^{8,9,12}, these data indicate important differences between EVT0185 and bempedoic acid which supported further development for HCC.

Extended Data Figure 7

-The data indicate that EVT0185 does have additional activities beyond ACly and there are several differences from bempedoic acid.

-However, with respect to the question of whether these additional activities, and specifically the inhibition ACS2, are critical for driving the anti-tumor phenotype of EVT0185, we believe this is highly unlikely for the following reason.

-As described above for reviewer #1, we have now completed detailed unbiased genomic (spatial transcriptomics) and single-cell proteomic analysis (MIBI-TOF) of tumors from both *Acly* KO and EVT0185-treated mice. Using these unbiased technologies we now show in both *Acly* KO and EVT0185-treated mice very similar changes in tumor metabolism, increases in plasma B-cells (and importantly not other types B-cells or immune cell populations), and lastly the development of tertiary lymphoid structures (Figure 5 and 6) (Extended Data Figure 10 and 11).

-Statistically speaking, we believe the odds of seeing the exact same changes in the above parameters when assessing thousands of genes simultaneously and large panel of proteins at the single cell level are incredibly low if the primary target of EVT0185 in the tumor was not ACly.

-Further supporting the importance of ACLY and not other targets our analysis of clinical populations shows that in contrast to inhibition of ACLY, there are no significant associations between ACC1/2 or ACSS2 and b-cell infiltration (Figure 6 and Extended Data Figure 15C).

-Collectively we believe these genetic data from both mice and humans strongly support our proposed model where inhibition of ACLY is the primary driver for reducing tumor burden and enhancing immunogenicity in MASH-HCC. While ACSS2 is important for compensating for the loss of ACLY in some model systems, this is not always the case as observed with pancreatic tumors (PMID: 30626590) or adipose tissue thermogenesis (PMID: 39402290). However, to address the potential for this possibility, we have inserted the following statement in the discussion lines: 501-512.

We find using spatial transcriptomics that ACSS2 was upregulated in HCC cells of *Acly* KO and EVT0185-treated mice. Despite this upregulation in *Acly* KO mice we still observed a significant reduction in tumor burden associated with enhanced plasma cells, an effect that was eliminated following the depletion of B cells indicating this was critical for the response. Subsequent studies with EVT0185 elicited similar effects on tumor burden and tumor-infiltrating B cells as observed with genetic inhibition of ACLY. Further, we show in clinical data sets from individuals with MASH-HCC that ACLY but not ACSS2 is associated with increases in tumor-infiltrating B cells. Taken together, these data strongly suggest that the primary target by which EVT0185 reduces MASH-HCC involves the inhibition of ACLY. However, it remains plausible that ACSS2 may be important for other phenotypes, such as liver steatosis and fibrosis^{51,77} which may indirectly influence tumor development in MASH-HCC; findings that will need to be explored through additional studies in distinct models.

2. Relationship of EVT0185 to the FDA-approved drug bempedoic acid is not tested or even discussed as far as I can tell- this would seem to be a key point for understanding the potential translational importance and novelty of this study. Looking at the structures of the molecules screened, including EVT0185, it is evident that they are very similar to bempedoic acid, which the authors have previously used in many prior studies, including in a related study to test efficacy in NASH treatment (PMID: 35675800). Many of the analogs tested are also very similar to those described in a 2004 study, which also impaired lipid synthesis (<https://doi.org/10.1021/jm040006p>). Furthermore, the mechanism of action of EVT0185 appears to be similar if not identical to that of bempedoic acid, and both are prodrugs that require SLC27A2 for activation. Comparisons are warranted, both in terms of IC50 for inhibition of ACLY and in terms of in vivo effects on tumor growth. Bempedoic acid is already in clinical use- what advantage does EVT0185 provide over bempedoic acid?

-As described above (answered in reviewer no. 2, question 1) we have now provided significant evidence with head-to-head comparisons of EVT0185 and Bempedoic acid at the molecular level.

This includes direct comparisons in:

4. Isolated hepatocytes from WT and *Acly* KO mice comparing effects on acetate and lactate-induced denovo lipogenesis (Extended Data Figure 7B, C)
5. Tested effects of EVT0185-CoA in additional cell-free assays (ACLY, AMPK., ACC1/ACC2 and ACSS2) (Figure 3B) (Extended Data Figure 7D-7G)

- 6. Effects of bempedoic acid in cultured Hep3B and Hepa1-6 cell lines (Extended Data Figure 7H and 7I)
- 7. An addition of a bempedoic acid arm to our WD-DEN study (Figure 4A and 4B) (Extended Data Figure 8C, D) described in lines: 277-282.

Using the WD-DEN mouse model described in Figure 1 and 2, mice were randomized based on AFP to daily gavage with vehicle, EVT0185 at 30 or 100 mg/kg or bempedoic acid at 100 mg/kg for one month (Extended Data Figure 8C, D). Compared to Vehicle, EVT0185 at 30 and 100 mg/kg reduced tumor burden by 58% and 67%, respectively, while bempedoic acid (100 mg/kg) had limited efficacy (Figure 4A and 4B).

Extended Data Figure 8

Figure 4

- We believe these three independent studies highlight an important differential response between EVT0185 and bempedoic acid that may be attributed to the increased lipophilicity of EVT0185 compared to bempedoic acid and the observation that EVT0185 inhibits, not activates the pro-survival factor AMPK as discussed in greater detail below.

3. Related to the two points above, it has been shown that bempedoic acid activates AMPK (PMID: 23118444), which suppresses ACC, and the authors previously showed that ACC inhibition can suppress HCC (PMID: 30244972). It seems plausible that AMPK activation accounts for part of the effect of EVT0185 on tumor growth. Is AMPK activated by EVT0185, similar to that reported for bempedoic acid?

-This is an interesting point, that we were also puzzled by. As described in reviewer 2, question 1, in contrast to bempedoic acid EVT0185 does not activate AMPK α 1 β 1 γ 1 containing heterotrimer. Given the emerging data from the last 5 years showing that activation of AMPK exerts pro-survival pathways in cancer cells (reviewed in PMID: 36316383) we believe this may be critical difference compared to bempedoic acid to explain the improved efficacy of EVT0185 in MASH-HCC. It is important to note that in our previous study (PMID: 30244972) with the ACC inhibitor and the ACC Ser/Ala KI mice this did not affect AMPK activity.

4. B cell role is only correlative with the presented data. Figure 2Q is the missing key control to clarify this- the effect of anti-CD20 on tumor growth in WT animals. From current data, it is unclear if the depleting B cells will equally effect WT and KO tumors or whether it specifically reverses the anti-tumor effects of ACLY KO as claimed. In addition, B cell depletion experiments should be carried out in the context of the drug treatment studies.

-We have now completed the control experiment in Figure 6J-L and show that B-cell depletion in WT animals does not reduce tumor burden.

Figure 6

-As described above we have now completed detailed spatial transcriptomics and MIBI-TOF single-cell proteomics and show very similar signatures between both *Acly* KO and EVT0185-treated mice. It is important to note that these differences are observed despite the use of different models (WD-CCl₄ and WD-DEN) and using distinct unbiased techniques making it highly infeasible the similarity is merely a function of chance.

-Therefore, while we appreciate the reviewer's suggestion to also complete the b-cell depletion in mice treated with EVT0185, due to the lack of mice available that were used to address the reviewers' other

questions (bempedoic acid (Figure 4A), Atezo-Bev combination study (Figure 4J) and Anti-CD20 in WT animals (Figure 6J) and the more than 8 months it takes to get data and significant resources that would be needed to generate these data we do not think this is justified.

5. It is shown that *Cxcl13* expression increases with *ACLY* KO, suggesting a plausible mechanism linking *ACLY* deficiency to B cell infiltration. However, this mechanism has not been explored further, either in terms of a potential functional role of *Cxcl13* or in terms of how *ACLY* might be regulating *Cxcl13* expression. Additionally, the relationship between *Cxcl13* and *Acly* should also be examined in the human dataset.

-Very little is known about the transcriptional regulation of *CXCL13*. As described above in reviewer 1, question no. 2, We show using multiple orthogonal approaches including RNAseq from cultured DEN tumors of *Acly* KO mice from the Wellen lab that *Cxcl13* is increased. (Figure 5J, P) (Figure 6F) (Extended Data Figure 10F).

- Importantly, these results in mice also extend to humans (lines 445-447) (Extended Data Figure 16C). We believe a detailed understanding of the promoter region and factors regulating *CXCL13* transcription are outside of the scope of the current manuscript.

Consistent with observations in mice, *CXCL13* was significantly reduced in MASH-HCC samples with high *ACLY* expression and significantly upregulated in MASH-HCC samples with low *ACLY* expression (Extended Data Figure 16C).

Extended Data Figure 16

C

6. T cell response has been previously reported to be linked to B cells in the context of HCC (PMID: 26669617). Figure 2K suggests that T cell infiltration also increases with *ACLY* KO. Is this important in the mechanism of tumor growth suppression? T cell depletion experiments could also be carried out.

As described above, we have completed additional spatial transcriptomic, MIBI-TOF, and IHC analysis and can clearly state that there are no substantial changes in T cell populations following deletion of *ACLY* or treatment with EVT0185 (Figure 6D) (Extended Data Figure 10A, B) (Extended Data Figure 11 E-H).

7. Proposed mechanism of inhibition needs biochemical support. The authors claim that this inhibitor is

CoA competitive by structural analysis. This claim should also be supported biochemical analysis, i.e. mode of inhibition assays. The authors also suggest that the terminal carboxyl group of EVT0185-CoA could anchor to the citrate binding pocket of ACLY. However, the inhibitor is only partially resolved where it contacts the CoA binding pocket while the rest of the molecular is disordered, suggesting that the carboxyl end is not bound. Again, mode of inhibition studies are necessary to validate the proposed model.

-This is an important point. We have now completed this analysis as requested and the data are presented in Figure 3C, 3D, and Extended Data Figure 5A. The drastic shift of K_i' compared to K_i by global fit means the compound has almost no affinity to ACLY/ CoA complex compared to free ACLY and indicates that EVT0185-CoA competes with CoA to bind ACLY, thus, the compound shows competitive inhibition for ACLY with respect to Coenzyme A. This is described below and in the revised manuscript line no: 214-217.

ACLY is inhibited by palmitoyl-CoA⁵, therefore we hypothesized that the conversion of EVT0185 to its CoA-thioester may drive the inhibition of the ACLY activity. In cell-free assays EVT0185-CoA (not the unconjugated diacid) inhibited recombinant human ACLY activity (Figure 3B) and this effect was competitive with CoA (Figure 3C, D) (Extended Data Figure 5A).

Figure 3

Figure 3. C, Michaelis-Menten Plot and D, Lineweaver-Burk Plot for ACLY with EVT0185-CoA using GraphPad Prism

Extended Data Figure 5

A

Extended Data Figure 5. A, Global fit of Michaelis-Menten Plots by GraFit software for EVT0185-CoA in ACLY.

Minor points:

8. Fig 1L: Please clarify what is measured here- free fatty acids or total saponified fatty acids

We measured free fatty acids but not total saponified fatty acids. We clarify this in method section of the revised manuscript (Line no 943 and 944).

9. Figure 2A-C- test is too small to read

As per the reviewer's suggestion, we have increased the font size to make the graphs more visible.

10. In figure 2/extended figure 3, the authors perform RNA-seq on multiple tumor timepoints. While the authors did demonstrate the experimental rationale for using multiple time points, the computational modeling strategy is not adequately explained in the text or methods.

-Method section pertaining to statistical modelling of differential gene expression analysis has now been edited and described in the revised manuscript.

Line no: 795-806

For RNA-Seq, overall and timepoint-specific differential gene expression results were obtained using the DESeq2 package v1.36.0 by modeling the additive and interaction effect of timepoint and *Acly* KO. To obtain the overall result, we adjusted association of gene expression with *Acly* KO by a binary indicator of experimental timepoint (i.e. early or late) as a confounding variable using an additive model (i.e., time + *Acly* KO). Timepoint-specific results were obtained by modelling gene-expression association using only samples belonging to either early or late timepoints. Finally, we modelled the modifying effect of timepoint on differential gene expression by using an interaction model (i.e., time × *Acly* KO). We used a false discovery rate threshold of 0.05 to determine gene significance. Individual sample read counts underwent variance stabilizing transformation. Gene ranks were obtained for downstream gene set enrichment analysis (GSEA) by performing natural logarithmic transformation of p-value and applying the numerical sign of log₂ fold change based on DESeq2 result.

11. Fig 1K: clarify what is meant by % lipid area? Does that mean within the circle drawn, the % of area that is lipid droplets? How are the lipid droplets defined- just by white area? It may be more appropriate to carry out neutral lipid staining.

-What is meant is that we have assessed the average area covered by lipid droplets in each tumor as a percent of the total field of view. To add to this analysis, we have now measured the lipid droplet-associated protein Plin2 (Perilipin 2, lipid droplet marker) using MIBI. This is in addition to the biochemical assessment of lipids (Reviewer 1 Question 5).

The results are described in lines no. 376-377.

Consistent with a reduction in the lipid droplet area, lipid droplet-associated protein (Plin2) pixel area was reduced in *Acly* KO compared to WT mice (Extended Data Figure 11B).

Extended Data Figure 11

B

12. Please clarify the statement on line 202: *Acly* KO tumors were depleted of unassigned cell types.

-Unassigned cell types were the cells that did not express any of the antibodies utilized. We have now increased the panel of antibodies being used and repeated our MIBI-analysis in *Acly* KO and EVT0185 treated mice and do not have any “unassigned” cell types.

13. Fig 4A- please clarify timeline- how long were mice on diet before the 1 month drug treatment?

-The mice were on the diet for 24 weeks before the 1 month drug treatment.

-The timeline for Fig 4A has been added in Extended Data Figure 8C of the revised manuscript.

Extended Data Figure 8

C

14. Further discussion about what is known about the role of B cells in HCC would be appreciated.

Consistent with our findings, the emerging studies that have shown the protective role of tumor infiltrating B cells along with the intratumoral TLS in HCC patients have been discussed and cited in the revised manuscript in line no: 480-485, 470-475.

Our findings indicating a critical role for B cells are consistent with emerging studies involving mouse models and HCC patients that have shown favorable outcomes linked to increased tumoral B-cell infiltration⁷⁰⁻⁷⁶. Importantly, our findings in mice showing linkages between reduced *Acly* and increased *Cxcl13* and B cells, are also observed when analyzing tumor samples from patients with MASH-driven HCC, suggesting that inhibition of ACLY may also be important for enhancing immunogenicity in humans.

Intratumoral TLS has been associated with favourable clinical outcome in several cancer types including HCC^{57,58,66-68}. B cells in TLS are known to communicate with follicular T-cells which help them to proliferate and differentiate into long-lived plasma cells that can produce antibodies and induce antitumor immunity through antibody-dependent cytotoxicity (ADCC), activation of the complement system and complement-dependent cytotoxicity⁶⁹.

Referee #3 (Remarks to the Author):

Gautam, Wu, Lally, et al. present a study of the antiproliferative effects of ATP citrate lysase (ACLY) inhibition by both genetic and chemical means. The structure of ACLY with a newly identified inhibitor, EVT0185, was determined by cryo-EM. This review focuses on the structure determination. The cryo-EM maps are of high quality and I have no concerns about their generation or interpretation. I commend the authors for showing model-in-map fit in figures where necessary to support their interpretation.

1. Line 281-2 should be re-written for appropriate use of terminology and rounding of resolution (using the odd-even rule) to one decimal place. E.g.: "A 3D reconstruction without the application of symmetry had a resolution of 3.6 Å following a gold-standard refinement". Note the use of "gold" not "golden". Also, it is the refinement that is gold standard, which means keeping the half maps separate, not the FSC (although people often say "gold standard FSC, as in line 817 here). The phrase "in C1 symmetry" used throughout the manuscript is strange. Maps are not in a symmetry but have symmetry applied. Further, C1 symmetry means no symmetry was applied.

-We implemented the textual changes as kindly provided by reviewer #3.

-We indeed used the phrase "C1 symmetry" to indicate that no symmetry was applied. We have now replaced the phrase "in C1 symmetry" with "without application of symmetry" or "without symmetry applied" in line no. 228-229.

2. Line 791-792 should be improved to address the same points as line 281-2.

-We implemented the textual changes as kindly provided by reviewer #3 in 1116-1117.

3. A reference should be added for non-uniform refinement in the methods section.

-We added the following reference to the methods section:

Punjani, A., Zhang, H. & Fleet, D.J. Non-uniform refinement: adaptive regularization improves single-particle cryo-EM reconstruction. Nat Methods 17, 1214–1221 (2020). <https://doi.org/10.1038/s41592-020-00990-8>

4. Figure ED10 is useful but the image quality should be improved: it is hard to read the text when zoomed in.

-We have now provided a version of Figure ED10 with a better image quality. The result has been presented in Extended Data Figure 6.

Extended Data Figure 6

5. Something does not look right with the model-to-map FSC. The falloff in FSC is far more abrupt than I would expect. It looks as if either the map or the model-derived-map was filtered to 3.37 Å before calculation of the FSC (or perhaps this appearance is a result of how the FSC is plotted). The authors should double check that everything is as it should be.

We thank the reviewer for pointing this out. The model-to-map FSC was indeed calculated against a sharpened map. We now provide the model-to-map FSC calculated against the unsharpened and unfiltered full map.

6. Resolutions should be generally stated to one decimal place throughout the manuscript and in figures, although it is not necessary to change the FSC output from cryoSPARC in Fig. S10.

Resolutions are now reported to one decimal place throughout the manuscript.

I do not know what “contoured at 0.176 V”, “contoured at 0.12 V”, or “contoured at 0.2 V” means in the

figure captions (Figure 3, Figure ED9) and in the text (lines 952-957). What is “V”? Is it a fraction of the highest density value in the map? Please define.

We thank the reviewer for pointing out this error. The symbol “V” was referring to the unit (Volt) as shown in Coot. We have omitted the symbol V and now denote the contour level as a unit-less number on lines no: 1369, 1371, 1375, 1498, and 1502 of the revised manuscript.

Referees' comments:

Referee #1 (Remarks to the Author):

In the revised version of this manuscript the authors have tried to answer most of the questions raised by the Editors and the referees. Most of the responses were convincing to this referee and indicate the mode of action of ATP citrate lyase inhibition - however, in my point of view there are several points not understood.

What is not convincing to me at that point is whether this therapy will work in pathophysiological state of the art model on MASH HCC - like WD induced liver cancer (without the use of chemical carcinogens).

DEN-WD is not a mouse model in which MASH is causally linked to liver cancer (like in the human situation). It is a mouse model in which cancer is induced by DEN and then accelerated by WD (due to its immunosuppressive, proproliferative function of the diet).

The reason why I am pertinent on this is the fact that DEN (used in their models) is a genotoxic and alkylating agent, that will induce a lot of tumor neoantigens - thus driving an immune response against tumor cells. Thus, the effects on anti-tumor surveillance are artificial and hyperreactive.

The authors state that due to time reasons they were not able to do these experiments, however I am not sure whether this is an argument not to do the most important experiment of the manuscript when it comes to application in human liver cancer caused by MASH - according to the standard of the field (see also PMID: 11199145 DOI: 10.1038/s42255-024-01043-6).

-We acknowledge the reviewer's comments highlighting the importance of evaluating ACLY inhibition effect in WD-HCC model.

-We demonstrate the anti-tumor effect of EVT0185 in the WD-HCC model and prove that ACLY inhibition therapy works in the pathophysiological state-of-the-art model on MASH-HCC without using a chemical carcinogen. The results are described in lines 289-293 of the revised manuscript.

Line no: 289-293

To examine the effects of EVT0185 in animals in which DEN or CCl₄ was not utilized, mice were fed a WD for 18-months, and out of 30 mice, 10 with detectable AFP were randomized into two groups (Extended Data Figure 8H) and treated with either vehicle or EVT0185 (100 mg/kg) for four weeks. Consistent with the WD-DEN and WD-CCl₄ models, EVT0185 dramatically reduced the number of tumors (Figures 4F and 4G).

Extended Data Figure 8

H

Figure 4

F

G

2. This also relates to their findings on B-cells - it is not unexpected/novel (B cell might prime T cells; they might also potentially actually contribute to killing) but novel and not clear to me would rather be to understand what the role of B cells exactly is. Will they interact/collaborate with T cells (CD8 killers) or even NK cells? Would the authors see an effect when combining anti-CD8 with anti-B-cell depletion.

To understand the interaction/collaboration of B cells with other immune cells in the tumor microenvironment, we performed cellular neighborhood analysis on a subset of MIBI images representative of the invasive margin or leading edge of the tumors.

We did not find the abundance of CD8 T cells above the threshold (1 in 10,000) for inclusion in the cellular neighborhood analysis. However, B cells, CD4 T cells, CD11c DC/Macrophages, and NK cells did satisfy this threshold for analysis. The analysis revealed that CD4 T cells and CD11c DC/Macrophages were in close proximity to B cells. This spatial feature that describes an area within the tumor microenvironment rich in cells classically involved in antigen presentation activities (B cells, CD4 T cells, and CD11c DC/Macrophages) determined by cellular neighborhood analysis was named as "Antigen Presenting Complex". The region with antigen-presenting complex was significantly enriched in *Acly* KO compared to WT tumors.

In addition, NK cells were found to be mapped to the areas outside this region, indicating no interaction with B cells based on this analysis.

The result has been described in lines 393-396.

Furthermore, MIBI spatial neighborhood and phenotype map analysis revealed that in *Acly* KO mice, B cells in the tumor periphery were in close proximity to cells involved in antigen presentation (CD11c+Dendritic cells/macrophages and CD4+ T cells) compared to WT controls (Figure 6G-6I) (Extended Data Figure 11F).

Figure 6

Extended Data Figure 11

F

Intratumoral TLS (Tertiary Lymphoid Structure) has been associated with favourable clinical outcomes in several cancer types, including HCC (PMID: 30213589, PMID: 38036041). Furthermore, B cells associated with TLS play a crucial role in anti-tumor immunity (PMCID: PMC4337382). In the present study, we have observed B cells in TLS in the *Acl*y KO/EVT0185-treated group compared to the WT/Vehicle-treated group (Figure 6F) (Extended Data Figure 11E, M). Along with the MIBI cell neighborhood analysis presented above, we found that an EVT0185-treated tumor has a germinal center-like structure comprised of antigen-presenting cells (CD3-CD11c+) and follicular T (CD3+CD4+) and B cells (CD3-CD19+) compared with a vehicle-treated tumor (Extended Data Figure 11N).

B cells in TLS are known to communicate with follicular T-cells, which help them to proliferate and differentiate into long-lived plasma cells (PMCID: PMC4337382) that can produce antibodies and induce antitumor immunity through antibody-dependent cytotoxicity (ADCC), activation of the complement system, and complement-dependent cytotoxicity (PMCID: PMC4337382). In the present study, our scRNA sequencing on B cells in the WD-CCl₄ model indicated that the pathways related to complement activation and complement-dependent cytotoxicity were significantly upregulated in the EVT0185-treated group (Extended Data Figure 11O) compared to the vehicle control, indicating this is the tumor-killing mechanism.

Extended Data Figure 11

O

3. It is also interesting to see that in their models the combination of immunotherapy and anti VEGFR2 is not made better by the addition of ACLY inhibition, when it comes to tumor load.

Also the human data are not clear to me - why did the authors not look into immunotherapy trials or cohorts of HCC patients with and without MASH and Atezo/Bev- when it comes to their markers?

A) It is also interesting to see that in their models the combination of immunotherapy and anti VEGFR2 is not made better by the addition of ACLY inhibition, when it comes to tumor load.

- We show in Figure 4K-4M and Extended Data Figure 8J of the manuscript that the addition of EVT0185 with anti-PDL1+anti VEGFR2 reduces tumor burden.

B) Also the human data are not clear to me - why did the authors not look into immunotherapy trials or cohorts of HCC patients with and without MASH and Atezo/Bev- when it comes to their markers?

-We appreciate the reviewer pointing us in this direction. We did not initially undertake this analysis as MASH etiology has not always been clearly defined in existing HCC clinical trials and was usually non-specifically classified by non-viral HCC. Therefore, specific conclusions on markers of immunotherapy response among patients with MASH-HCC may not be possible. Nevertheless, we attempted to obtain linked transcriptome and clinical etiology data from the IMBrave150 and GO30140 trials (PMID: 35739268) which we believe the reviewer was referring to. We found transcriptome data deposited in the European Genome Archive under accession number EGAD00001008128 while disease etiology data was stored in the Roche Vivli platform. We attempted data linkage however, as per Roche data sharing policy described below this is not possible (<https://vivli.org/ourmember/roche/>):

“If individual patient data from the same study is made available outside of Vivli this can not and should not be linked to data on the Vivli platform by researchers external to Roche due to a potential increase in risk of patient re-identification. Requests to merge patient-level data for a study with additional data from the same study hosted on another platform (e.g. European Genome Archive) is not possible.”

In the manuscript we have provided strong evidence that consistent with observations in mice that inhibition of tumor ACLY in humans is linked to increases in Cxcl13 and B-cell infiltration (Figures 5D, F, J, L and P, 6P-Q) (Extended Data Figures 10F, 11C, 16C). We also show that the combination of EVT0185 with the equivalent of Atezo-Bev therapy in mice improves therapeutic responses (Figure 4K-M) (Extended Data Figure 8J). These data strongly support the clinical relevance of our observations. As more patients with MASH-HCC are treated with Atezo-Bev therapy, future studies analyzing these genetic data will be needed to stratify non-responders, but unfortunately this is currently not possible and is therefore outside of the scope of the current study.

4. In summary, this is an interesting manuscript, that in my view lacks one important experiment and does not address several obvious points (human cohorts, role of B cells and T cells) - for the reasons discussed above I believe that the latter experiments are needed to make a sufficiently important claim - as this manuscript intends to do.

As detailed above we have now addressed all reviewer's points to support our claims.

Referee #2 (Remarks to the Author):

1. The authors have addressed my concerns. The manuscript is greatly improved and I believe will be impactful. I congratulate the authors on an excellent and exciting study.

We appreciate the reviewer's feedback and suggestions to improve the manuscript.

2. One recommended additional citation for the discussion is that aligning with this study, a prior study also showed that ACLY targeting by shRNA or bempedoic acid in liver cancer cells sensitized to anti-PDL1 (PMID: 38055816).

We have cited the paper in line 100 of the revised manuscript.

Referee #3 (Remarks to the Author):

The reviewers have addressed my concerns about the cryo-EM aspects of the manuscript. The only exception is related to stating the contour level of the map. It is not strictly necessary to state the contour level each time one shows the map. However, if it is to be stated, it shouldn't be as an arbitrary value. If the authors wish to state the contour level they could normalize the map (e.g. in Chimera) so that the levels correspond to the standard deviation of the noise in the map. They could then specify the contour level in terms of sigma.

We appreciate the reviewer's feedback and have updated the manuscript to include map contour levels only in main Figure 3. Both the sigma-level and the absolute level are now specified in lines 1412-1415.

The sharpened map is contoured at 8.74 sigma (absolute level = 0.176) and colored by the different structural domains as in panel C. The red dashed line indicates the region used for local refinement. **H, I**, Sharpened cryo-EM map contoured at 9.94 sigma (absolute level = 0.12).

Additionally, we have added the following statement to the Methods section (Lines 1191-1192)

Cryo-EM map contour sigma-levels are reported based on map normalization in Coot.